# ROBUSTNESS IN THE FACE OF PARTIAL IDENTIFIABILITY IN REWARD LEARNING

**Filippo Lazzati**
Politecnico di Milano
Milan, Italy
filippo.lazzati@polimi.it

**Alberto Maria Metelli***
Politecnico di Milano
Milan, Italy
albertomaria.metelli@polimi.it

## ABSTRACT

In Reward Learning (ReL), we are given *feedback* on an unknown *target reward*, and the goal is to use this information to recover it in order to carry out some downstream *application*, e.g., planning. When the feedback is not informative enough, the target reward is only *partially identifiable*, i.e., there exists a set of rewards, called the *feasible set*, that are equally plausible candidates for the target reward. In these cases, the ReL algorithm might recover a reward function different from the target reward, possibly leading to a failure in the application. In this paper, we introduce a general ReL framework that permits to *quantify* the drop in "performance" suffered in the considered application because of identifiability issues. Building on this, we propose a *robust* approach to address the identifiability problem in a principled way, by maximizing the "performance" with respect to the worst-case reward in the feasible set. We then develop **Rob-ReL**, a ReL algorithm that applies this robust approach to the subset of ReL problems aimed at assessing a preference between two policies, and we provide theoretical guarantees on sample and iteration complexity for **Rob-ReL**. We conclude with some numerical simulations to illustrate the setting and empirically characterize **Rob-ReL**.

## 1 INTRODUCTION

Reward Learning (ReL) is the problem of learning a reward function from data (Jeon et al., 2020). When the data are demonstrations, ReL is known as Inverse Reinforcement Learning (IRL) (Russell, 1998), whereas when the data are (pairwise) comparisons of trajectories, ReL is usually called Preference-based Reinforcement Learning (PbRL) (Wirth et al., 2017) or Reinforcement Learning from Human Feedback (Kaufmann et al., 2024).

The main strength of ReL is that the reward function that it aims to learn, referred to as the *target reward*, corresponds to "a succinct and transferable representation of the preferences of an agent" (Russell, 1998; Arora & Doshi, 2021). As such, *ideally*, ReL allows the use of datasets of demonstrations and comparisons for a variety of important applications, such as reward design (Hadfield-Menell et al., 2017), Imitation Learning (IL) (Abbeel & Ng, 2004), risk-sensitive IL (Lacotte et al., 2019), preference inference (Hadfield-Menell et al., 2016), behavior transfer across environments (Fu et al., 2017), behavior improvement (Syed & Schapire, 2007), and, more generally, any task that can be carried out using a reward.

However, in practice, ReL has been successfully applied mainly to IL (Finn et al., 2016) and reward design (Christiano et al., 2017). The primary obstacle to the adoption of ReL algorithms for other applications is *partial identifiability* (Cao et al., 2021; Kim et al., 2021; Skalse et al., 2023). This arises when the available feedback (demonstrations, comparisons, or otherwise) does not allow to *uniquely* identify the target reward, but instead leads to a set of rewards (referred to as the *feasible set* Metelli et al. (2021; 2023)) that represent equally-plausible candidates for the target reward. As a result, the recovered reward might differ from the target reward, potentially causing failure in the downstream application. As noted by several works (Cao et al., 2021; Skalse et al., 2023; Finn et al., 2016), most existing ReL methods, including (Ng & Russell, 2000; Ziebart et al., 2008; Boularias et al., 2011; Wulfmeier et al., 2016; Christiano et al., 2017), are sensitive to this issue.

---

*Corresponding author.

The standard solution in the literature is to try to ensure that the feasible set contains (almost) only the target reward. This is typically achieved by collecting additional feedback (Amin & Singh, 2016; Cao et al., 2021; Schlaginhaufen & Kamgarpour, 2024; Lazzati & Metelli, 2024) or by imposing further assumptions on the available feedback (Kim et al., 2021). However, in practice, additional feedback may not be available, and the added assumptions may be too strong.

A more powerful and general approach was recently proposed by Skalse et al. (2023). Rather than requiring the target reward to be uniquely identifiable, they instead ask that the feasible set contains only reward functions that are "equivalent" to the target reward *w.r.t. the considered application*. Intuitively, this milder condition is applicable only when we have prior knowledge of the intended use (i.e., the application) of the learned reward, which is almost always the case (Ng & Russell, 2000; Ziebart et al., 2008; Fu et al., 2017; Christiano et al., 2017).

However, this approach has two drawbacks. First, it is *difficult to apply* in practice because, except for simple feedback and applications, it is non-trivial to verify whether the equivalence condition holds. Second, it is *qualitative*: if the feasible set contains a reward that is not equivalent to all others, then Skalse et al. (2023) classify the ReL problem as prone to failure, without quantifying how severe the difference is. Intuitively, if this is small, the downstream application might still be carried out nearly successfully.

In this paper, we present a novel general framework for ReL that enables *quantitative* considerations. Based on this, we propose an *easy-to-apply* robust approach for addressing the identifiability problem.

**Contributions.** The contributions of this paper are summarized as follows.

- We introduce a new quantitative framework for ReL (Section 3).
- We propose a robust approach for tackling the identifiability problem (Section 4).
- We present `Rob-ReL`, an efficient algorithm implementing the robust approach (Section 5).
- We conduct some simulations to characterize the problem setting and the method (Section 6).

All results are proved in Appendix E, while additional related work is discussed in Appendix A.

## 2 PRELIMINARIES

**Notation.** Given $N \in \mathbb{N}$, we denote $[\![N]\!] := \{1, \ldots, N\}$. Given a finite set $\mathcal{X}$, we denote by $|\mathcal{X}|$ its cardinality and by $\Delta^{\mathcal{X}}$ the probability simplex on $\mathcal{X}$. Given two sets $\mathcal{X}$ and $\mathcal{Y}$, we denote the set of conditional distributions as $\Delta_{\mathcal{Y}}^{\mathcal{X}} := \{q : \mathcal{Y} \to \Delta^{\mathcal{X}}\}$. We use $\mathbb{R}_+^k$ to denote the non-negative orthant in $k$ dimensions. A vector $v \in \mathbb{R}^k$ is a subgradient for a function $h : \mathbb{R}^k \to \mathbb{R}$ at $u \in \mathbb{R}^k$ if, for all $w \in \mathbb{R}^k$ in the domain of $h$, it holds that $h(w) \geqslant h(u) + v^{\mathsf{T}}(w - u)$. Sometimes, we use $\langle v, w \rangle = v^{\mathsf{T}} w$ for the dot product of vectors $v, w \in \mathbb{R}^k$. We say that a function $d : \mathcal{X} \times \mathcal{X} \to \mathbb{R}_+$ is a *premetric* if, for all $x \in \mathcal{X}$, we have $d(x, x) = 0$. Moreover, for any $x \in \mathcal{X}$, we denote the $\ell_2$-projection onto a set $\mathcal{Y}$ as any point such that: $\Pi_{\mathcal{Y}}(x) \in \arg\min_{y \in \mathcal{Y}} \|x - y\|_2$.

**Markov Decision Processes (MDPs).** A finite-horizon *Markov decision process* (MDP) (Puterman, 1994) is defined as a tuple $\mathcal{M} := (\mathcal{S}, \mathcal{A}, H, s_0, p, r)$, where $\mathcal{S}$ is the finite state space ($S := |\mathcal{S}|$), $\mathcal{A}$ is the finite action space ($A := |\mathcal{A}|$), $s_0 \in \mathcal{S}$ is the initial state, $H \in \mathbb{N}$ is the horizon, $p \in \Delta_{\mathcal{S} \times \mathcal{A} \times [\![H]\!]}^{\mathcal{S}}$ is the transition model, and $r \in \mathfrak{R} := \{r : \mathcal{S} \times \mathcal{A} \times [\![H]\!] \to [0, 1]\}$ is the reward. A policy is a mapping $\pi \in \Pi := \Delta_{\mathcal{S} \times [\![H]\!]}^{\mathcal{A}}$. We let $\mathbb{P}_\pi$ denote the probability distribution induced by $\pi$ in $\mathcal{M}$ starting from $s_0$ (we omit $s_0, p$ for simplicity), and $\mathbb{E}_\pi$ denote the expectation w.r.t. $\mathbb{P}_\pi$. The visitation distribution induced by $\pi$ in $\mathcal{M}$ is defined as $d_h^\pi(s, a) := \mathbb{P}_\pi(s_h = s, a_h = a)$ for all $s, a, h$, so that $\sum_{(s,a) \in \mathcal{S} \times \mathcal{A}} d_h^\pi(s, a) = 1$ for every $h \in [\![H]\!]$. We denote the set of all state-action trajectories as $\Omega := (\mathcal{S} \times \mathcal{A})^H \times \mathcal{S}$. Given a trajectory $\omega = (s_1, a_1, \ldots, s_H, a_H, s_{H+1}) \in \Omega$, we define the "visitation distribution" $d^\omega$ of $\omega$ at each $s, a, h$ as $d_h^\omega(s, a) = \mathbb{1}\{s = s_h, a = a_h\}$. Moreover, we let $G(\omega; r) := \sum_{h \in [\![H]\!]} r_h(s_h, a_h)$ be the return of $\omega$ under reward $r \in \mathfrak{R}$, and note that $G(\omega; r) = \langle d^\omega, r \rangle$. We denote the expected return of a policy $\pi$ in MDP $\mathcal{M}$ as $J^\pi(r; p) := \mathbb{E}_\pi[\sum_{h \in [\![H]\!]} r_h(s_h, a_h)] = \langle d^\pi, r \rangle$, the optimal policy $\pi^*$ as any policy in $\arg\max_\pi J^\pi(r; p)$, and the optimal expected return as $J^*(r; p) := \max_\pi J^\pi(r; p)$. Finally, for any $\beta \geqslant 0$ and stochastic policy $\pi$, we let $J_\beta^\pi(r; p) := \mathbb{E}_\pi[\sum_{h \in [\![H]\!]} (r_h(s_h, a_h) - \beta \log \pi_h(a_h|s_h))]$ be the entropy-regularized return (Ziebart, 2010; Haarnoja et al., 2017).

| Feeback $f$ | Feedback type and $Q_f$ | Feasible set $\mathcal{R}_f$ |
|---|---|---|
| optimal expert (Ng & Russell, 2000) | demonstrations $\pi^E$ | $\{r : J^{\pi^E}(r;p) = J^*(r;p)\}$ |
| $\beta$-MCE expert (Ziebart, 2010) | | $\{r : \pi^E = \arg\max_\pi J_\beta^\pi(r;p)\}$ |
| $t$-suboptimal expert (Poiani et al., 2024), $t \geqslant 0$ | | $\{r : J^{\pi^E}(r;p) \geqslant J^*(r;p) - t\}$ |
| BTL with $q$ (Christiano et al., 2017) | trajectory comparison $(\omega^1, \omega^2)$ | $\{r : q = e^{G(\omega^1;r)}/\sum_{i\in\{1,2\}} e^{G(\omega^i;r)}\}$ |
| hard preference (Jeon et al., 2020) | | $\{r : G(\omega^1;r) \leqslant G(\omega^2;r)\}$ |
| BTL with $q$ (*new*) | policy comparison $(\pi^1, \pi^2)$ | $\{r : q = e^{J^{\pi^1}(r;p)}/\sum_{i\in\{1,2\}} e^{J^{\pi^i}(r;p)}\}$ |
| hard preference (*new*) | | $\{r : J^{\pi^1}(r;p) \leqslant J^{\pi^2}(r;p)\}$ |

Table 1: A list of some feedback considered in literature. For simplicity, we have grouped different feedback $f$ based on the quantity $Q_f$, obtaining the three categories in the first column. Note that the *policy comparison* feedback are introduced in this paper for the first time and capture the situation in which we are given a preference on the behavior of two other agents (more in Appendix B). MCE stands for "Maximum Causal Entropy", while BTL abbreviates the "Bradley-Terry-Luce" model.

**Reward Learning (ReL).** In the literature (Russell, 1998; Jeon et al., 2020; Skalse et al., 2023), ReL is defined as the problem of learning an unknown *target reward* $r^\star$ from a certain amount of *feedback*, i.e., data, like demonstrations (Ng & Russell, 2000) or trajectory comparisons (Wirth et al., 2017), that "leak information" about $r^\star$. The ultimate goal is to use the recovered reward for some downstream *application* (Skalse et al., 2023), such as finding the optimal policy (planning). The concept of *partial identifiability* (Cao et al., 2021; Kim et al., 2021; Skalse et al., 2023) refers to the existence of multiple rewards that are equally plausible candidates for the target reward with respect to the given feedback. This set of rewards is called the *feasible set* (Metelli et al., 2021; 2023).

## 3 A Quantitative Framework for Reward Learning

In this section, we present a new framework for studying ReL problems. Beyond modeling feedback in a simple yet flexible way, our framework crucially models applications in a *quantitative* manner,[1] paving the way to new approaches to ReL (e.g., see Section 4).

In our framework, we define a ReL problem (Russell, 1998; Jeon et al., 2020) as a pair $(\mathcal{F}, g)$, where $\mathcal{F} = \{f_i\}_i$ is a set of *feedback* and $g$ is an *application*. Informally, the feedback $\mathcal{F}$ represent what we know about the unknown target reward $r^\star$, while the application $g$ represents what we want to do with it. In the following two sections, we formalize these important concepts.

### 3.1 Feedback

A feedback $f$ relates a known quantity $Q_f$ with the unknown target reward $r^\star$. We consider as feedback only those statements that can be translated into a constraint on $r^\star$ of the type $r^\star \in \mathcal{R}_f$, where $\mathcal{R}_f \subseteq \mathfrak{R}$ is some set of rewards associated with feedback $f$, that we call *feasible set*. See Table 1 for a list of popular feedback and their corresponding feasible sets.

For instance, saying that "*policy $Q_f = \pi^E$ is optimal for $r^\star$*" (i.e., the "optimal expert" (Ng & Russell, 2000) entry in Table 1) is an example of feedback drawn from the IRL literature, and it is equivalent to saying that $r^\star \in \mathcal{R}_f = \{r \in \mathfrak{R} : J^{\pi^E}(r;p) = J^*(r;p)\}$. Another example of feedback, taken from the PbRL literature, is "*trajectories $Q_f = (\omega^1, \omega^2)$ are such that the return under $r^\star$ of $\omega^1$ is no more than that of $\omega^2$*" (i.e., the "hard preference" (Jeon et al., 2020) entry in Table 1), and corresponds to $r^\star \in \mathcal{R}_f = \{r \in \mathfrak{R} : G(\omega^1;r) \leqslant G(\omega^2;r)\}$. Note that our formulation is very flexible and allows us to work with almost any feedback we desire, such as "*given $Q_f = (s,a,\omega)$, the reward $r^\star$ of the pair $(s,a)$ is 80% of the return of $\omega$*", corresponding to $r^\star \in \mathcal{R}_f = \{r \in \mathfrak{R} : r(s,a) = 0.8 \cdot G(\omega;r)\}$.

If we are given multiple feedback $\mathcal{F} = \{f_i\}_i$, then we can combine them to obtain a smaller feasible set $\mathcal{R}_\mathcal{F}$ of candidates for $r^\star$. Formally, we define the feasible set of $\mathcal{F}$ as the intersection $\mathcal{R}_\mathcal{F} := \bigcap_i \mathcal{R}_{f_i}$ of the feasible sets of all the feedback in $\mathcal{F}$. Note that $\mathcal{R}_\mathcal{F} \subseteq \mathcal{R}_{f_i}$ for every $i$, meaning that combining multiple feedback permits to reduce our "uncertainty" on $r^\star$. If $\mathcal{R}_\mathcal{F} \neq \{r^\star\}$, then we suffer from *partial identifiability*.

---

[1]In Appendix A.2, we provide a comparison of our framework with the *qualitative* framework of Skalse et al. (2023), while in Appendix D we provide a quantitative discussion on model selection through our new framework.

| Application $g$ | Set $\mathcal{X}_g$ | Loss $\mathcal{L}_g(r,x)$ |
|---|---|---|
| Imitation of $\overline{\pi}$ (Abbeel & Ng, 2004) | $\Pi$ | $\|J^{\overline{\pi}}(r;p) - J^x(r;p)\|$ |
| Planning in $p'$ (Christiano et al., 2017; Fu et al., 2017) | $\Pi$ | $J^*(r;p') - J^x(r;p')$ |
| Constrained planning with $c,k$ (Schlaginhaufen & Kamgarpour, 2023) | $\Pi_{c,k}$ | $\max_{\pi \in \Pi_{c,k}} J^\pi(r;p) - J^x(r;p)$ |
| Assessing a trajectory preference $\omega^1, \omega^2$ | $\mathbb{R}$ | $\|x - (G(\omega^1;r) - G(\omega^2;r))\|$ |
| Assessing a policy preference $\pi^1, \pi^2$ | $\mathbb{R}$ | $\|x - (J^{\pi^1}(r;p) - J^{\pi^2}(r;p))\|$ |
| Learning a reward (Ramachandran & Amir, 2007) | $\mathfrak{R}$ | $\|x - r\|_2$ |

Table 2: A list of some applications considered in literature. Note that we used $\Pi_{c,k} := \{\pi : J^\pi(c;p) \leqslant k\}$, where $c$ is the cost and $k$ the threshold.

## 3.2 Applications

We define an application $g$ as a pair $(\mathcal{X}_g, \mathcal{L}_g)$, where $\mathcal{X}_g$ is a set, and $\mathcal{L}_g : \mathfrak{R} \times \mathcal{X}_g \to \mathbb{R}_+$ is a "loss" function. An application $g$ is *carried out* by choosing an $x \in \mathcal{X}_g$, which results in suffering from a loss $\mathcal{L}_g(r^\star, x)$ (see examples in Table 2). To *solve a ReL problem* $(\mathcal{F}, g)$, we must carry out the application $g$ while incurring the minimum possible loss, i.e., we must select an object $x \in \mathcal{X}_g$ for deployment such that the loss $\mathcal{L}_g(r^\star, x)$ is as small as possible. Thus, ideally, the goal is to output:

$$x^\star \in \underset{x' \in \mathcal{X}_g}{\arg\min} \, \mathcal{L}_g(r^\star, x').$$

However, a ReL problem is not an optimization problem because the target reward $r^\star$, and therefore the loss $\mathcal{L}_g(r^\star, \cdot)$, are unknown. For this reason, the function $\mathcal{L}_g$ is defined over *all* rewards $r \in \mathfrak{R}$, with the meaning that $\mathcal{L}_g(r, \cdot)$ quantifies the *loss we would suffer if $r$ were the target reward $r^\star$*.

An example of application $g$ is the well-known IL problem (Ho et al., 2016; Osa et al., 2018) (see "imitation of $\overline{\pi}$" in Table 2), where we aim to output a policy $x$ that "imitates" some given policy $\overline{\pi}$, i.e., that matches its expected return under the unknown $r^\star$. Thus, we can set $\mathcal{X}_g = \Pi$ and $\mathcal{L}_g(r,x) = \|J^{\overline{\pi}}(r;p) - J^x(r;p)\|$, with the intuition that, if $r = r^\star$, then the error of $x$ is $\mathcal{L}_g(r,x)$.

Another example of a ReL application $g$ is planning (see "planning in $p'$" in Table 2), which arises in many contexts, including reward design (Christiano et al., 2017) and transferring behavior (Fu et al., 2017). Here, we aim to find a policy $x$ with the largest possible expected return under $r^\star$ in some environment with different dynamics $p'$. Thus, we have $\mathcal{X}_g = \Pi$ and $\mathcal{L}_g(r,x) = J^*(r;p') - J^x(r;p')$.

As a final example, consider the problem of assessing how much a trajectory $\omega^1$ is preferred to $\omega^2$ by some agent (see Table 2). Assuming that $r^\star$ models the agent's preferences, we can view this problem as an application $g$ where $\mathcal{X}_g = \mathbb{R}$ and $\mathcal{L}_g(r,x) = \|x - (G(\omega^1;r) - G(\omega^2;r))\|$.

Note that our framework can also be used in scenarios where the ultimate goal is learning $r^\star$ (e.g., because the application $g$ is not known yet) by setting $\mathcal{X}_g = \mathfrak{R}$ and using some distance between rewards for $\mathcal{L}_g$, e.g., $\mathcal{L}_g(r,x) = \|x - r\|_2$ (Ramachandran & Amir, 2007) (see more in Appendix C).

**Remark 3.1.** *In this section, we considered feedback $f$ and applications $g$ that are fully known, in the sense that all the quantities (e.g., policies, transition models and other parameters) involved in the definitions of $\mathcal{R}_f, \mathcal{X}_g, \mathcal{L}_g$ are known exactly. However, in practice, these quantities are unknown and must be estimated from finite samples. We will consider the finite-sample regime in Section 5.*

## 4 A Robust Approach to Tackle Partial Identifiability

In the previous section, we presented a framework for formalizing ReL problems. In this section, we introduce a novel, principled way to *solve* a ReL problem $(\mathcal{F}, g)$, i.e., to select the object $x \in \mathcal{X}_g$ to deploy. We begin by reviewing the existing approaches adopted in the literature.

**Existing approaches.** In the literature, the majority of existing ReL methods, including the most popular IRL (Ziebart et al., 2008; Boularias et al., 2011; Wulfmeier et al., 2016; Finn et al., 2016) and PbRL (Christiano et al., 2017; Ibarz et al., 2018; Jeon et al., 2020) algorithms, solve a ReL problem $(\mathcal{F}, g)$ by first drawing an *arbitrary* reward $\widetilde{r}$ from the feasible set $\mathcal{R}_\mathcal{F}$, and then deploying the object $\widetilde{x}$ that minimizes the "loss" $\mathcal{L}_g(\widetilde{r}, \cdot)$ w.r.t. the recovered reward $\widetilde{r}$, as if $\widetilde{r}$ were the true target reward $r^\star$:

$$\widetilde{x} \in \underset{x' \in \mathcal{X}_g}{\arg\min} \, \mathcal{L}_g(\widetilde{r}, x'). \tag{1}$$

However, there are two main problems with this approach: $(i)$ there is no clear motivation for why this choice of $\widetilde{x}$ is a "good" way for solving the ReL problem, because in general $\widetilde{r} \neq r^\star$ (Amin & Singh, 2016; Cao et al., 2021; Kim et al., 2021), and so the minima of $\mathcal{L}_g(\widetilde{r}, \cdot)$ might incur a very large value of the true loss $\mathcal{L}_g(r^\star, \cdot)$; $(ii)$ none of the aforementioned works provides an estimate of the true loss $\mathcal{L}_g(r^\star, \widetilde{x})$ incurred by selecting $\widetilde{x}$, thus providing no information on whether the chosen $\widetilde{x}$ can be safely deployed or not. In the next two paragraphs, we present our approach to overcome these limitations $(i)$ and $(ii)$.

**Our approach.** We propose a *robust* approach that arises quite naturally once the framework introduced in Section 3 is adopted. Specifically, the idea is that, whatever choice $x' \in \mathcal{X}_g$ we make, since we know that the target reward $r^\star$ belongs to the feasible set $\mathcal{R}_\mathcal{F}$, then, in the worst-case, the true loss $\mathcal{L}_g(r^\star, x')$ incurred by deploying $x'$ is upper bounded as:

$$\mathcal{L}_g(r^\star, x') \leqslant \max_{r \in \mathcal{R}_\mathcal{F}} \mathcal{L}_g(r, x').$$

For this reason, we propose to deploy the object $x_{\mathcal{F},g} \in \mathcal{X}_g$ that minimizes the loss associated with the worst possible value that $r^\star$ can take:

$$x_{\mathcal{F},g} \in \arg\min_{x' \in \mathcal{X}_g} \max_{r \in \mathcal{R}_\mathcal{F}} \mathcal{L}_g(r, x'). \tag{2}$$

Some observations are in order. First, whether the optimization problem in Eq. (2) can be solved efficiently depends on the specific application $g$ and feedback $\mathcal{F}$ in question. Next, we choose to be robust (minimax) because our problem setting is not Bayesian (Ramachandran & Amir, 2007), i.e., we do not have a distribution over the set of rewards, but we only know that $r^\star \in \mathcal{R}_\mathcal{F}$. Finally, note that many IL algorithms (e.g., Abbeel & Ng (2004); Syed & Schapire (2007); Ho et al. (2016)) can be seen as adopting our robust approach (see Appendix A).

**Quantifying the error.** Eq. (2) represents a principled way to solve ReL problems that, unlike the approach commonly adopted in the literature (see Eq. 1), provides worst-case guarantees. However, we can *not* solve every ReL problem $(\mathcal{F}, g)$ by merely outputting $x_{\mathcal{F},g}$, because although $x_{\mathcal{F},g}$ is the choice with the smallest worst-case loss, the loss associated with $x_{\mathcal{F},g}$ might still be too large in the worst case. In other words, there are ReL problems $(\mathcal{F}, g)$ that cannot be solved even robustly, because we cannot guarantee that, in the worst case, with the information available, the true loss falls below some pre-specified threshold. Intuitively, this happens when the application $g$ requires significant knowledge about $r^\star$, which is not sufficiently provided by the feedback $\mathcal{F}$. In such cases, we must collect additional feedback if available; otherwise, we must tolerate weaker guarantees than those worst-case.

For these reasons, it is important to *quantify* the loss suffered in the worst case by $x_{\mathcal{F},g}$. Thanks to our new framework, we can compute it as:

$$\mathcal{I}_{\mathcal{F},g} := \max_{r \in \mathcal{R}_\mathcal{F}} \mathcal{L}_g(r, x_{\mathcal{F},g}) = \min_{x' \in \mathcal{X}_g} \max_{r \in \mathcal{R}_\mathcal{F}} \mathcal{L}_g(r, x'). \tag{3}$$

Since $\mathcal{I}_{\mathcal{F},g}$ measures how *un*informative is $\mathcal{F}$ for $g$, we call it the *uninformativeness* of $\mathcal{F}$ for $g$.

**A special case.** We conclude this section with some observations on the special case where $\mathcal{X}_g = \mathfrak{R}$, i.e., when we aim to output a reward. Here, the robust choice $x_{\mathcal{F},g}$ in Eq. (2) can be interpreted as the *Chebyshev center* (Alimov & Tsar'kov, 2019) of the feasible set $\mathcal{R}_\mathcal{F}$ in the premetric space $(\mathfrak{R}, \mathcal{L}_g)$. Building on the properties of the Chebyshev center, it is possible to derive interesting results. The main of these is that the robust reward $x_{\mathcal{F},g} \in \mathfrak{R}$ *does not necessarily belong to the feasible set* $\mathcal{R}_\mathcal{F}$, which is rather counterintuitive, especially because *the entire ReL literature has focused on recovering a reward function from the feasible set* $\mathcal{R}_\mathcal{F}$ (moredetails in Appendix C).

## 5 ROB-ReL: A ROBUST ALGORITHM FOR ReL

The goal of this section is to introduce an algorithm for solving ReL problems using the robust approach presented in Section 4. To this aim, we make two important observations.

- Solving ReL problems using the robust approach is *not merely an optimization problem*. In fact, $\mathcal{F}, g$ usually have to be estimated from finite data (see Remark 3.1).

- *No single and simple algorithm* can solve robustly all ReL problems. Indeed, even with infinite data, depending on $\mathcal{F}$ and $g$, Eq. (2) exhibits different properties from the optimization viewpoint.

For these reasons, we now focus on a specific *subset* of ReL problems in the *finite-sample* regime and we present `Rob-ReL`, a provably efficient algorithm for solving this subclass of ReL problems.

## 5.1 THE FAMILY OF REL PROBLEMS SOLVABLE BY ROB-REL

We consider an *interesting* and *explanatory* family of ReL problems $(\mathcal{F}, g)$ that have received limited attention in the literature. Specifically, we let $g$ be the application of "assessing a policy preference" (see Table 2) between policies $\pi^1, \pi^2$ in a target MDP without reward $\mathcal{M} = (\mathcal{S}, \mathcal{A}, H, s_0, p)$, i.e., we aim to output a scalar $x$ as close as possible to the difference in their expected returns:

$$\mathcal{X}_g = \mathbb{R}, \qquad \mathcal{L}_g(r, x) = \left| x - (J^{\pi^1}(r; p) - J^{\pi^2}(r; p)) \right|.$$

In addition, we require that the set of feedback $\mathcal{F} = \mathcal{F}_D \cup \mathcal{F}_{TC} \cup \mathcal{F}_{PC}$ contains only *demonstrations* $\mathcal{F}_D$, *trajectory comparisons* $\mathcal{F}_{TC}$ and *policy comparisons* $\mathcal{F}_{PC}$ feedback (see Table 1), of the following kind.

We allow for $m_D \geqslant 0$ demonstrations feedback $\mathcal{F}_D = \{f_{D,i}\}_{i=1}^{m_D}$, where each $f_{D,i}$ is a "$t_i$-suboptimal expert" feedback (Table 1) in some MDP without reward[2] $\mathcal{M}_{D,i} = (\mathcal{S}, \mathcal{A}, H, s_{0,D,i}, p_{D,i})$, with $t_i \in [0, H]$, corresponding to the feasible set:

$$\mathcal{R}_{f_{D,i}} = \{r \in \mathfrak{R} : J^{\pi_{D,i}}(r; p_{D,i}) \geqslant J^*(r; p_{D,i}) - t_i\}, \qquad \forall i \in [\![m_D]\!]. \tag{4}$$

Moreover, we allow for $m_{TC} \geqslant 0$ trajectory comparison feedback $\mathcal{F}_{TC} = \{f_{TC,i}\}_{i=1}^{m_{TC}}$, where each $f_{TC,i}$ is a "hard preference" feedback (Table 1) in some MDP without reward $\mathcal{M}_{TC,i} = (\mathcal{S}, \mathcal{A}, H, s_{0,TC,i}, p_{TC,i})$, with feasible set:

$$\mathcal{R}_{f_{TC,i}} = \{r \in \mathfrak{R} : G(\omega_{TC,i}^1; r) \leqslant G(\omega_{TC,i}^2; r)\}, \qquad \forall i \in [\![m_{TC}]\!]. \tag{5}$$

Finally, we allow for $m_{PC} \geqslant 0$ policy comparison feedback $\mathcal{F}_{PC} = \{f_{PC,i}\}_{i=1}^{m_{PC}}$, where each $f_{PC,i}$ is a "hard preference" feedback (Table 1) in some MDP without reward $\mathcal{M}_{PC,i} = (\mathcal{S}, \mathcal{A}, H, s_{0,PC,i}, p_{PC,i})$, with feasible set:

$$\mathcal{R}_{f_{PC,i}} = \{r \in \mathfrak{R} : J^{\pi_{PC,i}^1}(r; p_{PC,i}) \leqslant J^{\pi_{PC,i}^2}(r; p_{PC,i})\}, \qquad \forall i \in [\![m_{PC}]\!]. \tag{6}$$

**Finite data.** To keep things realistic, we assume that the policies $\pi^1, \pi^2, \pi_{D,i}, \pi_{PC,i}^1, \pi_{PC,i}^2$ and the transition models $p, p_{D,i}, p_{TC,i}, p_{PC,i}$ are *not* known and must instead be estimated from data. We adopt a mixed offline-online setting that is common in the literature (e.g., see GAIL Ho & Ermon (2016)). To estimate the policies $\pi^1, \pi^2, \pi_{D,i}, \pi_{PC,i}^1, \pi_{PC,i}^2$, we assume access to batch datasets of trajectories $\mathcal{D}^1, \mathcal{D}^2, \mathcal{D}_{D,i}, \mathcal{D}_{PC,i}^1, \mathcal{D}_{PC,i}^2$ obtained by executing the policies in the corresponding environments $\mathcal{M}, \mathcal{M}_{D,i}, \mathcal{M}_{PC,i}$ for $n^1, n^2, n_{D,i}, n_{PC,i}^1, n_{PC,i}^2$ trajectories, respectively. To estimate the transition models $p, p_{D,i}, p_{TC,i}, p_{PC,i}$, we assume access to a forward sampling model[3] for each MDP without reward $\mathcal{M}, \mathcal{M}_{D,i}, \mathcal{M}_{TC,i}, \mathcal{M}_{PC,i}$, from which we can collect $N, N_{D,i}, N_{TC,i}, N_{PC,i}$ trajectories, respectively.

## 5.2 ROB-REL

We now present `Rob-ReL` (**Rob**ustness for **Re**ward **L**earning, Algorithm 1), a ReL algorithm for solving this family of ReL problems using the robust approach from Section 4. Specifically, in this setting, even with *infinite* data available, the robust approach (Eq. 2) requires solving the following optimization problem, where the constraints define the feasible set $\mathcal{R}_{\mathcal{F}}$:

$$x_{\mathcal{F},g} \in \underset{x' \in \mathbb{R}}{\arg\min} \max_{r \in \mathfrak{R}} \left| x' - \langle d^{\pi^1} - d^{\pi^2}, r \rangle \right| \tag{7}$$

$$\text{s.t.:} \ \max_\pi J^\pi(r; p_{D,i}) - \langle d^{\pi_{D,i}}, r \rangle \leqslant t_i \qquad \forall i \in [\![m_D]\!],$$

$$\langle d^{\omega_{TC,i}^1} - d^{\omega_{TC,i}^2}, r \rangle \leqslant 0 \qquad \forall i \in [\![m_{TC}]\!],$$

---

[2]All the considered environments share the same $\mathcal{S}, \mathcal{A}, H$ because we work with rewards on this domain.

[3]A *forward model* (Dann & Brunskill, 2015; Kakade, 2003) of an MDP $\mathcal{M}'$ permits to collect trajectories from $\mathcal{M}'$ by exploring at will.

**Algorithm 1:** `Rob-ReL`

**Input** : iterations $K$, feedback data
// Estimation:
1 Estimate $\widehat{d}^{\pi^1}, \widehat{d}^{\pi^2}, \widehat{d}^{\pi_{\mathrm{D},i}}, \widehat{d}^{\pi^1_{\mathrm{PC},i}}, \widehat{d}^{\pi^2_{\mathrm{PC},i}}$ via Eq. (8)
2 Estimate $\widehat{p}_{\mathrm{D},i}$ via RF-Express (Menard et al., 2021)
// Optimization:
3 $\widehat{m}_K \leftarrow$ PDSM-MIN$(\widehat{L}, K)$
4 $\widehat{M}_K \leftarrow$ PDSM-MAX$(\widehat{L}, K)$
// Targets:
5 $\widehat{x}_K \leftarrow (\widehat{M}_K + \widehat{m}_K)/2$
6 $\widehat{\mathcal{I}}_K \leftarrow (\widehat{M}_K - \widehat{m}_K)/2$
7 **Return** $\widehat{x}_K, \widehat{\mathcal{I}}_K$

**Algorithm 2:** PDSM-MIN

**Input** : objective $\widehat{L}$, iterations $K$
1 $\lambda_0 \leftarrow 0$
2 $r_0 \leftarrow 0$
3 $\mathfrak{D}_+ \leftarrow \{\lambda \geqslant 0 : \|\lambda\|_2 \leqslant s\}$
4 **for** $k = 0, 1, \ldots, K$ **do**
5 $\quad r_{k+1} \leftarrow \Pi_{\mathfrak{R}}\big(r_k - \alpha\partial_r\widehat{L}(r_k, \lambda_k)\big)$
6 $\quad \lambda_{k+1} \leftarrow \Pi_{\mathfrak{D}_+}\big(\lambda_k + \alpha\partial_\lambda\widehat{L}(r_k, \lambda_k)\big)$
7 **end**
8 $\widehat{r}_K \leftarrow \frac{1}{K}\sum_{k=0}^K r_k$
9 $\widehat{m}_K \leftarrow \langle \widehat{d}^{\pi^1} - \widehat{d}^{\pi^2}, \widehat{r}_K \rangle$
10 **Return** $\widehat{m}_K$

$$\langle d^{\pi^1_{\mathrm{PC},i}} - d^{\pi^2_{\mathrm{PC},i}}, r \rangle \leqslant 0 \qquad\qquad \forall i \in [\![m_{\mathrm{PC}}]\!].$$

However, with *finite* data, the quantities highlighted in blue are not known. Therefore, `Rob-ReL` instead solves the optimization problem obtained by replacing these quantities with their estimates. The next two paragraphs describe these estimates are computed by `Rob-ReL` and the optimization method it employs.

**Estimation.** Given a dataset $\mathcal{D} = \{(s_1^j, a_1^j, \ldots, s_H^j, a_H^j, s_{H+1}^j)\}_{j \in [\![n]\!]}$ of $n$ trajectories collected by some policy $\pi$, we can estimate the visit distribution of $\pi$ at all $(s, a, h) \in \mathcal{S} \times \mathcal{A} \times [\![H]\!]$ as:

$$\widehat{d}_h^\pi(s, a) = \frac{1}{n}\sum_{j \in [\![n]\!]} \mathbb{1}\{s_h^j = s, a_h^j = a\}, \tag{8}$$

Namely, with its empirical estimate on $\mathcal{D}$. In this way, `Rob-ReL` estimates $d^{\pi^1}, d^{\pi^2}, d^{\pi_{\mathrm{D},i}}, d^{\pi^1_{\mathrm{PC},i}}, d^{\pi^2_{\mathrm{PC},i}}$ from the corresponding datasets $\mathcal{D}^1, \mathcal{D}^2, \mathcal{D}_{\mathrm{D},i}, \mathcal{D}^1_{\mathrm{PC},i}, \mathcal{D}^2_{\mathrm{PC},i}$ (Line 1). Next, to estimate $p_{\mathrm{D},i}$, `Rob-ReL` executes RF-Express (Menard et al. (2021), Line 2), a minimax-optimal *reward-free* exploration algorithm (Jin et al., 2020a). In short, RF-Express collects $N_{\mathrm{D},i}$ trajectories from each $\mathcal{M}_{\mathrm{D},i}$, and then uses the resulting data to estimate $p_{\mathrm{D},i}$. Note that, since $p$, $p_{\mathrm{TC},i}$, and $p_{\mathrm{PC},i}$ do not appear in Eq. (7), then `Rob-ReL` does not need to estimate them.

**Optimization.** Let $\widehat{\mathcal{R}}_\mathcal{F}$ and $\widehat{\mathcal{L}}_g$ be the empirical counterparts of $\mathcal{R}_\mathcal{F}$ and $\mathcal{L}_g$ obtained by replacing the quantities in blue in Eq. (7) with the estimates described above. Then, `Rob-ReL` addresses the optimization problem:

$$\widehat{x}_{\mathcal{F},g} \in \arg\min_{x' \in \mathbb{R}} \max_{r \in \widehat{\mathcal{R}}_\mathcal{F}} \widehat{\mathcal{L}}_g(r, x'). \tag{9}$$

This is a minimax problem with non-trivial constraints. Nevertheless, we can simplify it. Define $\widehat{M}, \widehat{m}$ as the largest and smallest values of $\langle \widehat{d}^{\pi^1} - \widehat{d}^{\pi^2}, r \rangle$ over $\widehat{\mathcal{R}}_\mathcal{F}$:

$$\widehat{M} := \max_{r \in \widehat{\mathcal{R}}_\mathcal{F}} \langle \widehat{d}^{\pi^1} - \widehat{d}^{\pi^2}, r \rangle, \qquad \widehat{m} := \min_{r \in \widehat{\mathcal{R}}_\mathcal{F}} \langle \widehat{d}^{\pi^1} - \widehat{d}^{\pi^2}, r \rangle. \tag{10}$$

Then, we can rewrite Eq. (9) in a more convenient form:

**Proposition 5.1.** *It holds that:* $\widehat{x}_{\mathcal{F},g} = (\widehat{M} + \widehat{m})/2$.

As a result, instead of directly solving Eq. (9), `Rob-ReL` computes $\widehat{x}_{\mathcal{F},g}$ by first computing $\widehat{M}$ and $\widehat{m}$ via Eq. (10) (see Lines 3-4), and then combining the results through Proposition 5.1 (see Line 5). Observe that the optimization problems in Eq. (10) are *convex*, since both the objective functions and constraints are linear or convex (the pointwise maximum of linear functions is convex (Boyd & Vandenberghe, 2004)). To solve them, `Rob-ReL` finds saddle points of the Lagrangian function using the *primal-dual subgradient method* (PDSM, Nedić & Ozdaglar (2009)), which alternates subgradient updates for the primal and dual variables. Specifically, for any $r \in \mathfrak{R}$ and

$\lambda := (\lambda_D^\intercal, \lambda_{TC}^\intercal, \lambda_{PC}^\intercal)^\intercal \in \mathbb{R}^{m_D+m_{TC}+m_{PC}}$, the Lagrangian $\widehat{L}$ of both problems in Eq. (10) is:

$$\widehat{L}(r,\lambda) = \langle \widehat{d}^{\pi^1} - \widehat{d}^{\pi^2}, r \rangle + \sum_{i\in[\![m_D]\!]} \lambda_D^i \big( \max_\pi J^\pi(r; \widehat{p}_{D,i}) - \langle \widehat{d}^{\pi_{D,i}}, r \rangle - t_i \big) \tag{11}$$

$$+ \sum_{i\in[\![m_{TC}]\!]} \lambda_{TC}^i \langle d^{\omega_{TC,i}^1} - d^{\omega_{TC,i}^2}, r \rangle + \sum_{i\in[\![m_{PC}]\!]} \lambda_{PC}^i \langle \widehat{d}^{\pi_{PC,i}^1} - \widehat{d}^{\pi_{PC,i}^2}, r \rangle.$$

Then, the subroutines `PDSM-MIN` (Algorithm 2) and `PDSM-MAX` (Algorithm 3, Appendix E), invoked by **Rob-ReL** at Lines 3-4, aim to compute $\max_{\lambda \geqslant 0} \min_{r \in \mathfrak{R}} \widehat{L}(r, \lambda)$ and $\min_{\lambda \leqslant 0} \max_{r \in \mathfrak{R}} \widehat{L}(r, \lambda)$ by alternating between one subgradient step for $r$ and one for $\lambda$. Here, $\alpha > 0$ is the step size, $s$ is a hyperparameter (see Theorem 5.3 for principled choices of $\alpha$ and $s$). The expressions $\partial_r \widehat{L}(r_k, \lambda_k)$ and $\partial_\lambda \widehat{L}(r_k, \lambda_k)$ denote subgradients of $\widehat{L}$ w.r.t. $r$ and $\lambda$, evaluated at $(r_k, \lambda_k)$ (see Appendix E.2 for their formulas). To compute $\max_\pi J^\pi(r; \widehat{p}_{D,i})$, both subroutines make use of the backward induction algorithm (Puterman (1994), see Appendix E.2).

Finally, recall from Section 4 that we are also interested in quantifying the worst-case loss $\mathcal{I}_{\mathcal{F},g}$ that might be incurred when solving problem $(\mathcal{F}, g)$. This is computed as $\widehat{\mathcal{I}}_K$ by **Rob-ReL** at Line 6 based on the following result:

**Proposition 5.2.** *It holds that:* $\widehat{\mathcal{I}}_{\mathcal{F},g} := \max_{r \in \widehat{\mathcal{R}}_{\mathcal{F}}} \widehat{\mathcal{L}}_g(r, \widehat{x}_{\mathcal{F},g}) = (\widehat{M} - \widehat{m})/2.$

### 5.3 THEORETICAL ANALYSIS

We now show that **Rob-ReL** is both computationally and sample efficient. To this aim, we make the assumption that the feasible set $\mathcal{R}_{\mathcal{F}}$ contains a *strictly* feasible reward $\overline{r}$, which is common in both the optimization (Nedić & Ozdaglar, 2009) and the RL (Ding et al., 2020) literature:

**Assumption 5.1** (Slater's condition). *There exist $\xi > 0$ and $\overline{r} \in \mathfrak{R}$ such that:*

$$\begin{cases} \max_\pi J^\pi(\overline{r}; p_{D,i}) - \langle d^{\pi_{D,i}}, \overline{r} \rangle - t_i \leqslant -\xi & \forall i \in [\![m_D]\!] \\ \langle d^{\omega_{TC,i}^1} - d^{\omega_{TC,i}^2}, \overline{r} \rangle \leqslant -\xi & \forall i \in [\![m_{TC}]\!] \\ \langle d^{\pi_{PC,i}^1} - d^{\pi_{PC,i}^2}, \overline{r} \rangle \leqslant -\xi & \forall i \in [\![m_{PC}]\!] \end{cases}.$$

Then, we can prove the following result for **Rob-ReL**:

**Theorem 5.3.** *Let $(\mathcal{F}, g)$ be a ReL problem as described in Section 5.1 for which Assumption 5.1 holds. Let $\epsilon \in (0, 2H]$ and $\delta \in (0, 1)$. If we set $s = 4H/\xi + \sqrt{(4H/\xi)^2 + SAH/4}$ and $\alpha = \epsilon/(16H(1 + s\sqrt{m_D + m_{TC} + m_{PC}})^2)$, then, with probability $1 - \delta$, **Rob-ReL** satisfies:*

$$\mathcal{L}_g(r^\star, \widehat{x}_K) \leqslant \mathcal{I}_{\mathcal{F},g} + \epsilon \quad and \quad |\mathcal{I}_{\mathcal{F},g} - \widehat{\mathcal{I}}_K| \leqslant \epsilon,$$

*with a number of samples:*

$$n^1, n^2, n_{D,i}, n_{PC,i}^1, n_{PC,i}^2 \leqslant \widetilde{\mathcal{O}}\Big( \frac{SAH^5}{\epsilon^2 \xi^2} \log \frac{1}{\delta} \Big),$$

$$N_{D,i} \leqslant \widetilde{\mathcal{O}}\Big( \frac{SAH^5}{\epsilon^2 \xi^2} \big( S + \log \frac{1}{\delta} \big) \Big), \quad N, N_{TC,i}, N_{PC,i} = 0,$$

*and a number of iterations:*

$$K \leqslant \mathcal{O}\Bigg( \frac{H^{5/2}}{\xi \epsilon^2} \Big( \sqrt{SA} + \frac{\sqrt{H}}{\xi} \Big) \Big( 1 + \sqrt{H(m_D + m_{TC} + m_{PC})\Big( \frac{H}{\xi^2} + SA \Big)} \Big)^2 \Bigg).$$

Simply put, Theorem 5.3 tells us that **Rob-ReL** enjoys sample and iteration complexities that are polynomial in the quantities of interest $S, A, H, \frac{1}{\epsilon}, \log\frac{1}{\delta}, \frac{1}{\xi}, m_D, m_{TC}, m_{PC}$. In Appendix E.6, we show how to extend **Rob-ReL** to other forms of applications and feedback, while in Appendix E.5 we discuss possible extension to non-tabular environments. Finally, we observe that **Rob-ReL** can also be used for estimating the worst-case loss $\max_{r \in \mathcal{R}_{\mathcal{F}}} \mathcal{L}_g(r, x)$ incurred by deploying an arbitrary object $x \in \mathcal{X}_g$ (see Appendix E.4.1 for details).

*Proof Sketch.* Define $M := \max_{r \in \mathcal{R}_{\mathcal{F}}} \langle d^{\pi^1} - d^{\pi^2}, r \rangle$ and $m := \min_{r \in \mathcal{R}_{\mathcal{F}}} \langle d^{\pi^1} - d^{\pi^2}, r \rangle$. Then, after having shown that $x_{\mathcal{F},g} = (M + m)/2$ and $\mathcal{I}_{\mathcal{F},g} = (M - m)/2$, the result can be proved by upper

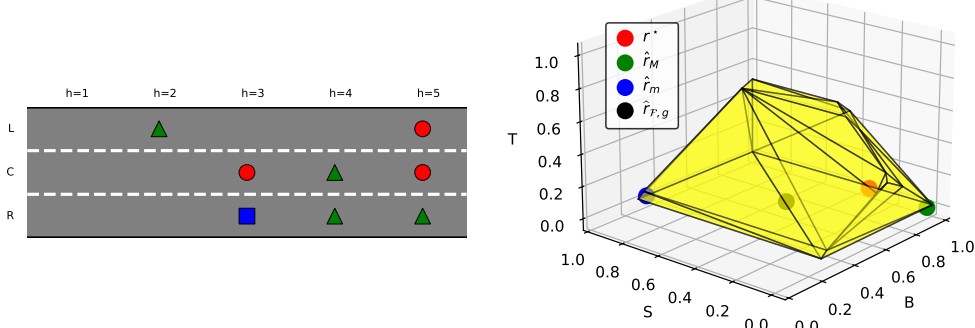

Figure 1: (Left) The target environment considered in the experiment. (Right) The feasible set with $r^\star$ and $\widehat{r}, \widehat{r}_M, \widehat{r}_m$. Axis $B, S, T$ refer to the reward values of B, S and T.

bounding the *estimation error* $|M - \widehat{M}| + |m - \widehat{m}|$ and the *iteration error* $|\widehat{M} - \widehat{M}_K| + |\widehat{m} - \widehat{m}_K|$. We upper bound the estimation error in Lemma E.2, by first bounding the error in estimating the visitation distribution of the policies (using Hoeffding's inequality) and the transition models (using the results in Menard et al. (2021)), and then showing that, under Assumption 5.1, all $M, \widehat{M}, m, \widehat{m}$ are saddle points with bounded optimal Lagrange multipliers (using Lemma 3 of Nedić & Ozdaglar (2009)). Regarding the iteration error (see Lemma E.6), we exploit the theoretical guarantees of the PDSM (Proposition 2 of Nedić & Ozdaglar (2009)). □

## 6 NUMERICAL SIMULATIONS

In this section, we present an illustrative experiment aimed at $(i)$ exemplifying the class of problems solvable by `Rob-ReL` and $(ii)$ providing a *graphical* intuition of the robust approach. For these reasons, we focus on a low-dimensional problem. We refer the reader to Appendix F.2 for a more extensive empirical analysis of `Rob-ReL`.[4]

**Environment.** We use the following state-action space $(\mathcal{S}, \mathcal{A}, H)$ to model the road depicted on the left of Figure 1. There are 12 states $s \in \mathcal{S}$ obtained by combining three lanes (left (L), center (C), right (R)) with four items (a ball (B), a square (S), a triangle (T), or nothing (N)). The action space contains three actions $\mathcal{A} = \{a_L, a_C, a_R\}$, that aim to bring an agent to, respectively, the left, keep the current lane, or go to the right, respectively. The horizon is $H = 5$.

**ReL problem.** We consider the ReL problem $(\mathcal{F}, g)$ in which an agent, `Alice` (e.g., a human), has a preference over the items B, S and T, that can be modeled through a three-dimensional reward $r^\star = [r_B^\star, r_S^\star, r_T^\star]$, where each component represents the value that `Alice` associates with visiting states containing items B, S and T ($r^\star$ is zero for N). $r^\star$ is unknown to our learner, and for the experiment we set $r^\star = [0.7, 0.1, 0.2]$. The application $g$ consists of assessing how much `Alice` prefers the policy $\pi^1$, that always takes action $a_R$, w.r.t. policy $\pi^2$, that always selects $a_L$, in an environment constructed using $(\mathcal{S}, \mathcal{A}, H)$ and some dynamics $s_0, p$ (see Appendix F). This value is $\Delta J(r^\star) := J^{\pi^1}(r^\star; p) - J^{\pi^2}(r^\star; p) = 0.39$, i.e., `Alice`'s preference for $\pi^1$ over $\pi^2$ has an "intensity" of 0.39. To simulate some feedback $\mathcal{F}$ from `Alice`, we randomly generated some demonstrations, policy comparisons and trajectory comparisons feedback consistent with $r^\star$ ($1 + 2 + 3 = 6$ feedback in total). The resulting feasible set $\mathcal{R}_\mathcal{F}$, along with $r^\star$, is plotted in Figure 1 (right).

**Simulation and results.** Our goal is to output a scalar $x \in \mathbb{R}$ that is as close as possible to $\Delta J(r^\star) = 0.39$ without knowing $r^\star$, but only that $r^\star$ belongs to the yellow set $\mathcal{R}_\mathcal{F}$ in Figure 1. We have executed `Rob-ReL` for $K = 1200$ iterations with a step size $\alpha = 0.01$ using for simplicity the exact values of policies and transition models, obtaining $\widehat{m}_K = -0.62, \widehat{M}_K = 1.02$, corresponding to the rewards in $\mathcal{R}_\mathcal{F}$ that provide the smallest and largest values of $\Delta J$ (see the blue $\widehat{r}_m$ and green $\widehat{r}_M$ rewards in Figure 1). The output of `Rob-ReL` is, therefore, $\widehat{x}_{\mathcal{F},g} = 0.2$ (corresponding to

---
[4]The code for running our simulations can be found at `https://github.com/filippolazzati/Rob-ReL`.

the black reward $\widehat{r}_{\mathcal{F},g}$ in Figure 1) and $\widehat{\mathcal{I}}_{\mathcal{F},g} = 0.82$. Thus, we know that the distance between $\widehat{x}_{\mathcal{F},g} = 0.2$ and $\Delta J(r^\star) = 0.39$ is at most $\widehat{\mathcal{I}}_{\mathcal{F},g} = 0.82$ in the worst case, and that it can be reduced by collecting more feedback. Moreover, looking at these numbers, we realize that, if our robust approach had not been adopted, i.e., if an arbitrary reward $r$ in $\mathcal{R}_{\mathcal{F}}$ had been used for prediction, then *the worst-case error might have been doubled* to approximately $1.64$ (e.g., if $r = \widehat{r}_M$ and $r^\star = \widehat{r}_m$).

## 7  CONCLUSION

In this paper, we presented a unifying and quantitative framework for studying ReL problems. We then introduced a principled approach for solving ReL problems in general and we described `Rob-ReL`, an algorithm tailored to a specific subset of ReL problems that uses our approach.

**Limitations and future work.**  The main limitation of this work is that the proposed algorithm, `Rob-ReL`, does not address all ReL problems. Therefore, we believe that future research should focus on developing additional algorithms that adopt the robust approach to tackle settings not covered by `Rob-ReL`. In this context, we also note that there may be ReL problems for which our robust approach leads to an intractable optimization problem or a complex estimation problem. Therefore, it will be important to develop meaningful approximations in such cases.

### ETHICS STATEMENT

This paper presents work whose goal is to advance the field of Machine Learning. There are many potential societal consequences of our work, none of which we feel must be specifically highlighted here.

### REPRODUCIBILITY STATEMENT

The proof of Theorem 5.3 is outlined in Section 5.3 and provided in full in Appendix E.3. The appendix also includes several additional theoretical insights, each accompanied by its own proof. For the numerical simulations, further details are given in Appendix F, and we include with the submission the code necessary to replicate the experiments.

### LLM USAGE

We relied on LLMs during paper writing solely to correct grammar and enhance clarity.

### ACKNOWLEDGEMENTS

This publication was funded with the contribution of Ministero dell'Università e della ricerca pursuant to D.D. n. 7206 of 17 April 2025 - BANDO FIS 2. Project FIS-2023-02598 (Starting Grant), title: "Unified Learning from Diverse Human Feedback" (HUmLrn). CUP: D53C25000710001.

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

## A  ADDITIONAL RELATED WORK

In this appendix, we provide a comprehensive presentation of the main related work of this paper. We group the related work into four categories: papers that "ignore" partial identifiability because it does not create issues in the applications that they consider, papers that explicitly address partial identifiability by looking for sufficient conditions that guarantee that it does not create issues, papers that aim to estimate the feasible set in a provably-efficient manner, and a miscellaneous of other papers. Next, in Appendix A.1, we mention that some IL algorithms can be seen as adopting our robust approach. Finally, in Appendix A.2, we provide a thorough comparison of our *quantitative* framework introduced in Section 3 with the *qualitative* framework provided by Skalse et al. (2023).

**Works that "ignore" partial identifiability.**   The partial identifiability in ReL is a topic that dates back to the seminal works of Russell (1998); Ng & Russell (2000). However, in many existing ReL works Ng & Russell (2000); Ratliff et al. (2006); Ziebart et al. (2008); Ziebart (2010); Boularias et al. (2011); Wulfmeier et al. (2016); Finn et al. (2016); Christiano et al. (2017), partial identifiability is not considered. The reason is that these works consider ReL problems $(\mathcal{F}, g)$ where partial identifiability does not create issues, because all the rewards in the feasible set $\mathcal{R}_{\mathcal{F}}$ behave as the unknown target reward $r^{\star}$ w.r.t. application $g$. Many of the mentioned papers are IRL works where the application $g$ is IL, and the set of feedback $\mathcal{F} = \{f\}$ contains a single demonstrations feedback $f$ able to provide enough information for IL. In particular, Ng & Russell (2000); Ratliff et al. (2006) introduce the heuristic of "margin maximization" to extract a reward from the feasible set that facilitates the IL task, and then perform planning on it to find the imitating policy to deploy. Ziebart et al. (2008); Ziebart (2010) introduce new ReL feedback consisting in demonstrations collected from a maximum (causal) entropy policy and present an algorithm that recovers an arbitrary reward from the corresponding feasible set (the authors let the inner optimization algorithm break the ties). Boularias et al. (2011); Wulfmeier et al. (2016); Finn et al. (2016) adopt the model of Ziebart et al. (2008) and extend its algorithm by, respectively, adopting a model-free approach, using neural networks for parameterizing the reward function to learn, and approximating the objective to speed-up the learning process under unknown dynamics in high-dimensional continuous systems. Finally, we mention Christiano et al. (2017) that, in the context of PbRL, adopts a maximum likelihood approach to recover an arbitrary reward from the corresponding feasible set (also Christiano et al. (2017) let the inner optimization algorithm break the ties) from finite data.

**Works on partial identifiability.**   When the aforementioned algorithms have been applied to other applications (i.e., by using the corresponding recovered reward function), then partial identifiability turned out to be a problem, as observed empirically for instance by Finn et al. (2016), who tried to transfer the recovered reward to a new environment. Consequently, many works addressing the identifiability problem have appeared Amin & Singh (2016); Fu et al. (2017); Cao et al. (2021); Kim et al. (2021); Viano et al. (2021); Rolland et al. (2022); Skalse et al. (2023); Schlaginhaufen & Kamgarpour (2023; 2024); Lang et al. (2024). In particular, Amin & Singh (2016) consider access to demonstrations feedback from multiple environments to reduce the size of the feasible set. Similarly, Cao et al. (2021) consider demonstrations feedback from multiple environments or using multiple discount factors to improve the identifiability of the target reward, while Kim et al. (2021) focuses on the properties of the considered MDPs. Fu et al. (2017) assume that the target reward is state-only and that the environment satisfies certain additional properties to guarantee that the feasible set contains only rewards that can be transferred to new environments in the same way as the true target reward. Viano et al. (2021) propose a robust approach for the reward transfer application that is rather different from ours. Specifically, the authors propose to deploy the policy that minimizes the loss w.r.t. the worst possible transition model in a certain rectangular uncertainty set Wiesemann et al. (2013) centered in the transition model of the target environment. Rolland et al. (2022) studies the identifiability problem using as feedback demonstrations from multiple experts, and focuses on the application of reward transfer. Schlaginhaufen & Kamgarpour (2023; 2024) provide a similar study but use, respectively, different kinds of feedback and a different method for studying the similarity of the environments. Skalse et al. (2023) present a general study of the identifiability problem in both IRL and PbRL, and we provide a detailed analysis of this work in Appendix A.2. Finally, Lang et al. (2024) study the identifiability problem in PbRL assuming that the feedback is based only on partial observations of the environment.

**Works that estimate the feasible set.** A recent line of research has focused on studying the sample complexity of estimating the feasible set Metelli et al. (2021); Lindner et al. (2022); Metelli et al. (2023); Zhao et al. (2024); Lazzati et al. (2024b;a); Poiani et al. (2024). All these papers consider the IRL problem settings where the feedback consists in demonstrations from an optimal or $\epsilon$-optimal Poiani et al. (2024) expert's policy. Specifically, Metelli et al. (2021) is the seminal work in this context. It assumes availability of a generative sampling model of the environment in the tabular setting, and provides an upper bound to the sample complexity for estimating the feasible set. This result is completed by Metelli et al. (2023) who present a lower bound in a similar setting. Lindner et al. (2022) and Zhao et al. (2024) focus on a forward sampling model in tabular MDPs, and provide upper bounds to the sample complexity. Lazzati et al. (2024a) extend these theoretical results to the Linear MDPs Jin et al. (2020b) setting. Instead, Lazzati et al. (2024a) and Zhao et al. (2024) consider the offline setting where only batch datasets of demonstrations are available, and provide results under some concentrability assumptions. Poiani et al. (2024) present both lower and upper bounds for estimating the feasible set in tabular settings assuming a suboptimal expert.

**Robust MDPs.** Robust MDPs (Wiesemann et al., 2013; Gadot et al., 2024) represent a problem setting in which the true reward is unknown, but it is known to belong to a given *uncertainty set* $\mathcal{R}$. There are some similarities and differences between robust MDPs and our robust ReL formulation. In particular, in both Reward-Robust MDPs and our Eq. (2), the goal is to be robust against the missing knowledge of the true reward, which is known to belong to a certain set of rewards, that is called feasible set in our ReL setting, while it is called uncertainty set in the context of Robust MDPs. However, there are three crucial differences. First, in robust MDPs, there is a single application, i.e., finding a good policy under the unknown reward, while in our ReL setting there can be a variety of different applications. Second, and most importantly, in robust MDPs the uncertainty set is given and known, while in ReL the feasible set must be estimated from finite data (e.g., demonstrations or trajectory/policy comparisons). Third, in the literature, the uncertainty set in Robust MDPs is almost always rectangular, while in ReL the shape of the feasible set can be different and more complex.

**Others.** Ramachandran & Amir (2007) adopt a Bayesian approach in the context of IRL. It assumes a prior on the set of rewards $\mathfrak{R}$, and assumes that the feedback provide a likelihood. The authors focus on two specific applications, i.e., IL and learning $r^\star$, and propose to minimize the *expected* loss. In some way, Ramachandran & Amir (2007) can be seen as deploying the object $x$ that minimizes the *average* loss w.r.t. some known distribution over $r^\star$ (instead of our *worst-case* loss), and Brown et al. (2020) that generalizes this approach using risk measures. We mention also Zhu et al. (2023), that in the context of PbRL make a "pessimistic" choice of policy given a "confidence" set of rewards. From a high-level perspective, their pessimistic approach is very close to our robust approach. However, their proposal aims to address estimation issues, while ours concerns identifiability issues. Jeon et al. (2020) introduce a framework for combining multiple and various ReL feedback, that are all modeled using a Boltzmann distribution constructed indirectly through the target reward. Cheng et al. (2024) adopt a robust approach to PbRL, but the considered robustness aims to address a noisy feedback, and not partial identifiability. Finally, we mention Huang et al. (2018) that adopt a robust approach for policy optimization using multiple demonstrations.

### A.1 A NOTE ON IL WORKS

In the IL literature Osa et al. (2018), there are some algorithms that can be interpreted as adopting the robust approach presented in Section 4. Specifically, such algorithms aim to solve the following optimization problem:

$$\pi \in \arg\min_{\pi' \in \Pi} \max_{r \in \mathcal{R}} \Big( J^{\overline{\pi}}(r; p) - J^{\pi'}(r; p) \Big), \tag{12}$$

where $\overline{\pi}$ is the expert's policy and $\mathcal{R}$ is some set of reward functions Ho et al. (2016), e.g., rewards linear Abbeel & Ng (2004) or convex Syed & Schapire (2007) in some known feature map $\phi : \mathcal{S} \times \mathcal{A} \to \mathbb{R}^d$. If we look at Eq. (12) through the lens of our framework, then $\pi$ can be seen as the *robust* choice of policy that minimizes the worst-case loss $\mathcal{L}_g = J^{\overline{\pi}}(r; p) - J^{\pi}(r; p)$ over the feasible set $\mathcal{R}_\mathcal{F} = \mathcal{R}$, and $\mathcal{X}_g = \Pi$.

### A.2   COMPARISON WITH SKALSE ET AL. (2023)

The work of Skalse et al. (2023) deserves its own section because it introduces a ReL framework that shares similarities with ours. Thus, it is important to remark the differences.

*Similarly* to our framework, also the framework of Skalse et al. (2023) provides a model of feedback $f$ that always corresponds to a feasible set $\mathcal{R}_f \subseteq \mathfrak{R}$. The *difference* lies in how they model the applications $g$. Specifically, Skalse et al. (2023) model an application $g$ as a function $g : \mathfrak{R} \to \mathcal{X}_g$, and, for every reward $r \in \mathfrak{R}$, they let $g(r)$ be the object that should be deployed in case reward $r$ were the target reward $r^\star$. Crucially, this model is *qualitative*, namely, it does not contemplate the possibility that another object $x \neq g(r^\star)$ can be deployed, and so the authors require that, for every pair of rewards $r, r' \in \mathcal{R}_f$ in the feasible set, it must hold that $g(r) = g(r')$ (since $r^\star \in \mathcal{R}_f$, this guarantees that no object other than $g(r^\star)$ can be deployed). Instead, in our *quantitative* framework, we allow for this to happen as long as the loss $\mathcal{L}_g(r^\star, x)$ of $x$ is sufficiently small.

In other words, the framework of Skalse et al. (2023) and ours give birth to different *sufficient conditions* on whether a set of feedback $\mathcal{F}$ can be used for carrying out an application $g$. In particular, the framework of Skalse et al. (2023) requires that:

$$\mathcal{R}_\mathcal{F} \subseteq g^{-1}\big(g(r^\star)\big), \tag{13}$$

i.e., that all the rewards in the feasible set $\mathcal{R}_\mathcal{F}$ prescribe the same object $x = g(r^\star)$, the object prescribed also by $r^\star$. Clearly, this is qualitative, and if we add a single reward $r'$ with $g(r') \neq g(r^\star)$ to $\mathcal{R}_\mathcal{F}$, then the framework of Skalse et al. (2023) concludes that we cannot use $\mathcal{F}$ for $g$ anymore. Our framework, instead, provides a *more general* quantitative sufficient condition. Specifically, for some threshold $\Delta \geqslant 0$, our framework requires that there exists at least an item $x \in \mathcal{X}_g$ such that:

$$\mathcal{L}_g(r, x) \leqslant \Delta \qquad \forall r \in \mathcal{R}_\mathcal{F}. \tag{14}$$

Intuitively, if we set $\Delta = 0$, then we recover Eq. (13), because we would be requiring that all the rewards $r$ in the feasible set have the same minima of $\mathcal{L}_g(r, \cdot)$, and so, that they all prescribe the same item. Instead, by enforcing Eq. (14), we are basically asking that it is possible to find at least one item (i.e., $x$ in Eq. 14), that suffers from a true loss upper bounded by $\Delta$, i.e., Eq. (14) guarantees that:

$$\mathcal{L}_g(r^\star, x) \leqslant \Delta,$$

since $r^\star \in \mathcal{R}_\mathcal{F}$.

A simple example can help in recognizing the advantages of our framework. Consider an MDP with a single state $s$, three actions $a_1, a_2, a_3$, and horizon $H = 1$. Assume that we have received some feedback $\mathcal{F}$ that results in the feasible set $\mathcal{R}_\mathcal{F} = \{r_1, r_2\}$ containing only two rewards, and that the application $g$ consists in outputting the optimal policy under $r^\star$. Then, Skalse et al. (2023) tell us that we can solve this problem *if* the optimal policies induced by $r_1$ and $r_2$ coincide (in other words, if it is irrelevant whether $r^\star$ is $r_1$ or $r_2$). However, what about the situation in which, e.g.:

$$r_1(s, a) = \begin{cases} 1 & \text{if } a = a_1 \\ 0 & \text{if } a = a_2 \\ 0 & \text{if } a = a_3 \end{cases}, \qquad r_2(s, a) = \begin{cases} 1 - \Delta & \text{if } a = a_1 \\ 1 & \text{if } a = a_2 \\ 0 & \text{if } a = a_3 \end{cases} \qquad ?$$

Clearly, $r_1$ makes $a_1$ optimal, while $r_2$ makes $a_2$ optimal. However, as long as $\Delta$ is small, $a_1$ is an almost-optimal policy also for $r_2$, thus, intuitively, it "will not be a problem for $g$ if we deploy $a_1$ even if $r^\star = r_2$". Note that our framework allows for this situation, by telling us that, as long as $\Delta$ is sufficiently small, then partial identifiability is not an issue for this problem, and we can solve it (by deploying the policy that plays $a_1$, or better, based on the robust approach presented in Section 4, by deploying the policy that plays a mixture of $a_1, a_2$ with equal probabilities). In addition, note that $\Delta$ corresponds to the uninformativeness $\mathcal{I}_{\mathcal{F},g}$, and, thus, can be computed.

To sum up, our framework has the advantage of allowing for quantitative considerations, while the framework of Skalse et al. (2023) cannot.

## B    MORE ON THE *Policy Comparison* FEEDBACK

In this appendix, we provide additional details on the new *policy comparison* feedback (see Table 1) introduced in Section 3.1.

In the IRL literature, we are given a dataset of demonstrations, i.e., state-action trajectories $\mathcal{D} = \{\omega^i\}_i \sim d^{\pi^E}$ collected by executing some (expert) policy $\pi^E$. This setting models the situation in which we observe an agent doing a task many times, and the assumption is that the agent behavior is guided by the reward $r^\star$.

In the PbRL literature, we are given two trajectories and a preference signal between them. This setting models the situation in which an agent expresses a preference between two trajectories, i.e., the agent observes two trajectories and says which one it prefers. In doing so, the choice is guided by the target reward $r^\star$.

The *policy comparison* feedback introduced in Section 3.1 concerns with the scenario in which we have an expert agent (i.e., the agent with $r^\star$ in mind) that expresses a preference (or, more generally, any statement) between two datasets of demonstrations collected by (but not necessarily) two other agents. Simply put, it can be seen as a mix of the demonstrations and the trajectory comparisons feedback types.

For example, assume that we observe two agents $A_1, A_2$ demonstrating the task of driving a car, and they provide two datasets of demonstrations $\mathcal{D}_1 = \{\omega_i^1\}_i \sim d^{\pi^1}, \mathcal{D}_2 = \{\omega_i^2\}_i \sim d^{\pi^2}$ where $\pi^1$ is the policy of $A_1$ and $\pi^2$ is the policy of $A_2$. Assume that we do not want to learn the reward function that guides the behavior of $A_1$ nor $A_2$, but we aim to learn the reward of a third agent $E$ (i.e., $r^\star$ is the reward of $E$). Thus, we can ask $E$ to provide us with a preference signal (or, more generally, any statement between the demonstrated olicies $\pi^1, \pi^2$) between the behavior of $A_1, A_2$. For instance, we can show to $E$ the video of how $A_1, A_2$ drive, and we can ask him who drives better. Then, we can use this feedback to infer $r^\star$ and carry out any downstream application $g$.

## C    WHEN THE APPLICATION IS TO DEPLOY A REWARD

In this appendix, we provide additional insights on the class of ReL problems in which the application $g$ requires the deployment of a reward function, i.e., $\mathcal{X}_g = \mathfrak{R}$. Before that, we need some additional notation.

**Additional notation.**    Let $\mathcal{X}$ be a set, and let $d : \mathcal{X} \times \mathcal{X} \to \mathbb{R}_+$ be a premetric in $\mathcal{X}$. If, in addition, $d$ satisfies $(i)$ $d(x, y) = 0$ if and only if $x = y$ (identity of indiscernibles), $(ii)$ $d(x, y) = d(y, x) \ \forall x, y \in \mathcal{X}$ (symmetry), $(iii)$ $d(x, y) \leqslant d(x, z) + d(z, y) \ \forall x, y, z \in \mathcal{X}$ (triangle inequality), then we say that $d$ is a *metric*. Let $d$ be a premetric in a set $\mathcal{X}$. The *Chebyshev center* Alimov & Tsar'kov (2019) of a set $\mathcal{Y} \subseteq \mathcal{X}$ is any of the points in $\arg\min_{x \in \mathcal{X}} \max_{y \in \mathcal{Y}} d(x, y)$. The *Chebyshev radius* of $\mathcal{Y}$ is defined as $\min_{x \in \mathcal{X}} \max_{y \in \mathcal{Y}} d(x, y)$, while the diameter of $\mathcal{Y}$ is $\max_{x, y \in \mathcal{Y}} d(x, y)$. Moreover, given a reward $r$ and an environment with dynamics $p$, we denote by $\Pi^*(r; p) := \arg\max_\pi J^\pi(r; p)$ the set of optimal policies under $r$ in $p$.

**Setting.**    There are situations in which we are interested in using some given feedback $\mathcal{F}$ for learning a reward function $\mathcal{X}_g = \mathfrak{R}$. In these cases, the loss $\mathcal{L}_g : \mathfrak{R} \times \mathfrak{R} \to \mathbb{R}_+$ can be seen as a premetric in the set of rewards $\mathfrak{R}$. These settings can happen for various reasons, like:

- The ultimate goal is to learn $r^\star$. Then, we can use as loss some "standard" metric in the set of rewards, like that induced by the 2-norm (see Ramachandran & Amir (2007)):

$$\mathcal{L}_2(r, r') := \|r - r'\|_2, \tag{15}$$

   or by the max norm:

$$\mathcal{L}_\infty(r, r') := \|r - r'\|_\infty := \max_{s, a, h} |r_h(s, a) - r'_h(s, a)|. \tag{16}$$

- The true application $g$ is unknown or revealed a posteriori, and we just know that it belongs to a given set of applications. For instance, in case we know that we want to use $r^\star$ to assess the return of some trajectory $\omega'$, but we do not know $\omega'$ yet, then solving all the

ReL problems corresponding to every possible trajectory may be inefficient (there is an exponential number of trajectories). Thus, we can simply learn a reward and then use it at a later time when we will observe $\omega'$. In such setting, we can use as loss:

$$\mathcal{L}_{\text{TR}}(r, r') := \max_{\omega \in \Omega} |G(\omega; r) - G(\omega; r')|, \qquad (17)$$

which quantifies the worst-case error among all possible trajectories.

- For some reason, we aim to pass through a reward function to solve the true application. For instance, in case the application is planning in some MDP with transition model $p$ (see Table 2), then we can add an intermediate step of passing through a reward function by using as loss:

$$\mathcal{L}_{\text{PL},p}(r, r') := J^*(r'; p) - \min_{\pi \in \Pi^*(r;p)} J^\pi(r'; p), \qquad (18)$$

which considers, among all the optimal policies of $r$, the one that maximizes the suboptimality under $r'$. Another similar example is the application of computing the greedy policy $\pi^{\text{gr}}(\cdot; r) \in \arg\max_{a \in \mathcal{A}} r(\cdot, a)$ in a stationary environment Zhu et al. (2023). Here, for some distribution $\rho \in \Delta^{\mathcal{S}}$, we can use:

$$\mathcal{L}_{\text{GR},\rho}(r, r') := \mathbb{E}_{s \sim \rho}\big[ \max_{a \in \mathcal{A}} r'(s, a) - r'(s, \pi^{\text{gr}}(s; r))\big]. \qquad (19)$$

Note that we introduced the notion of premetric because not all of the introduced dissimilarity functions are metrics:

**Proposition C.1.** *For any $p, \rho$, then $\mathcal{L}_{PL,p}, \mathcal{L}_{CO,p}, \mathcal{L}_{GR,\rho}$ are premetrics. Moreover, there are some $p, \rho$ such that:*

- *$\mathcal{L}_{PL,p}, \mathcal{L}_{CO,p}, \mathcal{L}_{GR,\rho}$ lack the identity of indiscernibles;*
- *$\mathcal{L}_{PL,p}, \mathcal{L}_{GR,\rho}$ are not simmetric;*
- *$\mathcal{L}_{PL,p}, \mathcal{L}_{GR,\rho}$ lack the triangle inequality.*

*Proof.* It is immediate that, for any transition model $p \in \Delta_{\mathcal{S}} \times \mathcal{A} \times [\![H]\!]^{\mathcal{S}}$ and distribution $\rho \in \Delta^{\mathcal{S}}$, the distances $\mathcal{L}_{\text{PL},p}, \mathcal{L}_{\text{CO},p}, \mathcal{L}_{\text{GR},\rho}$ are non-negative for any pair of rewards $r, r' \in \mathfrak{R}$, and also that they are all 0 when $r = r'$. Thus, we conclude that they are premetrics.

Now, consider the **identity of indiscernibles** property defined earlier. Given distance $\mathcal{L}_g$, it does not hold if there exist $r \neq r'$ s.t. $\mathcal{L}_g(r, r') = 0$. Concerning $\mathcal{L}_{\text{PL},p}$, we see that any pair of rewards $r, r'$ such that $\Pi^*(r; p) = \Pi^*(r'; p)$ satisfies $\mathcal{L}_{\text{PL},p}(r, r') = 0$. Thus, we can take $r'$ to be, e.g., a multiple of $r$ to get $\mathcal{L}_{\text{PL},p}(r, r') = 0$. Concerning $\mathcal{L}_{\text{CO},p}$, simply consider as $p$ a transition model for which there exists at least a $(s, a, h) \in \mathcal{S} \times \mathcal{A} \times [\![H]\!]$ s.t., for all $\pi \in \Delta_{\mathcal{S} \times [\![H]\!]}^{\mathcal{S}}$, $d_h^\pi(s, a) = 0$. Then, such triple does not contribute to the performance of any policy, and therefore any pair of rewards $r, r'$ that coincide everywhere except for $s, a, h$ satisfy $\mathcal{L}_{\text{CO},p}(r, r') = 0$. Concerning $\mathcal{L}_{\text{GR},\rho}$, we can take $r, r'$ for which there exists a state $s \in \mathcal{S}$ where $\max_{a \in \mathcal{A}} r(s, a) = \max_{a \in \mathcal{A}} r'(s, a)$, but the maximum is achieved by different actions $\arg\max_{a \in \mathcal{A}} r(s, a) \neq \arg\max_{a \in \mathcal{A}} r'(s, a)$. Clearly, $r \neq r'$, but $\mathcal{L}_{\text{GR},\rho} = 0$.

Consider now the **simmetry** property. For $\mathcal{L}_{\text{PL},p}$, take two rewards $r, r'$ such that $\Pi^*(r; p) = \{\pi^1\}, \Pi^*(r'; p) = \{\pi^1, \pi^2\}$ for some policies $\pi^1, \pi^2$. Then:

$$\mathcal{L}_{\text{PL},p}(r, r') := J^*(r'; p) - \min_{\pi \in \Pi^*(r;p)} J^\pi(r'; p) = J^{\pi^1}(r'; p) - J^{\pi^1}(r'; p) = 0,$$

$$\mathcal{L}_{\text{PL},p}(r', r) := J^*(r; p) - \min_{\pi \in \Pi^*(r';p)} J^\pi(r; p) = J^{\pi^1}(r; p) - J^{\pi^2}(r; p) \neq 0.$$

Thus, $\mathcal{L}_{\text{PL},p}(r, r') \neq \mathcal{L}_{\text{PL},p}(r', r)$, so simmetry does not hold. For $\mathcal{L}_{\text{CO},p}$ the simmetry property holds for any pair of rewards $r, r' \in \mathfrak{R}$:

$$\mathcal{L}_{\text{CO},p}(r, r') := \max_\pi |J^\pi(r'; p) - J^\pi(r; p)|$$
$$= \max_\pi |-(J^\pi(r; p) - J^\pi(r'; p))|$$
$$= \max_\pi |J^\pi(r; p) - J^\pi(r'; p)|$$
$$= \mathcal{L}_{\text{CO},p}(r', r).$$

Concerning $\mathcal{L}_{\mathrm{GR},\rho}$, the simmetry property does not hold. To see it, consider a problem with a single state $s$ and two actions $a_1, a_2$, and let $r, r'$ be two rewards such that:

$$r(s,a) = \begin{cases} 1 & \text{if } a = a_1, \\ 0 & \text{if } a = a_2, \end{cases} \qquad r'(s,a) = \begin{cases} 0 & \text{if } a = a_1, \\ 0.5 & \text{if } a = a_2, \end{cases}.$$

Then, we have that:

$$\mathcal{L}_{\mathrm{GR},\rho}(r,r') := \mathbb{E}_{s \sim \rho}\Big[ \max_{a \in \mathcal{A}} r'(s,a) - r'(s, \pi^{\mathrm{gr}}(s;r)) \Big] = r'(s,a_2) - r'(s,a_1) = 0.5,$$

$$\mathcal{L}_{\mathrm{GR},\rho}(r',r) := \mathbb{E}_{s \sim \rho}\Big[ \max_{a \in \mathcal{A}} r(s,a) - r(s, \pi^{\mathrm{gr}}(s;r')) \Big] = r(s,a_1) - r(s,a_2) = 1.$$

Finally, let us consider the **triangle inequality** property. First, we consider $\mathcal{L}_{\mathrm{PL},p}$. Let $r, r', r'' \in \mathfrak{R}$ be three rewards such that $\Pi^*(r;p) = \Pi^*(r'';p) = \{\pi_1\}, \Pi^*(r';p) = \{\pi^2\}$, and take $r'' = 0.5r$. Then:

$$\mathcal{L}_{\mathrm{PL},p}(r',r) = J^{\pi^1}(r;p) - J^{\pi^2}(r;p),$$

$$\mathcal{L}_{\mathrm{PL},p}(r',r'') = J^{\pi^1}(r'';p) - J^{\pi^2}(r'';p) = 0.5J^{\pi^1}(r;p) - 0.5J^{\pi^2}(r;p) = 0.5\mathcal{L}_{\mathrm{PL},p}(r',r),$$

$$\mathcal{L}_{\mathrm{PL},p}(r'',r) = J^{\pi^1}(r;p) - J^{\pi^1}(r;p) = 0.$$

Therefore, we have that:

$$\mathcal{L}_{\mathrm{PL},p}(r',r) > \mathcal{L}_{\mathrm{PL},p}(r',r'') + \mathcal{L}_{\mathrm{PL},p}(r'',r),$$

which proves that triangle inequality does not hold. Concerning $\mathcal{L}_{\mathrm{CO},p}$, triangle inequality holds for any $p$ and $r, r', r'' \in \mathfrak{R}$:

$$\mathcal{L}_{\mathrm{CO},p}(r',r) := \max_{\pi} \left| J^{\pi}(r;p) - J^{\pi}(r';p) \pm J^{\pi}(r'';p) \right|$$

$$\leqslant \max_{\pi} \left| J^{\pi}(r'';p) - J^{\pi}(r';p) \right| + \max_{\pi} \left| J^{\pi}(r;p) - J^{\pi}(r'';p) \right|$$

$$= \mathcal{L}_{\mathrm{CO},p}(r',r'') + \mathcal{L}_{\mathrm{CO},p}(r'',r),$$

where we have applied the triangle inequality of the absolute value and the fact that the maximum of a sum is smaller than the sum of maxima. As far as $\mathcal{L}_{\mathrm{GR},\rho}$ is concerned, we note that it lacks the triangle inequality property with a counterexample analogous to that for $\mathcal{L}_{\mathrm{PL},p}$. Consider a problem with a single state $s$ (or $\rho$ supported only on it) and two actions $a_1, a_2$, and consider the rewards $r, r', r'' \in \mathfrak{R}$ such that:

$$r(s,a) = \begin{cases} 1 & \text{if } a = a_1, \\ 0 & \text{if } a = a_2, \end{cases} \qquad r'(s,a) = \begin{cases} 0 & \text{if } a = a_1, \\ 1 & \text{if } a = a_2, \end{cases} \qquad r''(s,a) = \begin{cases} 0.5 & \text{if } a = a_1, \\ 0 & \text{if } a = a_2, \end{cases}.$$

Then, we have that:

$$\mathcal{L}_{\mathrm{GR},\rho}(r',r) = r(s,a_1) - r(s,a_1) = 1,$$

$$\mathcal{L}_{\mathrm{GR},\rho}(r',r'') = r''(s,a_1) - r''(s,a_1) = 0.5 = 0.5\mathcal{L}_{\mathrm{GR},\rho}(r',r),$$

$$\mathcal{L}_{\mathrm{GR},\rho}(r',r) = r(s,a_1) - r(s,a_1) = 0.$$

Thus:

$$\mathcal{L}_{\mathrm{GR},\rho}(r',r) = 1 > \mathcal{L}_{\mathrm{GR},\rho}(r',r'') + \mathcal{L}_{\mathrm{GR},\rho}(r'',r) = 0.5 + 0.$$

This concludes the proof.

$\square$

**Robust approach and Chebyshev center.** In these scenarios, the robust choice $x_{\mathcal{F},g}$ presented in Eq. (2) corresponds to the Chebyshev center of the feasible set $\mathcal{R}_{\mathcal{F}}$ in the premetric space $(\mathfrak{R}, \mathcal{L}_g)$. Since $x_{\mathcal{F},g} \in \mathfrak{R}$, then we denote it by $r_{\mathcal{F},g}$. Moreover, observe that the informativeness $\mathcal{I}_{\mathcal{F},g}$ (Eq. 3) can be interpreted as the Chebyshev radius of the feasible set $\mathcal{R}_{\mathcal{F}}$, and that, in the worst-case, the ReL procedure carried out by most algorithms in literature (see Section 4, Eq. 1) can be seen as the diameter of the feasible set, that we denote by $D_{\mathcal{F},g}$. Finally, given an arbitrary reward $r \in \mathfrak{R} = \mathcal{X}_g$, we can define its worst-case loss as:

$$\mathcal{C}_{\mathcal{F},g}(r) := \max_{r' \in \mathcal{R}_{\mathcal{F}}} \mathcal{L}_g(r,r'). \tag{20}$$

See Figure 2 for a simple graphical intuition of all these quantities.

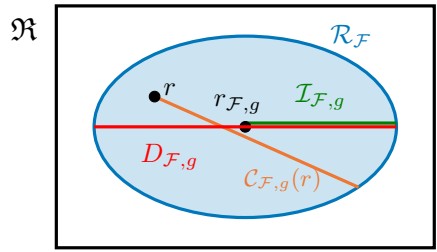

Figure 2: Illustration of the quantities of interest. $r$ is any reward.

In the following, we provide some interesting results on these quantities.

**Some results.** One of the most interesting results is that, depending on $\mathcal{F}, g$, the Chebyshev center might lie *outside* of the feasible set $\mathcal{R}_{\mathcal{F}}$. This is interesting because no ReL work in literature has looked for a reward function outside $\mathcal{R}_{\mathcal{F}}$ as far as we know. Formally, we show that this can happen for $\mathcal{L}_{\text{PL},p}$ and $\mathcal{L}_{\infty}$:

**Proposition C.2.** *If the loss is $\mathcal{L}_g = \mathcal{L}_{PL,p}$, then there exists a (convex) set $\mathcal{R}_{\mathcal{F}}$ for which $r_{\mathcal{F},g} \notin \mathcal{R}_{\mathcal{F}}$.*

*Proof.* Consider a simple MDP without reward with a single state $s$, three actions $a_1, a_2, a_3$, and horizon $H = 1$. Let $\pi^1, \pi^2, \pi^3$ be, respectively, the deterministic policies that play actions $a_1, a_2, a_3$. In this context, take $\mathcal{R}_{\mathcal{F}} = \{r, r'\}$, where:

$$r(s,a) = \begin{cases} 1 & \text{if } a = a_1, \\ 0 & \text{if } a = a_2, , \\ 0.5 & \text{if } a = a_3, \end{cases} \qquad r'(s,a) = \begin{cases} 0 & \text{if } a = a_1, \\ 1 & \text{if } a = a_2, . \\ 0.5 & \text{if } a = a_3, \end{cases}$$

Since $\Pi^*(r;p) = \{\pi^1\}$ and $\Pi^*(r';p) = \{\pi^2\}$, then it is immediate that:

$$\mathcal{L}_{\text{PL},p}(r,r') = \mathcal{L}_{\text{PL},p}(r',r) = 1.$$

The robust reward choice is any reward $r_{\mathcal{F},g}$ s.t. $\Pi^*(r_{\mathcal{F},g};p) = \{\pi^3\}$. Indeed, in this manner:

$$\mathcal{L}_{\text{PL},p}(r_{\mathcal{F},g},r) = \mathcal{L}_{\text{PL},p}(r_{\mathcal{F},g},r') = 0.5.$$

Thus, $r_{\mathcal{F},g} \notin \mathcal{R}_{\mathcal{F}}$.

Note that we can provide a counterexample with a convex feasible set by using as $\mathcal{R}_{\mathcal{F}}$ the convex hull of $r, r'$. In this way, it is simple to see that no reward in this new feasible set can make $\pi^3$ be the unique optimal policy. As such, the Chebyshev center is still external to $\mathcal{R}_{\mathcal{F}}$.

This concludes the proof. $\qquad\square$

**Proposition C.3.** *Let the feasible set $\mathcal{R}_{\mathcal{F}}$ be the 3-dimensional convex hull of rewards $[1,1,1]$, $[-1,1,1]$, $[1,-1,1]$, $[1,1,-1]$. Then, if we take $\mathcal{L}_g = \mathcal{L}_{\infty}$, the Chebyshev center is $r_{\mathcal{F},g} = [0,0,0]$ which lies outside $\mathcal{R}_{\mathcal{F}}$.*

*Proof.* See footnote 1 in Dabbene et al. (2014). $\qquad\square$

To quantify how convenient is our reward choice $r_{\mathcal{F},g}$ w.r.t. what is done in literature (see Eq. (1)), we have to understand how small is the ratio $\mathcal{I}_{\mathcal{F},g}/D_{\mathcal{F},g}$. However, since it coincides with the ratio between the Chebyshev radius and the diameter of the feasible set $\mathcal{R}_{\mathcal{F}}$ in premetric space $(\mathfrak{R}, \mathcal{L}_g)$, then the answer *depends* on the specific $\mathcal{R}_{\mathcal{F}}$ and $\mathcal{L}_g$ at stake.

If $\mathcal{L}_g$ is a metric, then the error reduction is at most 50%:

**Proposition C.4.** *If $\mathcal{L}_g$ satisfies simmetry and triangle inequality, then:*

$$\mathcal{I}_{\mathcal{F},g} \geqslant D_{\mathcal{F},g}/2.$$

*Proof.* Let $(r_1, r_2) \in \arg\max_{r,r'} \mathcal{L}_g(r,r')$, i.e., be the points in the diameter of the feasible set $\mathcal{R}_{\mathcal{F}}$, thus $D_{\mathcal{F},g} = \mathcal{L}_g(r_1, r_2)$. Because of triangle inequality, for any $r \in \mathfrak{R}$ including the Chebyshev

center $r_{\mathcal{F},g}$, it holds that:

$$D_{\mathcal{F},g} = \mathcal{L}_g(r_1, r_2) \leqslant \mathcal{L}_g(r_1, r) + \mathcal{L}_g(r, r_2).$$

If we take $r = r_{\mathcal{F},g}$ and apply simmetry, then we see that $\mathcal{L}_g(r_1, r_{\mathcal{F},g}) = \mathcal{L}_g(r_{\mathcal{F},g}, r_1) \leqslant \max_{r \in \mathcal{R}_{\mathcal{F}}} \mathcal{L}_g(r_{\mathcal{F},g}, r) = \mathcal{I}_{\mathcal{F},g}$, and also that $\mathcal{L}_g(r_{\mathcal{F},g}, r_2) \leqslant \max_{r \in \mathcal{R}_{\mathcal{F}}} \mathcal{L}_g(r_{\mathcal{F},g}, r) = \mathcal{I}_{\mathcal{F},g}$, from which:

$$D_{\mathcal{F},g} = \mathcal{L}_g(r_1, r_2) \leqslant 2\mathcal{I}_{\mathcal{F},g}.$$

$\square$

If $\mathcal{L}_g$ is induced by the 2-norm, we get an error reduction of at least $1/\sqrt{2}$:

**Proposition C.5.** *If $\mathcal{L}_g = \mathcal{L}_2$, then:*

$$\mathcal{I}_{\mathcal{F},g} \leqslant \sqrt{1 - 1/(SAH + 1)} D_{\mathcal{F},g}/\sqrt{2}.$$

As an example of why it is not exactly $50\%$, think to the equilateral triangle or to the simplex.

*Proof.* The result is immediate by an application of the Jung's theorem (Jung, 1901; Danzer et al., 1963; Scott, 1991), that states that in every Euclidean space with $n$ dimensions, the Chebyshev radius $r$ of a set with diameter $D$ satisfies:

$$r \leqslant \sqrt{\frac{n}{2(n+1)}} D.$$

$\square$

It should be noted that, in absence of the triangle inequality, Proposition C.4 does not hold, and the radius $\mathcal{I}_{\mathcal{F},g}$ can be zero when the diameter $D_{\mathcal{F},g}$ is not, providing an "infinite" increase in performance:

**Proposition C.6.** *There exist $p, \rho$ and feedback $\mathcal{F}$ such that the premetrics $\mathcal{L}_{PL,p}, \mathcal{L}_{GR,\rho}$ satisfy:*

$$\mathcal{I}_{\mathcal{F},(PL,p)} = 0 \text{ and } D_{\mathcal{F},(PL,p)} = H,$$
$$\mathcal{I}_{\mathcal{F},(GR,\rho)} = 0 \text{ and } D_{\mathcal{F},(GR,\rho)} = 1.$$

*Proof.* Let us begin with $\mathcal{L}_{PL,p}$. Consider a problem with horizon $H$ where $\mathcal{R}_{\mathcal{F}} = \{r \in \mathfrak{R} \mid \pi_1 \in \Pi^*(r; p)\}$ the feasible set contains all the rewards that make at least $\pi_1$ as optimal policy. Consider a transition model $p$ for which there exists a reward $\overline{r} \in \mathcal{R}_{\mathcal{F}}$ s.t. $J^*(\overline{r}; p) = J^{\pi^1}(\overline{r}; p) = H$, and for some other policy $\pi^2$ it holds that: $J^{\pi^2}(\overline{r}; p) = 0$ (for instance, if we construct $p$ to be deterministic, then, this is possible). Then, since $\mathcal{R}_{\mathcal{F}}$ contains also the reward $r'$ that makes all the policies optimal (in particular $\pi^2$), we have that:

$$D_{\mathcal{F},(PL,p)} \geqslant \mathcal{L}_{PL,p}(r', \overline{r}) = J^*(\overline{r}; p) - J^{\pi^2}(\overline{r}; p) = H.$$

Instead, if we consider any reward $r''$ s.t. $\Pi^*(r''; p) = \{\pi^1\}$ (as our robust reward choice), then it is clear that:

$$\mathcal{I}_{\mathcal{F},(PL,p)} \leqslant \max_{r \in \mathcal{R}_{\mathcal{F}}} \mathcal{L}_{PL,p}(r'', r) = J^*(r; p) - J^{\pi^1}(r; p) = 0.$$

The other claim of the proposition can be proved with an analagous construction. $\square$

## D  SOME INSIGHTS ON MODEL SELECTION AND ACTIVE LEARNING

In this appendix, we provide some considerations on model selection and active learning that make use of the framework and robust approach introduced in the main paper. Here, for simplicity of presentation, we introduce a new symbol $\mathbb{A}_f$: we will refer to a feedback $f$ as a pair $(Q_f, \mathbb{A}_f)$, where $\mathbb{A}_f$ denotes the assertion/statement that describes the relation between $Q_f$ and $r^*$, and, so, gives birth to the feasible set $\mathcal{R}_f$. It is convenient to introduce $\mathbb{A}_f$ because it permits to reason on different feedback $f, f'$ of the same type, i.e., sharing the same quantity $Q_f = Q_{f'}$, in a simple and intuitive way. We will call $\mathbb{A}_f$ an "assertion".

## D.1  Discussion on Model Selection

For simplicity of presentation, we consider a single feedback $\mathcal{F} = \{f\}$, with $f = (Q_f, \mathbb{A})$, and we assume that infinite data are available.

**Modelling feedback.** In practical applications, we observe the data $Q_f$, and we have to select the specific assertion $\mathbb{A}_f$ based on our knowledge on how $Q_f$ has been generated.

**Example D.1** (Feedback of driving safely). *Let the* unknown *target reward $r^\star$ represent the task of driving safely. Let $Q_f = \{\omega_i\}_i$ be a dataset of demonstrations collected by an agent $\pi^E$ that "aims to demonstrate how to drive safely in a certain environment". What assertion do we adopt?*

We can decide to model the problem in different ways, i.e., there are always multiple assertions $\mathbb{A}_f$ that can be associated to the given data $Q_f$ to *try* to capture the "true" relationship between $Q_f$ and the underlying unknown $r^\star$.

**Example D.1 (continue).** *We can model $\pi^E$ as an optimal policy for the "driving safely" task by using assertion $\mathbb{A}^{OPT}$, corresponding to the "optimal expert" entry in Table 1. If we think that the demonstrated $\pi^E$ sometimes takes suboptimal actions, then we might prefer using $\mathbb{A}^{MCE}$ ("$\beta$-MCE expert" in Table 1). Alternatively, we can simply assume that $\pi^E$ is at least $\epsilon$-optimal for some $\epsilon > 0$, i.e., $J^{\pi^E}(r^\star; p) \geqslant J^*(r^\star; p) - \epsilon$. We call this assertion $\mathbb{A}^{SUB}$ ("t-suboptimal expert" in Table 1).*

We incur in *misspecification* if $\mathbb{A}_f$ does not correctly describe the relationship between $r^\star$ and $Q_f$ (see Skalse & Abate (2024) for an analysis of misspecification in IRL).

**Example D.1 (continue).** *If we model data $Q_f$ using assertion $\mathbb{A}^{MCE}$, but, actually, $\pi^E$ is optimal (i.e., it follows $\mathbb{A}^{OPT}$), then our feedback is misspecified.*

**Choosing the correct modelling assertion.** The crucial question is: *what is the* best *modelling assertion for the given data $Q_f$?* Intuitively, *any* assertion that permits to carry out the downstream ReL application $g$ *effectively* is satisfactory. In other words, we can tolerate some misspecification error as long as the final outcome is acceptable.
Thanks to our framework, we can make these considerations more quantitative. As explained in Section 4, the uninformativeness $\mathcal{I}_{\{f\},g}$ represents the *minimum* error that can be achieved in the worst-case for doing $g$ using $f = (Q_f, \mathbb{A}_f)$. Under the model of feedback $f$ (i.e., the assertion $\mathbb{A}_f$) considered, the worst-case error cannot be smaller than $\mathcal{I}_{\{f\},g}$. Therefore, this is unsatisfactory if we aim to carry out $g$ with an error $\Delta < \mathcal{I}_{\{f\},g}$.
To solve this issue, we have to reduce the size of the feasible set $\mathcal{R}_f$ so that its Chebyshev radius $\mathcal{I}_{\{f\},g}$ reduces too. There are two ways for doing this. The best one $(i)$ consists in collecting additional feedback $f'$ to obtain $\mathcal{R}_f \cap \mathcal{R}_{f'} \subseteq \mathcal{R}_f$ (e.g., through active learning, see Appendix D.2). However, additional feedback might not be available in practice. The other way $(ii)$ consists in imposing "more structure" to the problem by changing the assertion $\mathbb{A}_f$ to $\mathbb{A}_{f'}$ to restrict the feasible set $\mathcal{R}_{\{Q_f, \mathbb{A}_f\}}$ to $\mathcal{R}_{\{Q_f, \mathbb{A}_{f'}\}}$.

**Example D.2.** *If we model $Q_f$ using $\mathbb{A}^{OPT}$, we obtain a feasible set that is* strictly *contained into the feasible set obtained using assertion $\mathbb{A}^{SUB}$. Thus, whatever the application $g$ at stake, $\mathcal{I}_{\{Q_f, \mathbb{A}^{OPT}\}, g} \leqslant \mathcal{I}_{\{Q_f, \mathbb{A}^{SUB}\}, g}$.*

By adopting a more restrictive (potentially less realistic) model $\mathbb{A}_{f'}$, we reduce the "measurable" error $\mathcal{I}_{\{Q_f, \mathbb{A}_{f'}\}, g}$ at the price of a larger unknown misspecification error. This is fine as long as $\mathbb{A}_{f'}$ is perceived as sufficiently realistic. *If there is no realistic assertion with a small enough value of informativeness*, then we conclude that the application $g$ cannot be carried out effectively with the only data $Q_f$. In other words, *the ReL problem cannot be solved with the desired accuracy*.

## D.2  Active Learning

In the Active Learning setting (Lopes et al., 2009), in addition to a given set of feedback $\mathcal{F}$, we can choose to receive a new feedback $f'$ from a set $\mathcal{F}' = \{f'_i\}_i$. Crucially, for any $f' \in \mathcal{F}'$, $f' = (Q_{f'}, \mathbb{A}_{f'})$, we consider the assumption $\mathbb{A}_{f'}$ to be known, while the actual data $Q_{f'}$ is revealed *after* our choice.

**Example D.3.** *We might choose between feedback $f_1 = (Q_1, \mathbb{A}^{OPT})$, consisting of demonstrations in environment $\mathcal{M}_1$ (i.e., $Q_1$) from an optimal expert (i.e., $\mathbb{A}^{OPT}$, see Example D.1), or $f_2 = (Q_2, \mathbb{A}^{MCE})$,*

*made of demonstrations in environment $\mathcal{M}_2$ (i.e., $Q_2$) from a maximal causal entropy expert (i.e., $\mathbb{A}^{MCE}$). After our choice, the expert demonstrates only one policy (i.e., either $Q_1$ or $Q_2$).*

What is the "best" choice of feedback $f$ to receive without knowing its data $Q_f$? Thanks to our framework, and specifically to the notion of uninformativeness defined in Section 4, the immediate choice is the feedback that, whatever its true data, is the most "informative":

**Definition D.1** (Information gain). *We define the* information gain $IG_{\mathcal{F},g}(f)$ *of a feedback $f$ w.r.t. $g$ given $\mathcal{F}$ as:*

$$IG_{\mathcal{F},g}(f) := \mathcal{I}_{\mathcal{F},g} - \mathcal{I}_{\mathcal{F}\cup\{f\},g}.$$

In words, $IG_{\mathcal{F},g}(f)$ represents the reduction of uninformativeness (worst-case error) of the ReL problem. Observe that $IG_{\mathcal{F},g}(f) \geqslant 0$, since $\mathcal{I}_{\mathcal{F}\cup\{f\},g} \leqslant \mathcal{I}_{\mathcal{F},g}$, i.e., the more feedback the less error.

Formally, the feedback that reduces the most the uninformativeness is the feedback $f'_j$ from set $\mathcal{F}' = \{f'_i\}_i, f'_i = (Q_i, \mathbb{A}_i)$, that maximizes the *worst-case* information gain w.r.t. the data $Q_i$:

$$j \in \arg\max_i \min_{Q_i} IG_{\mathcal{F},g}(\{Q_i, \mathbb{A}_i\}).$$

Simply put, we select the feedback that, whatever the true data we will receive, we know that it will bring the larger reduction of informativeness, i.e., of error, for the ReL problem.

# E    ADDITIONAL RESULTS ON SECTION 5

In this appendix, we provide additional results for Section 5. We begin by reporting the subroutine `PDSM-MAX` in Appendix E.1, which is not present in the main paper. Then, we provide explicit computation of the subgradients of the Lagrangian function $\widehat{L}$ for the estimated problem (Appendix E.2). We prove Theorem 5.3 in Appendix E.3. we provide a miscellaneous of other results in Appendix E.4, and we conclude with a discussion of possible extensions of **Rob-ReL** to continuous environments (Appendix E.5), and by showing that **Rob-ReL** can be easily extended to other ReL problems (Appendix E.6).

## E.1    PDSM-MAX

The pseudocode of the `PDSM-MAX` subroutine is reported in Algorithm 3. Note that it is analogous to that of `PDSM-MIN` (Algorithm 2), with the difference that set $\mathfrak{D}_-$ is mirrored w.r.t. $\mathfrak{D}_+$, and that the signs of the subgradients update are reversed.

We remark that operators $\Pi_{\mathfrak{R}}, \Pi_{\mathfrak{D}_-}, \Pi_{\mathfrak{D}_+}$ can be implemented in $\mathcal{O}(d)$ time for a vector in $\mathbb{R}^d$. Indeed, as explained in Boyd & Vandenberghe (2004), given a vector $x \in \mathbb{R}^d$, we can implement the projection onto set $\mathcal{Y} := [k_1, k_2]^d$ (with arbitrary $k_1 \leqslant k_2$) as:

$$[\Pi_{\mathcal{Y}}(x)]_j = \begin{cases} k_1 & \text{if } x_j \leqslant k_1 \\ k_2 & \text{if } x_j \geqslant k_2 \\ x_j & \text{otherwise} \end{cases},$$

where $[\cdot]_j$ denotes the $j$-th component.

Finally, we remark also that the explicit computation of the subgradients is explained in Appendix E.2.

## E.2    SUBGRADIENTS OF THE LAGRANGIAN

Both subroutines `PDSM-MIN` and `PDSM-MAX` require the computation of the subgradients $\partial_r \widehat{L}, \partial_\lambda \widehat{L}$ of the estimated Lagrangian $\widehat{L}$ (Eq. (11)) w.r.t. $r, \lambda$. We report Eq. (11) below for arbitrary $r, \lambda$:

$$\widehat{L}(r, \lambda) = \langle \widehat{d}^{\pi^1} - \widehat{d}^{\pi^2}, r \rangle + \sum_i \lambda_D^i \left( \max_\pi J^\pi(r; \widehat{p}_{D,i}) - \langle \widehat{d}^{\pi_{D,i}}, r \rangle - t_i \right)$$

$$+ \sum_i \lambda_{TC}^i \langle d^{\omega_{TC,i}^1} - d^{\omega_{TC,i}^2}, r \rangle + \sum_i \lambda_{PC}^i \langle \widehat{d}^{\pi_{PC,i}^1} - \widehat{d}^{\pi_{PC,i}^2}, r \rangle.$$

---

**Algorithm 3:** `PDSM-MAX`

---

**Input** : objective $\widehat{L}$, iterations $K$

1 $\lambda_0 \leftarrow 0$
2 $r_0 \leftarrow 0$
3 $\mathfrak{D}_- \leftarrow \{\lambda \leqslant 0 : \|\lambda\|_2 \leqslant s\}$
4 **for** $k = 0, 1, \ldots, K$ **do**
5  $\quad r_{k+1} \leftarrow \Pi_{\mathfrak{R}}\big(r_k + \alpha \partial_r \widehat{L}(r_k, \lambda_k)\big)$
6  $\quad \lambda_{k+1} \leftarrow \Pi_{\mathfrak{D}_-}\big(\lambda_k - \alpha \partial_\lambda \widehat{L}(r_k, \lambda_k)\big)$
7 **end**
8 $\widehat{r}_K \leftarrow \frac{1}{K} \sum_{k=0}^{K} r_k$
9 **Return** $\langle \widehat{d}^{\pi^1} - \widehat{d}^{\pi^2}, \widehat{r}_K \rangle$

---

Then, the following quantities are subgradients of $\widehat{L}$ evaluated at any $r \in \mathfrak{R}$ and $\lambda := [\lambda_{\text{TC}}, \lambda_{\text{C}}, \lambda_{\text{D}}]$:

$$\partial_r \widehat{L}(r, \lambda) = (\widehat{d}^{\pi^1} - \widehat{d}^{\pi^2}) + \sum_i \lambda_{\text{D}}^i \big(d^{\widehat{\pi}_{\text{D},i}^{r,*}} - \widehat{d}^{\pi_{\text{D},i}}\big) \tag{21}$$
$$+ \sum_i \lambda_{\text{TC}}^i \big(d^{\omega_{\text{TC},i}^1} - d^{\omega_{\text{TC},i}^2}\big) + \sum_i \lambda_{\text{PC}}^i \big(\widehat{d}^{\pi_{\text{PC},i}^1} - \widehat{d}^{\pi_{\text{PC},i}^2}\big),$$
$$\partial_{\lambda_{\text{D}}^i} \widehat{L}(r', \lambda') = \max_\pi J^\pi(r; \widehat{p}_{\text{D},i}) - \langle \widehat{d}^{\pi_{\text{D},i}}, r \rangle - t_i, \qquad \forall i \in [\![ m_{\text{D}} ]\!]$$
$$\partial_{\lambda_{\text{TC}}^i} \widehat{L}(r', \lambda') = \langle d^{\omega_{\text{TC},i}^1} - d^{\omega_{\text{TC},i}^2}, r \rangle, \qquad \forall i \in [\![ m_{\text{TC}} ]\!]$$
$$\partial_{\lambda_{\text{PC}}^i} \widehat{L}(r', \lambda') = \langle \widehat{d}^{\pi_{\text{PC},i}^1} - \widehat{d}^{\pi_{\text{PC},i}^2}, r \rangle, \qquad \forall i \in [\![ m_{\text{PC}} ]\!]$$

where $\widehat{\pi}_{\text{D},i}^{r,*} \in \arg\max_\pi J^\pi(r; \widehat{p}_{\text{D},i})$ represents an arbitrary optimal policy in the *estimated* MDP $(\mathcal{S}, \mathcal{A}, H, s_{0,\text{D},i}, \widehat{p}_{\text{D},i}, r)$, for all $i \in [\![ m_{\text{D}} ]\!]$, and $\widehat{d}^{\widehat{\pi}_{\text{D},i}^{r,*}}$ represents the visit distribution of $\widehat{\pi}_{\text{D},i}^{r,*}$ in the corresponding *estimated* MDP without reward $(\mathcal{S}, \mathcal{A}, H, s_{0,\text{D},i}, \widehat{p}_{\text{D},i})$.

To see that this is actually the subgradient of $\widehat{L}$, simply note that $\widehat{L}$ is linear in $\lambda$, thus the subgradients simply correspond to the gradients. Concerning $\partial_r \widehat{L}$, observe that $\widehat{L}$ is not differentiable in $r$ because it contains the pointwise maximum of differentiable functions $\max_\pi J^\pi(r; \widehat{p}_{\text{D},i})$. Thus, the subgradient $\partial_r \widehat{L}(r, \lambda)$ can be obtained as any convex combination of the gradients $d^{\widehat{\pi}_{\text{D},i}^{r,*}}$ of the functions that attain the maximum (Boyd et al., 2022).

We remark that the computation of $\max_\pi J^\pi(r; \widehat{p}_{\text{D},i})$ and $\widehat{\pi}_{\text{D},i}^{r,*} \in \arg\max_\pi J^\pi(r; \widehat{p}_{\text{D},i})$ can be done exactly using the *backward induction* algorithm Puterman (1994). Then, also $d^{\widehat{\pi}_{\text{D},i}^{r,*}}$ can be obtained exactly given $\widehat{\pi}_{\text{D},i}^{r,*}$ and $(\mathcal{S}, \mathcal{A}, H, s_{0,\text{D},i}, \widehat{p}_{\text{D},i})$ by simply starting from:

$$d_1^{\widehat{\pi}_{\text{D},i}^{r,*}}(s, a) = \widehat{\pi}_{\text{D},i,1}^{r,*}(a|s) \mathbb{1}\{s = s_{0,\text{D},i}\},$$

and then constructing the visit distribution for $h = 2, \ldots, H$ recursively using the relation Puterman (1994):

$$d_{h+1}^{\widehat{\pi}_{\text{D},i}^{r,*}}(s, a) = \widehat{\pi}_{\text{D},i,h+1}^{r,*}(a|s) \sum_{s', a'} d_h^{\widehat{\pi}_{\text{D},i}^{r,*}}(s', a') \widehat{p}_{\text{D},i,h}(s|s', a').$$

### E.3 PROOF OF THEOREM 5.3

**Theorem 5.3.** *Let $(\mathcal{F}, g)$ be a ReL problem as described in Section 5.1 for which Assumption 5.1 holds. Let $\epsilon \in (0, 2H]$ and $\delta \in (0, 1)$. If we set $s = 4H/\xi + \sqrt{(4H/\xi)^2 + SAH/4}$ and $\alpha = \epsilon/(16H(1 + s\sqrt{m_D + m_{TC} + m_{PC}})^2)$, then, with probability $1 - \delta$, **Rob-ReL** satisfies:*

$$\mathcal{L}_g(r^\star, \widehat{x}_K) \leqslant \mathcal{I}_{\mathcal{F},g} + \epsilon \quad \text{and} \quad |\mathcal{I}_{\mathcal{F},g} - \widehat{\mathcal{I}}_K| \leqslant \epsilon,$$

*with a number of samples:*

$$n^1, n^2, n_{D,i}, n_{PC,i}^1, n_{PC,i}^2 \leqslant \widetilde{\mathcal{O}}\Big(\frac{SAH^5}{\epsilon^2 \xi^2} \log \frac{1}{\delta}\Big),$$

$$N_{D,i} \leqslant \tilde{\mathcal{O}}\Big(\frac{SAH^5}{\epsilon^2 \xi^2}\Big(S + \log \frac{1}{\delta}\Big)\Big), \; N, N_{TC,i}, N_{PC,i} = 0,$$

*and a number of iterations:*

$$K \leqslant \mathcal{O}\left(\frac{H^{5/2}}{\xi \epsilon^2}\Big(\sqrt{SA} + \frac{\sqrt{H}}{\xi}\Big)\Big(1 + \sqrt{H(m_D + m_{TC} + m_{PC})\Big(\frac{H}{\xi^2} + SA\Big)}\Big)^2\right).$$

*Proof.* Define $M, m$ as:

$$M := \max_{r \in \mathcal{R}_{\mathcal{F}}} \langle d^{\pi^1} - d^{\pi^2}, r \rangle, \qquad m := \min_{r \in \mathcal{R}_{\mathcal{F}}} \langle d^{\pi^1} - d^{\pi^2}, r \rangle. \tag{22}$$

Then, we can write:

$$\mathcal{L}_g(r^\star, \widehat{x}_K) - \mathcal{I}_{\mathcal{F},g} \overset{(1)}{\leqslant} \max_{r \in \mathcal{R}_{\mathcal{F}}} \mathcal{L}_g(r, \widehat{x}_K) - \mathcal{I}_{\mathcal{F},g}$$

$$\overset{(2)}{=} \max\Big\{\widehat{x}_K - m, M - \widehat{x}_K\Big\} - \frac{M - m}{2}$$

$$= \max\Big\{\widehat{x}_K - \frac{M+m}{2}, \frac{M+m}{2} - \widehat{x}_K\Big\}$$

$$\overset{(3)}{=} \max\Big\{\frac{\widehat{M}_K + \widehat{m}_K}{2} - \frac{M+m}{2}, \frac{M+m}{2} - \frac{\widehat{M}_K + \widehat{m}_K}{2}\Big\}$$

$$\overset{(4)}{\leqslant} \frac{1}{2}\Big(\big|\widehat{M}_K - M\big| + \big|\widehat{m}_K - m\big|\Big)$$

$$\overset{(5)}{\leqslant} \frac{1}{2}\Big(\big|\widehat{M}_K - \widehat{M}\big| + \big|\widehat{M} - M\big| + \big|\widehat{m}_K - \widehat{m}\big| + \big|\widehat{m} - m\big|\Big),$$

where at (1) we used that $r^\star \in \mathcal{R}_{\mathcal{F}}$ by hypothesis, at (2) we apply Lemma E.8 twice using the definitions of $\mathcal{I}_{\mathcal{F},g}$ and $\mathcal{L}_g$, at (3) we insert the definitions of $\widehat{M}_K, \widehat{m}_K$ as computed by `Rob-ReL`, at (4) we apply triangle's inequality, and at (5) we add and subtract $\widehat{M}, \widehat{m}$ and apply again triangle's inequality.

Similarly, note that:

$$\Big|\mathcal{I}_{\mathcal{F},g} - \widehat{\mathcal{I}}_{\mathcal{F},g}\Big| = \Big|\frac{M-m}{2} - \frac{\widehat{M}_K - \widehat{m}_K}{2}\Big|$$

$$\leqslant \frac{1}{2}\Big(\big|\widehat{M}_K - \widehat{M}\big| + \big|\widehat{M} - M\big| + \big|\widehat{m}_K - \widehat{m}\big| + \big|\widehat{m} - m\big|\Big).$$

Observe that $\big|\widehat{M}_K - \widehat{M}\big| + \big|\widehat{m}_K - \widehat{m}\big|$ represents the iteration error, while $\big|\widehat{M} - M\big| + \big|\widehat{m} - m\big|$ the estimation error. Let $\epsilon_{\text{EST}} \in (0,1)$, and define the good event $\mathcal{E}$ as the intersection of the following events:

$$\mathcal{E}_g := \Big\{\sup_{r \in \mathfrak{R}} \big|\langle \widehat{d}^{\pi^i}, r \rangle - J^{\pi^i}(r; p)\big| \leqslant \epsilon_{\text{EST}}/2 \quad \forall i \in \{1, 2\}\Big\},$$

$$\mathcal{E}_D := \Big\{\sup_{r \in \mathfrak{R}} \big|\langle \widehat{d}^{\pi_{\text{D},i}}, r \rangle - J^{\pi_{\text{D},i}}(r; p_{\text{D},i})\big| \leqslant \epsilon_{\text{EST}}/2 \quad \forall i \in [\![m_{\text{D}}]\!]\Big\},$$

$$\mathcal{E}_{D,*} := \Big\{\sup_{r \in \mathfrak{R}} \big|\max_{\pi} J^{\pi}(r; \widehat{p}_{\text{D},i}) - J^*(r; p_{\text{D},i})\big| \leqslant \epsilon_{\text{EST}}/2 \quad \forall i \in [\![m_{\text{D}}]\!]\Big\}, \tag{23}$$

$$\mathcal{E}_{\text{PC},1} := \Big\{\sup_{r \in \mathfrak{R}} \big|\langle \widehat{d}^{\pi^1_{\text{PC},i}}, r \rangle - J^{\pi^1_{\text{PC},i}}(r; p_{\text{PC},i})\big| \leqslant \epsilon_{\text{EST}}/2 \quad \forall i \in [\![m_{\text{PC}}]\!]\Big\},$$

$$\mathcal{E}_{\text{PC},2} := \Big\{\sup_{r \in \mathfrak{R}} \big|\langle \widehat{d}^{\pi^2_{\text{PC},i}}, r \rangle - J^{\pi^2_{\text{PC},i}}(r; p_{\text{PC},i})\big| \leqslant \epsilon_{\text{EST}}/2 \quad \forall i \in [\![m_{\text{PC}}]\!]\Big\}.$$

Then, conditioning on $\mathcal{E}$, Lemma E.2 allows to bound the estimation error, while Lemma E.6 permits to bound the iteration error.

The result follows by applying Lemma E.1 with $\epsilon_{\text{EST}} < \frac{\xi \epsilon}{10H}$. $\qquad\qquad\square$

### E.3.1 CONCENTRATION

**Lemma E.1** (Concentration). *Let $\epsilon_{EST} \in (0,1), \delta \in (0,1)$. Then, the good event $\mathcal{E} := \mathcal{E}_g \cap \mathcal{E}_D \cap \mathcal{E}_{D,*} \cap \mathcal{E}_{PC,1} \cap \mathcal{E}_{PC,2}$, where these events are defined in Eq. (23), holds with probability at least $1 - \delta$, with at most:*

$$n^1, n^2, n_{D,i}, n^1_{PC,i}, n^2_{PC,i} \leqslant \tilde{\mathcal{O}}\Big(\frac{SAH^3}{\epsilon^2_{EST}} \log \frac{m_{PC} + m_D + 2}{\delta}\Big),$$

$$N_{D,i} \leqslant \tilde{\mathcal{O}}\Big(\frac{SAH^3}{\epsilon^2_{EST}} \Big(S + \log \frac{m_{PC} + m_D + 2}{\delta}\Big)\Big). \tag{24}$$

*Proof.* `Rob-ReL` estimates the visit distribution of the considered policies through their empirical estimates, defined in Eq. (8). Thus, events $\mathcal{E}_g, \mathcal{E}_D, \mathcal{E}_{PC,1} \cap \mathcal{E}_{PC,2}$ can be shown to hold using the same proof as in Theorem 5.1 of Lazzati et al. (2024a) (see also Shani et al. (2022)), and a union bound.

Regarding event $\mathcal{E}_{D,*}$, we can use the guarantees in Menard et al. (2021) (see also Lazzati et al. (2024a)).

The result follows by an application of the union bound.

We remark that, of course, in case $m_{PC} = m_D = 0$, then the good event holds with a number of samples:

$$n_{D,i}, n^1_{PC,i}, n^2_{PC,i}, N_{D,i} = 0,$$

$$n^1, n^2 \leqslant \tilde{\mathcal{O}}\Big(\frac{SAH^3}{\epsilon^2_{EST}} \log \frac{1}{\delta}\Big).$$

Nevertheless, for simplicity of presentation, we will mention only the bound in Eq. (24). $\square$

### E.3.2 BOUNDING THE *Estimation Error*

**Lemma E.2** (Estimation error). *Let $\epsilon \in (0, 2H], \delta \in (0,1)$. Make Assumption 5.1, and assume that event $\mathcal{E}$ holds with $\epsilon_{EST} < \frac{\xi\epsilon}{10H}$. Then, we have that:*

$$|M - \widehat{M}| + |m - \widehat{m}| \leqslant \epsilon.$$

*Proof.* We proceed for $M, \widehat{M}$ (the proof for $m, \widehat{m}$ is completely analogous):

$$\Big|M - \widehat{M}\Big| \stackrel{(1)}{=} \Big| \min_{\lambda \in \mathfrak{D}_-} \max_{r \in \mathfrak{R}} L(r, \lambda) - \min_{\lambda \in \mathfrak{D}_-} \max_{r \in \mathfrak{R}} \widehat{L}(r, \lambda)\Big|$$

$$\stackrel{(2)}{\leqslant} \max_{\lambda \in \mathfrak{D}_-}\Big| \max_{r \in \mathfrak{R}} L(r, \lambda) - \max_{r \in \mathfrak{R}} \widehat{L}(r, \lambda)\Big|$$

$$\stackrel{(3)}{\leqslant} \max_{\lambda \in \mathfrak{D}_-} \max_{r \in \mathfrak{R}}\Big| L(r, \lambda) - \widehat{L}(r, \lambda)\Big|$$

$$= \max_{\lambda \in \mathfrak{D}_-} \max_{r \in \mathfrak{R}}\Big| \langle \widehat{d}^{\pi^1} - \widehat{d}^{\pi^2}, r\rangle - \langle d^{\pi^1} - d^{\pi^2}, r\rangle$$

$$+ \sum_i \lambda^i_D\Big( \big(\max_\pi J^\pi(r; \widehat{p}_{D,i}) - \langle \widehat{d}^{\pi_{D,i}}, r\rangle\big) - \max_\pi J^\pi(r; p_{D,i}) - \langle d^{\pi_{D,i}}, r\rangle\Big)$$

$$+ \sum_i \lambda^i_{PC}\Big( \langle \widehat{d}^{\pi^1_{PC,i}} - \widehat{d}^{\pi^2_{PC,i}}, r\rangle - \langle d^{\pi^1_{PC,i}} - d^{\pi^2_{PC,i}}, r\rangle\Big)$$

$$\stackrel{(4)}{\leqslant} \epsilon_{EST}\Big(1 + \max_{\lambda \in \mathfrak{D}_-} \|\lambda\|_1\Big)$$

$$\stackrel{(5)}{\leqslant} \epsilon_{EST} \frac{5H}{\xi},$$

where at (1) we have applied Lemma E.3 and Lemma E.4, at (2) and at (3) the Lipschitzianity of the maximum operator. At (4) we use triangle's inequality, that event $\mathcal{E}$ holds and we upper bound with the 1-norm of $\lambda$, and at (5) we use the size of set $\mathfrak{D}_-$ provided by Lemma E.4.

By choosing:

$$\epsilon_{\text{EST}} < \frac{\xi\epsilon}{10H},$$

which is smaller than $\xi/2$ for $\epsilon \in (0, 2H]$, as required by Lemma E.4, we get the result. $\square$

**Lemma E.3.** *Let $L$ be the Lagrangian of the problem:*

$$L(r, \lambda) = \langle d^{\pi^1} - d^{\pi^2}, r \rangle + \sum_i \lambda_D^i \big( \max_\pi J^\pi(r; p_{D,i}) - \langle d^{\pi_{D,i}}, r \rangle - t_i \big) \tag{25}$$
$$+ \sum_i \lambda_{TC}^i \langle d^{\omega_{TC,i}^1} - d^{\omega_{TC,i}^2}, r \rangle + \sum_i \lambda_{PC}^i \langle d^{\pi_{PC,i}^1} - d^{\pi_{PC,i}^2}, r \rangle.$$

*Then, under Assumption 5.1, it holds that:*

$$M = \min_{\lambda \in \mathfrak{D}_-} \max_{r \in \mathfrak{R}} L(r, \lambda),$$
$$m = \max_{\lambda \in \mathfrak{D}_+} \min_{r \in \mathfrak{R}} L(r, \lambda),$$

*where $\mathfrak{D}_+, \mathfrak{D}_-$ correspond to $s = 4H/\xi$ in Algorithms 2 and 3, namely, $\mathfrak{D}_+ = \{\lambda \geqslant 0 \,|\, \|\lambda\|_1 \leqslant 4H/\xi\}$ and $\mathfrak{D}_- = \{\lambda \leqslant 0 \,|\, \|\lambda\|_1 \leqslant 4H/\xi\}$.*

*Proof.* Under Assumption 5.1, we have that Slater's constraint qualification holds. Thanks also to the linearity of the objective function and to the convexity of $\mathcal{R}_\mathcal{F}$, we have that strong duality holds Boyd & Vandenberghe (2004), thus:

$$M = \min_{\lambda \leqslant 0} \max_{r \in \mathfrak{R}} L(r, \lambda),$$
$$m = \max_{\lambda \geqslant 0} \min_{r \in \mathfrak{R}} L(r, \lambda).$$

To prove the boundedness of the Lagrange multipliers corresponding to the saddle point, let $(r^*, \lambda^*)$ be any saddle point for problem $M$ (the proof for $m$ is analogous). Under Assumption 5.1, we can apply Lemma 3 of Nedić & Ozdaglar (2009) to obtain that, for the reward $\bar{r}$ in Assumption 5.1:

$$\|\lambda^*\|_1 \leqslant \frac{\langle d^{\pi^1} - d^{\pi^2}, \bar{r} \rangle - M}{\xi}$$
$$\leqslant \frac{\langle d^{\pi^1} - d^{\pi^2}, \bar{r} \rangle + H}{\xi}$$
$$\leqslant \frac{2H}{\xi}.$$

Thus, the values of the Lagrange multipliers in saddle points can be found in bounded sets $\mathfrak{D}_+, \mathfrak{D}_-$. Note that we upper bound again with $4H/\xi$ to comply with the bound for the estimated problem in Lemma E.4. $\square$

**Lemma E.4.** *Make Assumption 5.1. Under the good event $\mathcal{E}$ with $\epsilon_{EST} \in (0, \xi/2)$, we have:*

$$\widehat{M} = \min_{\lambda \in \mathfrak{D}_-} \max_{r \in \mathfrak{R}} \widehat{L}(r, \lambda),$$
$$\widehat{m} = \max_{\lambda \in \mathfrak{D}_+} \min_{r \in \mathfrak{R}} \widehat{L}(r, \lambda),$$

*where $\mathfrak{D}_- = \{\lambda \leqslant 0 \,|\, \|\lambda\|_1 \leqslant 4H/\xi\}$ and $\mathfrak{D}_+ = \{\lambda \geqslant 0 \,|\, \|\lambda\|_1 \leqslant 4H/\xi\}$.*

*Proof.* The proof is analogous to that of Lemma E.3 as long as we use Lemma E.5. $\square$

**Lemma E.5** (Slater's Condition)**.** *Make Assumption 5.1. Under the good event $\mathcal{E}$ with $\epsilon_{EST} \in (0, \xi/2)$, for the reward $\bar{r}$ in Assumption 5.1, we have:*

$$\begin{cases} \max_\pi J^\pi(\bar{r}; \widehat{p}_{D,i}) - \langle \widehat{d}^{\pi_{D,i}}, \bar{r} \rangle - t_i \leqslant -\frac{\xi}{2} & \forall i \in [\![m_D]\!] \\ \langle d^{\omega_{TC,i}^1} - d^{\omega_{TC,i}^2}, \bar{r} \rangle \leqslant -\frac{\xi}{2} & \forall i \in [\![m_{PC}]\!] \\ \langle \widehat{d}^{\pi_{PC,i}^1} - \widehat{d}^{\pi_{PC,i}^2}, \bar{r} \rangle \leqslant -\frac{\xi}{2} & \forall i \in [\![m_{TC}]\!] \end{cases}.$$

*Proof.* The result follows directly by using the conditions in the good event $\mathcal{E}$. For instance, concerning the policy comparison feedback, we have:

$$\langle \widehat{d}^{\pi^1_{\text{PC}},i} - \widehat{d}^{\pi^2_{\text{PC}},i}, \overline{r} \rangle = \langle \widehat{d}^{\pi^1_{\text{PC}},i} - \widehat{d}^{\pi^2_{\text{PC}},i}, \overline{r} \rangle \pm \left( \langle d^{\pi^1_{\text{PC}},i} - d^{\pi^2_{\text{PC}},i}, \overline{r} \rangle \right)$$

$$\leqslant \epsilon_{\text{EST}} + \left( \langle d^{\pi^1_{\text{PC}},i} - d^{\pi^2_{\text{PC}},i}, \overline{r} \rangle \right)$$

$$\leqslant \epsilon_{\text{EST}} - \xi.$$

By using that $\epsilon_{\text{EST}} \in (0, \xi/2)$ we get the result. $\qquad\square$

### E.3.3 BOUNDING THE *Iteration Error*

**Lemma E.6** (Iteration error). *Let $\epsilon \in (0, 2H]$, make Assumption 5.1 and assume that event $\mathcal{E}$ with $\epsilon_{\text{EST}} \in (0, \xi/2)$ holds. Then, if we choose:*

$$s = 4H/\xi + \sqrt{(4H/\xi)^2 + SAH/4},$$

$$\alpha = \epsilon/(16H(1 + s\sqrt{m_D + m_{TC} + m_{PC}})^2),$$

*we have that:*

$$|\widehat{M} - \widehat{M}_K| + |\widehat{m} - \widehat{m}_K| \leqslant \epsilon,$$

*with a number of iterations:*

$$K \leqslant \mathcal{O}\left( \frac{H^{5/2}}{\xi\epsilon^2} \left( \sqrt{SA} + \frac{\sqrt{H}}{\xi} \right) \left( 1 + \sqrt{H(m_D + m_{TC} + m_{PC})(H/\xi^2 + SA)} \right)^2 \right).$$

*Proof.* We prove the result by imposing that both the terms $|\widehat{M} - \widehat{M}_K|$ and $|\widehat{m} - \widehat{m}_K|$ are smaller than $\epsilon/2$. We present the proof only for $|\widehat{M} - \widehat{M}_K|$, because the proof for $|\widehat{m} - \widehat{m}_K|$ is completely analogous. For simplicity, define $\ell$ as:

$$\ell := \sqrt{\left( \frac{4H}{\xi} \right)^2 + \left( \frac{SAH}{4} \right)},$$

and note that, in this way, for the choice of $s$ in the statement, we have:

$$s = \frac{4H}{\xi} + \ell.$$

Thanks to Lemma E.7, using this choice of $s$, we obtain the following upper bound to the norm of the subgradients, that we call $U$:

$$U := 2\sqrt{H}\left( 1 + s\sqrt{|\mathcal{F}|} \right) = 2\sqrt{H} + 2\left( \frac{4H}{\xi} + \ell \right)\sqrt{H|\mathcal{F}|}.$$

Observe that the choice of $\alpha$ corresponds to:

$$\alpha = \frac{\epsilon}{4U^2}.$$

Using Lemma E.5 and Lemma E.7, we can apply Proposition 2 of Nedić & Ozdaglar (2009) to obtain (recall that we start with $r_0 = 0, \lambda_0 = 0$):

$$\widehat{M}_K - \widehat{M} \leqslant \frac{\|\widehat{r}_K\|_2^2}{2K\alpha} + \alpha U^2, \tag{26}$$

$$\widehat{M} - \widehat{M}_K \leqslant \frac{4H}{\xi}\left( \frac{2}{K\alpha\ell}\left( \frac{4H}{\xi} + \ell \right)^2 + \frac{\|\widehat{r}_K\|_2^2}{2K\alpha\ell} + \frac{\alpha U^2}{2\ell} \right). \tag{27}$$

We now bound each of these terms.

Concerning the first term, i.e., Eq. (26), using that $\widehat{r}_K \in \mathfrak{R}$ and the choice $\alpha = \epsilon/(4U^2)$ we have:

$$\widehat{M}_K - \widehat{M} \leqslant \frac{SAH}{2K\alpha} + \alpha U^2,$$

$$\leqslant \frac{2SAHU^2}{\epsilon K} + \frac{\epsilon}{4},$$

which is smaller than $\epsilon/4$ if:

$$K \geqslant \frac{8SAHU^2}{\epsilon^2}.$$

Regarding the second term, namely, Eq. (27), we write:

$$
\begin{aligned}
\widehat{M} - \widehat{M}_K &\leqslant \frac{4H}{\xi}\frac{2}{K\alpha\ell}\Big(\frac{4H}{\xi}+\ell\Big)^2 + \frac{4H}{\xi}\frac{SAH}{2K\alpha\ell} + \frac{4H}{\xi}\frac{\alpha U^2}{2\ell} \\
&\overset{(1)}{\leqslant} \frac{4H}{\xi}\frac{2}{K\alpha\ell}\Big(\frac{4H}{\xi}+\ell\Big)^2 + \frac{4H}{\xi}\frac{SAH}{2K\alpha\ell} + \frac{\epsilon}{4} \\
&\overset{(2)}{\leqslant} \frac{4H}{\xi}\frac{2}{K\alpha\ell}\Big(\frac{4H}{\xi}+\ell\Big)^2 + \frac{4H}{\xi}\frac{\sqrt{SAH}}{K\alpha} + \frac{\epsilon}{4} \\
&\overset{(3)}{\leqslant} \frac{4H}{\xi}\frac{2}{K\alpha}\Big(\frac{16H}{\xi}+\frac{\sqrt{SAH}}{2}\Big) + \frac{4H}{\xi}\frac{\sqrt{SAH}}{K\alpha} + \frac{\epsilon}{4} \\
&\leqslant \frac{4H}{\xi}\frac{2}{K\alpha}\Big(\frac{16H}{\xi}+\sqrt{SAH}\Big) + \frac{\epsilon}{4} \\
&= \frac{8H^{3/2}}{\xi K\alpha}\Big(\frac{16\sqrt{H}}{\xi}+\sqrt{SA}\Big) + \frac{\epsilon}{4} \\
&\overset{(4)}{=} \frac{32U^2 H^{3/2}}{\xi K\epsilon}\Big(\frac{16\sqrt{H}}{\xi}+\sqrt{SA}\Big) + \frac{\epsilon}{4},
\end{aligned}
$$

where at (1) we use that $\ell \geqslant 2H/\xi$ and the choice of $\alpha = \epsilon/(4U^2)$, at (2) we use that $\ell \geqslant \sqrt{\frac{SAH}{4}}$, at (3) we use that $\Big(\frac{4H}{\xi}+\ell\Big)^2/\ell \leqslant \frac{16H}{\xi} + \frac{\sqrt{SAH}}{2}$ and at (4) we use that $\alpha = \epsilon/(4U^2)$.

The resulting quantity is smaller than $\epsilon/2$ if:

$$\frac{32U^2 H^{3/2}}{\xi K\epsilon}\Big(\frac{16\sqrt{H}}{\xi}+\sqrt{SA}\Big) \leqslant \frac{\epsilon}{4} \quad \Longleftrightarrow \quad K \geqslant \frac{128H^{3/2}U^2}{\xi\epsilon^2}\Big(\sqrt{SA}+\frac{16\sqrt{H}}{\xi}\Big).$$

By inserting the definition of $U$, we get:

$$
\begin{aligned}
K &\geqslant \frac{128H^{3/2}U^2}{\xi\epsilon^2}\Big(\sqrt{SA}+\frac{16\sqrt{H}}{\xi}\Big) \\
&= \frac{128H^{3/2}}{\xi\epsilon^2}\Big(\sqrt{SA}+\frac{16\sqrt{H}}{\xi}\Big)\Big(2\sqrt{H}+2\Big(\frac{4H}{\xi}+\ell\Big)\sqrt{H|\mathcal{F}|}\Big)^2 \\
&\leqslant \mathcal{O}\Big(\frac{H^{5/2}}{\xi\epsilon^2}\Big(\sqrt{SA}+\frac{\sqrt{H}}{\xi}\Big)\Big(1+\sqrt{H|\mathcal{F}|(H/\xi^2+SA)}\Big)^2\Big).
\end{aligned}
$$

The result follows by noting that this quantity is larger than the upper bound derived for Eq. (26).

$\square$

**Lemma E.7.** *If we execute* `PDSM-MIN` *or* `PDSM-MAX` *for $K \geqslant 0$ iterations using any choice of $s \geqslant 0$, then it holds that:*

$$\max_{k \leqslant K}\max\Big\{\|\partial_r \widehat{L}(r_k,\lambda_k)\|_2, \|\partial_\lambda \widehat{L}(r_k,\lambda_k)\|_2\Big\} \leqslant 2\max\Big\{\sqrt{H}\big(1+s\sqrt{|\mathcal{F}|}\big), H\sqrt{|\mathcal{F}|}\Big\},$$

*where we denoted $|\mathcal{F}| := m_D + m_{TC} + m_{PC}$.*

*Proof.* We prove the result for `PDSM-MAX` only, because for `PDSM-MIN` the proof is analogous. Consider the subgradient expressions in Eq. (21). We have (the dependence on $s$ is implicit in $\mathfrak{D}_-$):

$$
\begin{aligned}
\max_{k \leqslant K}\|\partial_r \widehat{L}(r_k,\lambda_k)\|_2 &\overset{(1)}{\leqslant} \max_{r \in \mathfrak{R}}\max_{\lambda \in \mathfrak{D}_-}\|\partial_r \widehat{L}(r,\lambda)\|_2 \\
&= \max_{r \in \mathfrak{R}}\max_{\lambda \in \mathfrak{D}_-}\Big\|\big(\widehat{d}^{\pi^1}-\widehat{d}^{\pi^2}\big)+\sum_i \lambda_D^i\big(d^{\widehat{\pi}_{D,i}^{r,*}}-\widehat{d}^{\pi_{D,i}}\big)
\end{aligned}
$$

$$+ \sum_i \lambda_{\mathrm{TC}}^i \big(d^{\omega_{\mathrm{TC,i}}^1} - d^{\omega_{\mathrm{TC,i}}^2}\big) + \sum_i \lambda_{\mathrm{PC}}^i \big(\widehat{d}^{\pi_{\mathrm{PC}}^1,i} - \widehat{d}^{\pi_{\mathrm{PC}}^2,i}\big) \Big\|_2$$

$$\leqslant \max_{r \in \mathfrak{R}} \max_{\lambda \in \mathfrak{D}_-} \Big\| \widehat{d}^{\pi^1} - \widehat{d}^{\pi^2} \Big\|_2 + \sum_i |\lambda_{\mathrm{D}}^i| \Big\| d^{\widehat{\pi}_{\mathrm{D},i}^{r,*}} - \widehat{d}^{\pi_{\mathrm{D},i}} \Big\|_2$$

$$+ \sum_i |\lambda_{\mathrm{TC}}^i| \Big\| d^{\omega_{\mathrm{TC,i}}^1} - d^{\omega_{\mathrm{TC,i}}^2} \Big\|_2 + \sum_i |\lambda_{\mathrm{PC}}^i| \Big\| \widehat{d}^{\pi_{\mathrm{PC}}^1,i} - \widehat{d}^{\pi_{\mathrm{PC}}^2,i} \Big\|_2$$

$$\overset{(2)}{\leqslant} 2\sqrt{H} \max_{\lambda \in \mathfrak{D}_-} \big(1 + \|\lambda\|_1\big)$$

$$\overset{(3)}{\leqslant} 2\sqrt{H} \Big(1 + \sqrt{m_{\mathrm{D}} + m_{\mathrm{TC}} + m_{\mathrm{PC}}} \max_{\lambda \in \mathfrak{D}_-} \|\lambda\|_2 \Big)$$

$$= 2\sqrt{H} \Big[1 + s\sqrt{m_{\mathrm{D}} + m_{\mathrm{TC}} + m_{\mathrm{PC}}} \Big],$$

where at (1) we use that the PDSM projects onto those sets, thus $r_k, \lambda_k$ cannot lie outside, at (2) we use the fact that the 2-norm of every visit distribution is at most $\sqrt{H}$, at (3) we use the relationship between the 1-norm and the 2-norm, recalling that $\lambda$ is $(m_{\mathrm{D}} + m_{\mathrm{TC}} + m_{\mathrm{PC}})$-dimensional.

Regarding the subgradients w.r.t. $\lambda$, we can write:

$$\max_{k \leqslant K} \|\partial_\lambda \widehat{L}(r_k, \lambda_k)\|_2 \overset{(4)}{\leqslant} \max_{r \in \mathfrak{R}} \max_{\lambda \in \mathfrak{D}_-} \|\partial_\lambda \widehat{L}(r, \lambda)\|_2$$

$$\overset{(5)}{=} \max_{r \in \mathfrak{R}} \Big\| \Big[ \partial_{\lambda_{\mathrm{D}}^i} \widehat{L}(r, \lambda), \ldots, \partial_{\lambda_{\mathrm{TC}}^i} \widehat{L}(r, \lambda), \ldots, \partial_{\lambda_{\mathrm{PC}}^i} \widehat{L}(r, \lambda) \Big] \Big\|_2$$

$$\overset{(6)}{\leqslant} \Big\| \Big[ 2H, \ldots, H, \ldots, H \Big] \Big\|_2$$

$$\leqslant 2H\sqrt{m_{\mathrm{D}} + m_{\mathrm{TC}} + m_{\mathrm{PC}}},$$

where at (4) we use that the PDSM projects onto those sets, thus $r_k, \lambda_k$ cannot lie outside, at (5) we note that the subgradient vector is $(m_{\mathrm{D}} + m_{\mathrm{TC}} + m_{\mathrm{PC}})$-dimensional, and write down its components, noting also that there is no dependence on $\lambda$. At (6) we use the formulas in Eq. (21) and note that the difference between returns and expected returns is bounded in $[-H, +H]$ because $r \in \mathfrak{R}$, and also that $t_i \in [0, H]$ for all $i \in [\![m_{\mathrm{D}}]\!]$ by hypothesis.

The result follows by joining the two upper bounds. $\qquad\square$

### E.4 OTHER PROOFS AND RESULTS

**Lemma E.8.** *Let $d \in \mathbb{N}$, and let $\mathcal{Y}$ be a subset of $\mathbb{R}^d$. Let $h : \mathcal{Y} \to \mathbb{R}$ be any function that attains both minimum and maximum in $\mathcal{Y}$, and define $M_h := \max_{y \in \mathcal{Y}} h(y)$ and $m_h := \min_{y \in \mathcal{Y}} h(y)$. Then:*

$$\frac{M_h - m_h}{2} = \min_{k \in \mathbb{R}} \max_{y \in \mathcal{Y}} \Big| k - h(y) \Big|,$$

$$\frac{M_h + m_h}{2} = \arg\min_{k \in \mathbb{R}} \max_{y \in \mathcal{Y}} \Big| k - h(y) \Big|,$$

$$\max_{y \in \mathcal{Y}} \Big| k - h(y) \Big| = \max\Big\{ k - m_h, M_h - k \Big\} \quad \forall k \in \mathbb{R}.$$

*Proof.* For any $k \in \mathbb{R}$, we can write:

$$\max_{y \in \mathcal{Y}} \Big| k - h(y) \Big| = \max_{y \in \mathcal{Y}} \max\Big\{ k - h(y), h(y) - k \Big\}$$

$$= \max\Big\{ \max_{y \in \mathcal{Y}} k - h(y), \max_{y \in \mathcal{Y}} h(y) - k \Big\}$$

$$= \max\Big\{ k - \min_{y \in \mathcal{Y}} h(y), \max_{y \in \mathcal{Y}} h(y) - k \Big\}$$

$$= \max\Big\{ k - m_h, M_h - k \Big\}.$$

Then, we know that the maximum between two items is minimized when the two items coincide. Thus, the argmin must satisfy:

$$k - m_h = M_h - k \quad \Longleftrightarrow \quad k = \frac{M_h + m_h}{2}.$$

Substituting this value into the previous expression, we get:

$$\max\left\{\frac{M_h + m_h}{2} - m_h, M_h - \frac{M_h + m_h}{2}\right\} = \frac{M_h - m_h}{2}.$$

This concludes the proof. $\qquad\square$

**Proposition 5.1.** *It holds that:* $\widehat{x}_{\mathcal{F},g} = (\widehat{M} + \widehat{m})/2.$

*Proof.* Apply Lemma E.8. $\qquad\square$

**Proposition 5.2.** *It holds that:* $\widehat{\mathcal{I}}_{\mathcal{F},g} := \max_{r \in \widehat{\mathcal{R}}_{\mathcal{F}}} \widehat{\mathcal{L}}_g(r, \widehat{x}_{\mathcal{F},g}) = (\widehat{M} - \widehat{m})/2.$

*Proof.* Apply Lemma E.8. $\qquad\square$

### E.4.1 USING ROB−REL FOR ESTIMATING THE WORST-CASE LOSS

Thanks to Lemma E.8, we realize that the worst-case loss suffered by any object $x \in \mathcal{X}_g$ can be equivalently computed as:

$$\max_{r \in \mathcal{R}_{\mathcal{F}}} \mathcal{L}_g(r, x) = \max\left\{x - m, M - x\right\}.$$

This suggests that we can use **Rob−ReL** to quantify the error suffered in the worst case by arbitrary $x \in \mathcal{X}_g$, potentially the output provided by other ReL algorithms. To do so, we can use the estimate:

$$\max\left\{x - \widehat{m}_K, \widehat{M}_K - x\right\},$$

and note that:

$$\left|\max\left\{x - \widehat{m}_K, \widehat{M}_K - x\right\} - \max\left\{x - m, M - x\right\}\right| \leqslant \max\left\{|m - \widehat{m}_K|, |M - \widehat{M}_K|\right\}$$
$$\leqslant |m - \widehat{m}_K| + |M - \widehat{M}_K|$$
$$\leqslant (|m - \widehat{m}| + |M - \widehat{M}|) + (|\widehat{m}_K - \widehat{m}| + |\widehat{M}_K - \widehat{M}|),$$

namely, the error can be upper bounded by the estimation and iteration errors considered by Theorem 5.3. Therefore, we have the following corollary:

**Corollary E.9.** *Let $x \in \mathcal{X}_g$ be arbitrary. Under the setting of Theorem 5.3, with the number of samples and iterations specified in the statement of the theorem, with probability $1 - \delta$, it holds that:*

$$\left|\max_{r \in \mathcal{R}_{\mathcal{F}}} \mathcal{L}_g(r, x) - \max\left\{x - \widehat{m}_K, \widehat{M}_K - x\right\}\right| \leqslant 2\epsilon.$$

### E.5 EXTENSION TO CONTINUOUS ENVIRONMENTS

Our algorithm, **Rob−ReL**, is designed for the "vanilla" tabular setting, where no structural assumption is made on the MDP nor on the target reward. In this setting, a polynomial dependence on the size of state space (see Theorem 5.3) is almost always present in RL theory is and also unavoidable (e.g., see Agarwal et al. (2021)).

Scaling **Rob−ReL** to large or continuous state spaces while preserving theoretical guarantees is possible as long as additional structural assumptions on the dynamics of the MDP and on the target reward are introduced, and if some changes are made to the algorithm. To make an example, assume that the MDP has an infinite/continuous state space and that it is a Linear MDP (Jin et al., 2020b), i.e., the transition model $p$ and the reward $r^\star$ are linear in some given $d$-dimensional feature map $\phi : \mathcal{S} \times \mathcal{A} \to [-1, +1]^d$. Then, robust ReL can be performed with a variant of **Rob−ReL** that:

- Replaces the occupancy measures with the feature expectations (Arora & Doshi, 2021);

- Replaces RF-Express with any reward-free algorithm for linear MDPs (e.g., Wagenmaker et al. (2022));

- Applies the primal-dual subgradient method in the $d$-dimensional space of reward parameters.

Crucially, we believe that, through a proof analogous to that of our Theorem 5.3, it should be possible to derive sample and time complexity bounds that replace the dependence on the size of the state space $S$ with a dependence on the feature dimension $d$.

If instead one wants to scale `Rob-ReL` without preserving the theoretical guarantees, then we do not even need a reward-free exploration subroutine nor the structure of Linear MDPs, but assuming access to an online planner subroutine and assuming the target reward is linear in some known $\phi$ suffice. Indeed, in such a case, we could work with feature expectations instead of occupancy measures, and the online planner would allow us to estimate the feature expectations of the optimal policy in the demonstration feedback.

### E.6 OTHER ReL PROBLEMS

The algorithm presented in Section 5 can be straightforwardly extended, along with its theoretical guarantees, to consider other kinds of feedback and applications that preserve the convexity of the problem.

For instance, if we replace the application $g$ with that of assessing a preference between two trajectories (see Table 2), then it is clear that the scheme of the algorithm does not change, but we just have to modify the computation of the subgradients.

As another example, we might consider demonstrations from "bad" policies $\pi$, i.e., demonstrations from policies whose performance is almost the worst possible:

$$J^\pi(r^\star; p) \leqslant \min_{\pi'} J^{\pi'}(r^\star; p) + t,$$

for some $t > 0$. Note that the reward-free exploration algorithm of Menard et al. (2021) works also in this setting, and its theoretical guarantees can be easily adapted to this setting.

We mention that we could also consider as feedback a *fractional* comparison of policies $\pi^1, \pi^2$ (or trajectories $\omega^1, \omega^2$):

$$J^{\pi^1}(r^\star; p) \geqslant \alpha J^{\pi^2}(r^\star; p),$$

for some $\alpha \in (0, 1]$.

## F EXPERIMENTAL DETAILS AND ADDITIONAL SIMULATIONS

We provide here additional simulations (Appendix F.2) and details on the simulation in Section 6 (Appendix F.1). Note that all the experiments have been conducted on a server with processor "Intel(R) Xeon(R) Gold 6418H", and required dozens of minutes for execution.

### F.1 ILLUSTRATIVE SIMULATION

We describe here precisely the illustrative simulation reported in Section 6.

#### F.1.1 TARGET ENVIRONMENT, TARGET REWARD $r^\star$, APPLICATION $g$

For the application $g$, we considered the environment reported in Figure 3 (left) and described in Section 5, with initial state the C lane, and the stationary transition model $p$ described below. Note that $p$ depends only on the lane and the action played, thus, with abuse of notation, we write $p$ as:

$$p_h(\cdot | L, a_L) = \begin{cases} 1, \text{if } \cdot = L \\ 0, \text{if } \cdot = C \\ 0, \text{if } \cdot = R \end{cases} \quad p_h(\cdot | C, a_L) = \begin{cases} 0.6, \text{if } \cdot = L \\ 0.4, \text{if } \cdot = C \\ 0, \text{if } \cdot = R \end{cases} \quad p_h(\cdot | R, a_L) = \begin{cases} 0, \text{if } \cdot = L \\ 0.6, \text{if } \cdot = C \\ 0.4, \text{if } \cdot = R \end{cases}$$

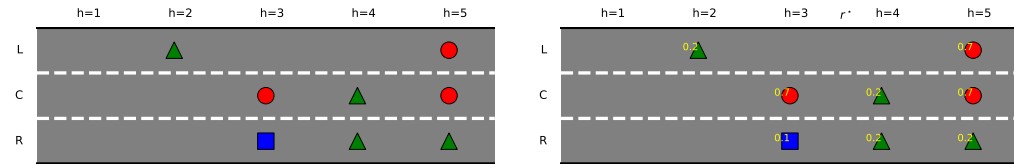

Figure 3: (Left) The target environment considered in the experiment. (Right) Representation of $r^\star$ for the target environment.

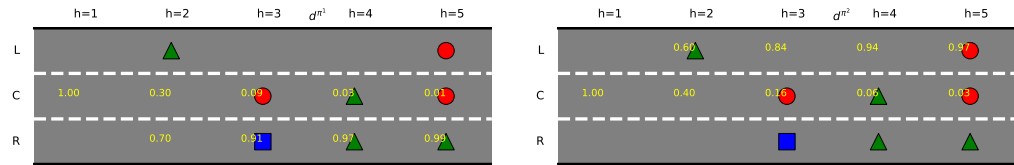

Figure 4: (Left) Plot of $d^{\pi^1}$. (Right) Plot of $d^{\pi^2}$.

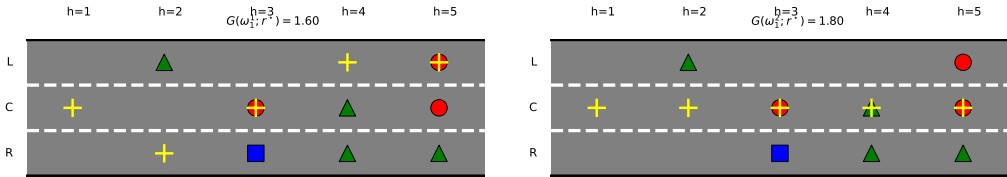

Figure 5: The trajectories compared in the first feedback. $\omega_1^1$ is on the left, and $\omega_1^2$ on the right.

$$p_h(\cdot|L, a_C) = \begin{cases} 0.55, \text{if } \cdot = L \\ 0.45, \text{if } \cdot = C \\ 0, \text{if } \cdot = R \end{cases} \quad p_h(\cdot|C, a_C) = \begin{cases} 0.3, \text{if } \cdot = L \\ 0.4, \text{if } \cdot = C \\ 0.3, \text{if } \cdot = R \end{cases} \quad p_h(\cdot|R, a_C) = \begin{cases} 0, \text{if } \cdot = L \\ 0.45, \text{if } \cdot = C \\ 0.55, \text{if } \cdot = R \end{cases}$$

$$p_h(\cdot|L, a_R) = \begin{cases} 0.3, \text{if } \cdot = L \\ 0.7, \text{if } \cdot = C \\ 0, \text{if } \cdot = R \end{cases} \quad p_h(\cdot|C, a_R) = \begin{cases} 0, \text{if } \cdot = L \\ 0.3, \text{if } \cdot = C \\ 0.7, \text{if } \cdot = R \end{cases} \quad p_h(\cdot|R, a_R) = \begin{cases} 0, \text{if } \cdot = L \\ 0, \text{if } \cdot = C \\ 1, \text{if } \cdot = R \end{cases}$$

Intuitively, action $a_L$ moves to the left w.p. $0.6$, and keeps the lane w.p. $0.4$, except when it is on the left lane, where it keeps the lane. Action $a_R$ is analogous but with a different bias. Instead, action $a_C$ keeps the lane w.p. $0.4$, and moves to the left or to the right w.p. $0.3$. When it is on the borders, it cannot move in a certain direction, thus the remaining probability is splitted equally in the other two lanes. The target reward $r^\star$ considered is described in Section 5, and is shown in Figure 3, on the right. Finally, the occupancy measures $d^{\pi^1}, d^{\pi^2}$ describing the application $g$ arise from the policies described in Section 5 and the transition model $p$ presented earlier, and are shown in Figure 4.

### F.1.2 FEEDBACK $\mathcal{F}$

**Trajectory comparisons.** We construct three trajectory comparison feedback. The first pair is in Figure 5, and we associated $t_1 = 0.3$ to it. The second pair of trajectories is in Figure 6, and we set $t_2 = 1$. Finally, the third pair is in Figure 7 and has $t_3 = -0.5$.

**Comparisons.** Concerning the comparisons feedback, we considered the new environment shown in Figure 8 on the left, keeping the same transition model $p$ described earlier but using the left $L$ lane as initial state. We compared the two occupancy measures in Figure 9 using $t_1 = 0$, and the two occupancy measures in Figure 10 using $t_2 = 0.5$.

**Demonstrations.** For the demonstrations feedback, we adopted the map in Figure 8 on the right, preserving the transition model $p$, but using lane R as initial state. We considered only one feedback, whose policy has the occupancy measure in Figure 11, to which we associated $t_1 = 1$.

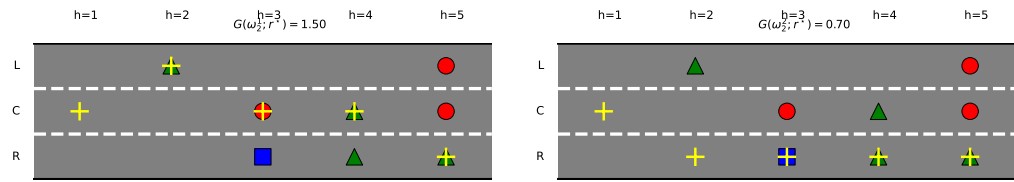

Figure 6: The trajectories compared in the second feedback. $\omega_2^1$ is on the left, and $\omega_2^2$ on the right.

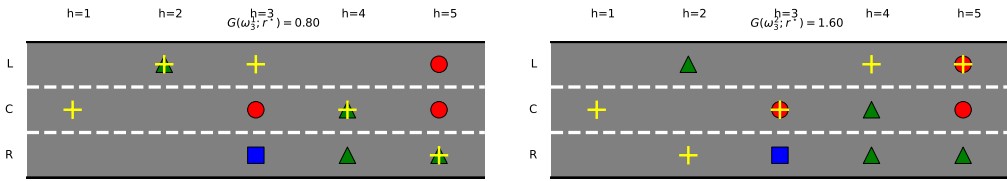

Figure 7: The trajectories compared in the third feedback. $\omega_3^1$ is on the left, and $\omega_3^2$ on the right.

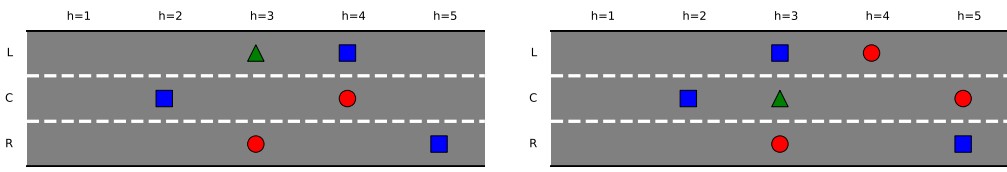

Figure 8: (Left) The new map considered for the comparisons feedback. (Right) The new map considered for the demonstrations feedback.

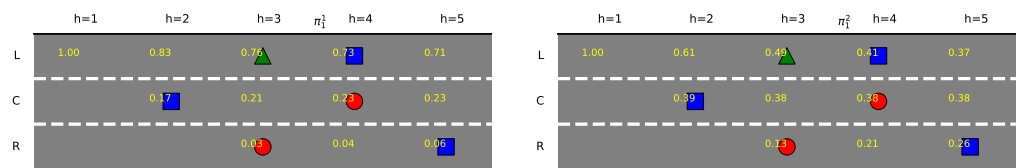

Figure 9: The occupancy measures compared in the first comparisons feedback. $\pi_1^1$ is on the left, and $\pi_1^2$ on the right.

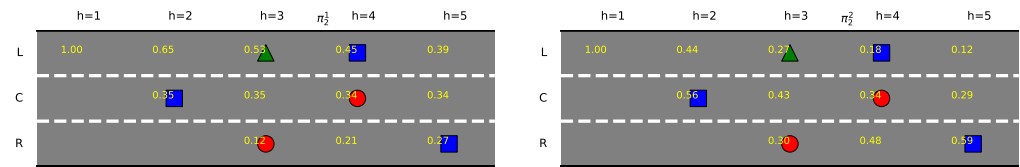

Figure 10: The occupancy measures compared in the second comparisons feedback. $\pi_2^1$ is on the left, and $\pi_2^2$ on the right.

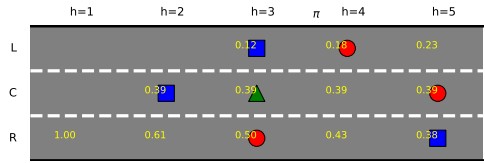

Figure 11: The occupancy measure of the expert's policy for the demonstrations feedback.

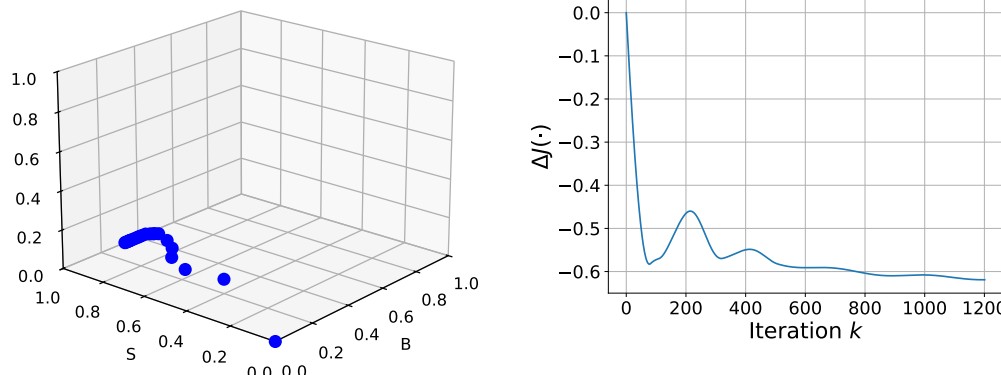

Figure 12: (Left) The sequence of rewards $\hat{r}_{m,k}$ computed by our algorithm. (Right) The corresponding values of the objective function.

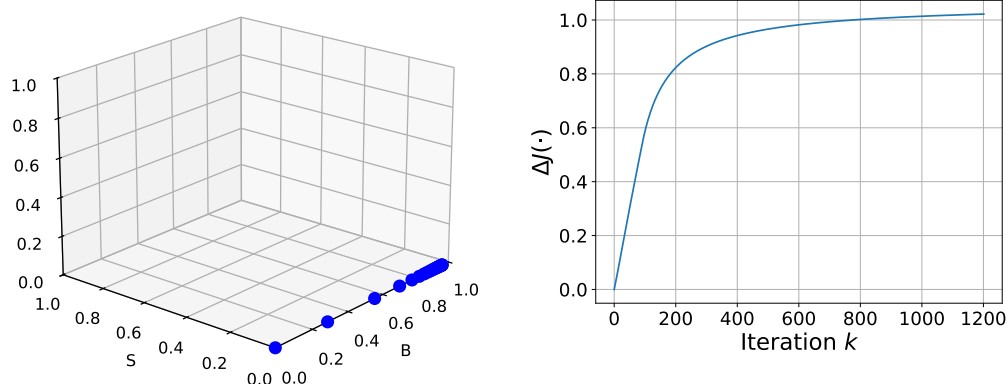

Figure 13: (Left) The sequence of rewards $\hat{r}_{M,k}$ computed by our algorithm. (Right) The corresponding values of the objective function.

### F.1.3    SIMULATION

The execution of **Rob-ReL** generated the sequence of reward functions $\hat{r}_{m,k}$ for finding $m$ in Figure 12 on the left, while on the right we plotted the corresponding value of the objective function $\Delta J(\hat{r}_{m,k})$.

The analogous plots for $M$ are in Figure 13.

### F.1.4    VALIDATION THROUGH DISCRETIZATION

To understand if the values of $\hat{r}_M, \hat{r}_m, \hat{r}_{\mathcal{F},g}, \widehat{M}_K, \hat{m}_k$ computed by the execution of **Rob-ReL** (see Figure 1, right) make sense, i.e., are close to the true values $r_M, r_m, r_{\mathcal{F},g}$, we have computed them also through an "exact" method, by computing a discretization of the feasible set and then taken the rewards that maximize/minimize the objective function (for approximating $r_M, r_m$), and then averaged them. The results of this computation are reported in Figure 14. Clearly, these values are close to those in Figure 1, thus the computation carried out by **Rob-ReL** makes sense.

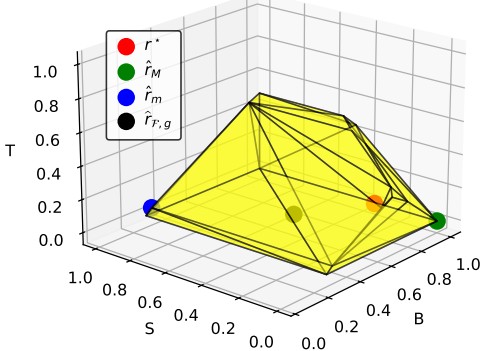

Figure 14: The rewards $r_M, r_m, r_{\mathcal{F},g}$ computed through a discretization of the feasible set.

|   | $S, A, H$ | $m_D, m_{PC}, m_{TC}$ | $n, N$ | err $x$ | err $\mathcal{I}$ |
|---|---|---|---|---|---|
| 1 | 3, 3, 5 | 1, 2, 2 | 50, 100 | $0.07 \pm 0.04$ | $0.09 \pm 0.05$ |
| 2 | 3, 3, 5 | 1, 2, 2 | 500, 1000 | $0.02 \pm 0.01$ | $0.02 \pm 0.02$ |
| 3 | 3, 3, 5 | 5, 15, 15 | 50, 100 | $0.07 \pm 0.04$ | $0.08 \pm 0.06$ |
| 4 | 3, 3, 5 | 5, 15, 15 | 500, 1000 | $0.02 \pm 0.01$ | $0.03 \pm 0.02$ |
| 5 | 100, 10, 15 | 1, 2, 2 | 50, 100 | $0.13 \pm 0.07$ | $0.33 \pm 0.19$ |
| 6 | 100, 10, 15 | 1, 2, 2 | 500, 1000 | $0.04 \pm 0.03$ | $0.06 \pm 0.05$ |
| 7 | 100, 10, 15 | 5, 15, 15 | 50, 100 | $0.14 \pm 0.12$ | $0.17 \pm 0.15$ |
| 8 | 100, 10, 15 | 5, 15, 15 | 500, 1000 | $0.04 \pm 0.03$ | $0.06 \pm 0.07$ |

Table 3: Results of experiment $(i)$.

## F.2 ADDITIONAL SIMULATIONS

The simulation presented in Section 6 is illustrative, i.e., it aims to clarify the importance of our robust approach. Here, we conduct additional simulations to better characterize how `Rob-ReL` scales to larger problems.

To this aim, we generated at random a large and various number of ReL problems analogous to that in Section 6, i.e., where the state space can be interpreted as a road with some objects, the actions allow to change lane (with some noise), and the reward is stationary and state-only, and in particular depends only on the object in the considered state. We generated two kinds of these problems, one $(i)$ where the number of objects is small (specifically, 4), and so it is possible to compute $x_{\mathcal{F},g}$ by simply discretizing the space of rewards and keeping the one that optimizes the minimax objective, and another $(ii)$ where the number of objects is large (20 and 50), and so the optimal solution cannot be approximated through discretization. Both simulations have been conducted with values of hyperparameters $\alpha = 0.01$ (the step size), $K = 3000$ (number of iterations), and $s = 1000$ (the bound on the Lagrange multipliers).

Regarding $(i)$, we conducted 8 simulations with the values specified in the first three columns of Table 3, where $n$ denotes the number of trajectories provided by any comparisons feedback, and $N$ the number of exploration episodes allowed by any demonstrations feedback. Each simulation consists of generating 20 different problem instances with the specified features (size $S, A, H$, number of feedback $m_D, m_{PC}, m_{TC}$, and number data $n, N$), for each problem instance "finding" the optimum $x_{\mathcal{F},g}, \mathcal{I}_{\mathcal{F},g}$ using the discretization approach, computing the outputs $\widehat{x}_K, \widehat{\mathcal{I}}_K$ of `Rob-ReL`, and then compute the mean absolute errors between $x_{\mathcal{F},g}$ and $\widehat{x}_K$, and between $\mathcal{I}_{\mathcal{F},g}$ and $\widehat{\mathcal{I}}_K$. The results are reported in the last two columns of Table 3. Clearly, the error reduces as the number of samples increases. Moreover, larger problem instances require more data as expected also by theory (Theorem 5.3). We mention that as $N$ increases the time required for running RF-Express increases not trivially, as its execution cannot be vectorized (Menard et al., 2021).

Regarding $(ii)$, we conducted 4 simulations all with $S, A, H = 20, 5, 12$ and $m_D, m_{PC}, m_{TC} = 3, 7, 7$, where we changed the number of objects and the number of data (see Table 4). As aforementioned, here we cannot assess performance using the solution outputted through discretization. To this aim, for each problem instance, we compute $\Delta J(r^\star)$ and compare it with the output $\widehat{x}_K$ of `Rob-ReL` to

| | n objects | $n, N$ | err $x$ |
|---|---|---|---|
| 1 | 20 | 50, 100 | $0.11 \pm 0.07$ |
| 2 | 20 | 500, 1000 | $0.08 \pm 0.05$ |
| 3 | 50 | 50, 100 | $0.07 \pm 0.05$ |
| 4 | 50 | 500, 1000 | $0.11 \pm 0.08$ |

Table 4: Results of experiment $(ii)$.

have an idea of the error. Results are in the last column of Table 4. We mention that some error cannot be avoided because $\widehat{x}_K$ does not directly estimate $\Delta J(r^\star)$, and also that increasing the number of objects from 4 to 20 or 50 basically reduces to performing gradient descent to a 20(50)-dimensional space, which is still very efficient.

