# OpenReview forum: "Robustness in the Face of Partial Identifiability in Reward Learning"
_ICLR.cc/2026/Conference — ICLR 2026 Poster_

### Official Review · Reviewer_Kgr5 · 2025-10-28

**Soundness:** 3
**Presentation:** 3
**Contribution:** 2
**Rating:** 4
**Confidence:** 4

**Summary:**

The paper proposes a unified framework for robust reward learning in partially identifiable settings. Given some form of feedback (e.g., preferences, demonstrations), the method constructs a feasible set of reward functions consistent with observed data and then optimizes the worst case over this set (for a downstream decision such as a policy). The formulation aims to generalize across multiple feedback modalities and tasks.

**Strengths:**

+ The framework unifies various feedback types (e.g., pairwise comparisons, demonstrations) and downstream objectives under one minmax formalism.
+ The worst-case formulation in (7) is intuitive and could be more realistic than assuming a point estimate of the reward.

**Weaknesses:**

- The general minmax structure resembles many existing approaches in IRL, robust MDPs, and RLHF (e.g., distributionally robust RL, reward uncertainty sets). The paper does not sufficiently delineate what is new in (2) beyond framing and unification.

- The paper needs sharper theoretical distinctions (e.g., identifiability analysis, sample complexity bounds, or formal generalization of previous frameworks). Without this the contribution risks being incremental.

- The approach requires estimating or approximating policy-induced dynamics and state-action visitation measures, which can be computationally prohibitive and may not be scalable to realistic domains.

- The proposed method replaces one set of unknowns (the reward) with multiple estimated quantities (transition probabilities, occupancy measures, feasible reward bounds), which may not improve robustness in practice.

- The experiments are limited to small toy domains (even with the ones in the appendix). The paper needs results on more standard RL or preference-learning benchmarks (e.g., MuJoCo or D4RL tasks) to demonstrate practical value. The lack of scalability experiments weakens the claimed generality.

- It is unclear whether ablation or sensitivity studies were conducted to assess dependence on feasible set or uncertainty size.

- The paper does not discuss how many trajectories or feedback samples are required to characterize the feasible reward set. It remains also unclear how the method handles poor coverage or unobserved states which is an important concern when identifiability is partial.

- The framework implicitly assumes access to accurate simulators or transition estimates, which limits realism in practical RLHF settings.

**Questions:**

1) How does (2) differ theoretically or algorithmically from robust IRL or minimax RLHF formulations?
2) How sensitive is performance to the feasible reward set?
3) Can the approach handle partially observed dynamics or limited coverage?
4) What is the empirical complexity compared to baselines? how does it scale with state or trajectory length?

---

> ### Author Response · Authors · 2025-11-17
>
> We thank the Reviewer for recognizing the relevance of our unifying framework as well as the strong intuition behind our robust approach. Below, we respond to the Reviewer's comments and questions.
>
> > The general minmax structure resembles many existing approaches in IRL, robust MDPs, and RLHF (e.g., distributionally robust RL, reward uncertainty sets). The paper does not sufficiently delineate what is new in (2) beyond framing and unification.
>
> **Regarding IRL and RLHF**: We refer the Reviewer to the paragraph "Existing approaches" in Section 4 and to Appendix A, where we describe in detail the differences and importance of our novel robust approach w.r.t. existing approaches.
>
> In brief, our idea of solving a minimax to reduce the worst-case error due to partial identifiability is *novel* in ReL. Indeed, as discussed in Appendix A, there are two kinds of ReL papers:
>
> - Papers that "ignore" partial identifiability: their algorithms make an arbitrary choice of reward from the feasible set, which might suffer from a large error in the worst case;
> - Papers that "take into account" partial identifiability: they look for conditions under which any reward in the feasible set is "fine", and so the arbitrary reward choice of the previous algorithms has *zero* worst-case error (see also Appendix A.2).
>
> Our insight is simply that, by making a smart (robust) choice of reward, we can improve performance without requiring additional conditions.
>
> As far as we know, there is no IRL or RLHF algorithm in the literature that employs a minimax structure *to address the partial identifiability problem*, but just for *other* purposes. The only exception is represented by  IL algorithms like [1,2,3] (see Lines 233-235 and Appendix A.1), in which, however, the feedback is a simple assumption on the class of functions to which the target reward belong, and does not require any estimation. For instance, [1] and [2] respectively assume that $r^\star$ is linear and convex in a known feature map.
>
> **Regarding Robust MDPs**: Reward-Robust MDPs (e.g., see [4]) are closely related to our robust ReL formulation, but there are some crucial differences.
>
> In particular, in both Reward-Robust MDPs and our Eq. (2), the goal is to be robust against the missing knowledge of the true reward. In both settings the reward is known to belong to a certain set of rewards, which is called feasible set in our ReL setting (and in the IRL and ReL literature), while it is called uncertainty set in the context of Robust MDPs.
>
> However, there are three crucial differences:
>
> - First, in robust MDPs, there is a single application, i.e., finding a good policy under the unknown reward, while in our ReL setting there can be a variety of different applications (see Table 2).
> - Second, in robust MDPs the uncertainty set is *given and known*, while in ReL the feasible set must be *estimated* from finite data (e.g., from demonstrations or trajectory/policy comparisons).
> - Third, in the literature, the uncertainty set in Robust MDPs is almost always rectangular, while in ReL the shape of the feasible set depends on the feedback and can be different and more complex.
>
> We have updated the submission to take these considerations on Robust MDPs into account in the Additional Related Work section.

---

> > ### Author Response · Authors · 2025-11-17
> >
> > > The paper needs sharper theoretical distinctions (e.g., identifiability analysis, sample complexity bounds, or formal generalization of previous frameworks). Without this the contribution risks being incremental.
> >
> > Regarding the three points mentioned by the Reviewer, our contributions significantly improve upon the state of the art:
> >
> > - **Identifiability analysis**: We are the first to propose a *robust* approach to address the partial identifiability problem in ReL, as explained in Section 4. Indeed, this is novel, as existing ReL works either extract an *arbitrary* reward from the feasible set, or impose additional conditions/assumptions to guarantee that the feasible set contains a single (or just a few) rewards, but none of them try to be robust w.r.t. the rewards in the feasible set.
> > - **Sample complexity bounds**: We present a new algorithm (Rob-ReL) together with an original sample complexity analysis showing that the ReL problem of assessing a policy preference from multiple types of feedback (demonstration, trajectory comparisons and policy comparisons), can be solved with the *polynomial* number of samples and iterations stated in Theorem 5.3. These results rely on an interesting combination of the theoretical guarantees for reward-free exploration [10] with those for the PDSM [11] (see the proof sketch in Section 5), and provide a useful reference for practitioners studying their own ReL problems.
> > - **Generalization of previous frameworks**: Our framework is the first ReL framework that accounts for partial identifiability in a *quantitative* way. As explained in Appendix A.2, it shares similarities with the framework of [5] in that both include feedback and applications (compared to, e.g., [9], which includes only feedback). *However*, crucially, [5] models applications only in a *qualitative* manner, resulting in a binary output on whether partial identifiability is or is not problematic in the ReL problem at stake. Instead, our more expressive framework permits quantifying the error due to partial identifiability, consequently allowing us to discriminate between cases in which the error can be tolerated and those in which it cannot (remarkably, in the framework of [5], all cases with error strictly larger than zeros are problematic). Furthermore, our framework is more general than that of [5], as we consider additional feedback (e.g., policy comparisons) and applications (e.g., constrained planning).
> >
> > > The approach requires estimating or approximating policy-induced dynamics and state-action visitation measures, which can be computationally prohibitive and may not be scalable to realistic domains.
> >
> > We remark that our algorithm, Rob-ReL, has been designed for tabular MDPs. If one wants to tackle problems with large or continuous state-action spaces, then the algorithm must be, of course, modified depending on the structure of the problem.
> >
> > For instance, let us consider problems in which the target reward is assumed to be linear in some known feature map $\phi$ (which is standard in IRL, see [6]). In this setting, all the occupancy measures in Eq. (7) can be replaced by the corresponding feature expectations (see Section 2 in [6]), which can be estimated and computed efficiently. Moreover, we can avoid estimating the dynamics $p$ by performing, at each iteration of the PDSM loop, rollouts in the environment to compute the feature expectations associated with the optimal policy in the demonstrations feedback.
> >
> > This represents a very simple but powerful example of how to extend our algorithm to problems with continuous state-action spaces with a specific structure on the target reward. Depending on the structure, different extensions can be devised, and we leave it to practitioners to design the most appropriate extension depending on $(a)$ the specific ReL problem at stake, and $(b)$ the specific structure of the reward/dynamics.
> >
> > > The proposed method replaces one set of unknowns (the reward) with multiple estimated quantities (transition probabilities, occupancy measures, feasible reward bounds), which may not improve robustness in practice.
> >
> > We emphasize that:
> >
> > - Theorem 5.3 *demonstrates* that our method improves robustness, as it provably outputs solutions with minimum worst-case error using only a polynomial number of samples.
> > - Any non-robust ReL algorithm needs to estimate all such quantities, either explicitly or implicitly, to obtain a reward in the feasible set (and thus needs a polynomial number of samples comparable to that of Rob-ReL). However, being non-robust, its output generally has a worst-case error larger than that of Rob-ReL.
> > - Even in standard RL, where the reward function is known, the transition probabilities (and the occupancy measures of policies) remain unknown and must be estimated, either explicitly or implicitly.

---

> > > ### Author Response · Authors · 2025-11-17
> > >
> > > > The experiments are limited to small toy domains (even with the ones in the appendix). The paper needs results on more standard RL or preference-learning benchmarks (e.g., MuJoCo or D4RL tasks) to demonstrate practical value. The lack of scalability experiments weakens the claimed generality.
> > >
> > > We respectfully disagree, because the generality of our framework is completely *independent* of the "lack of scalability experiments". Our framework addresses any ReL problem, and shows that our robust approach *provably* achieves the best possible worst-case performance by definition, as explained in Section 4.
> > >
> > > > It is unclear whether ablation or sensitivity studies were conducted to assess dependence on feasible set or uncertainty size.
> > >
> > > We do not perform any ablation or sensitivity studies. As mentioned above, we emphasize that an extensive empirical validation of the proposed algorithm is beyond the scope of this work, whose main goal is to introduce the new framework, the robust method and to show that it can be applied in a *provably efficient* way.
> > >
> > > From a theoretical perspective, we note that if the size of the feasible set is measured in terms of the number of feedback, then Theorem 5.3 shows that our algorithm exhibits a logarithmic dependence in the sample complexity (hidden by the $\widetilde{\mathcal{O}}$ notation, see Lemma E.1 in Appendix E.3.1), and a linear dependence in the iteration complexity.
> > >
> > > > The paper does not discuss how many trajectories or feedback samples are required to characterize the feasible reward set. It remains also unclear how the method handles poor coverage or unobserved states which is an important concern when identifiability is partial.
> > >
> > > We would like to note that the goal of our algorithm is to solve a ReL problem, and *not* to characterize the feasible reward set. That task belongs to a closely related but different line of research (we refer the reader to Appendix A, paragraph "Works that estimate the feasible set", for a detailed discussion of those related works).
> > >
> > > Regarding the second point, we note that coverage issues generally occur when using offline data to estimate the dynamics of the MDP in states not covered by the data. This issue does arise here because our offline data are used for different purposes (i.e., estimating the occupancy measure of the generating policy), while we use online data to estimate the transition model. If one wanted to extend our algorithm to a problem setting where only offline data collected by a behavior policy are available for estimating the transition model, then this issue would arise, and a possible mitigation might be to adopt a pessimistic approach (e.g., see [7], where pessimism is applied in terms of the feasible set).
> > >
> > > > The framework implicitly assumes access to accurate simulators or transition estimates, which limits realism in practical RLHF settings.
> > >
> > > We remark that common RLHF settings are much easier than our setting, in that no demonstration feedback is present in standard RLHF problems, and thus there is no need for simulators to estimate the transition model. In particular, note that:
> > >
> > > - If we do not have demonstration feedback, i.e., $m_{\text{D}}=0$, then we do no need access to a simulator (specifically, we can avoid running Line 2 of Algorithm 1).
> > > - With demonstration feedback, i.e., $m_{\text{D}}>0$, our setting is a generalization of the common IRL setting, where access to simulators or transition estimates is *necessary* (e.g., see [6,8]).
> > >
> > > > How does (2) differ theoretically or algorithmically from robust IRL or minimax RLHF formulations?
> > >
> > > Please refer to the comment above.

---

> > > > ### Author Response · Authors · 2025-11-17
> > > >
> > > > > How sensitive is performance to the feasible reward set?
> > > >
> > > > As aforementioned, the sample complexity of Rob-ReL scales logarithmically with the number of constraints defining the feasible set, while its iteration complexity scales linearly. See Theorem 5.3 in Section 5.
> > > >
> > > > > Can the approach handle partially observed dynamics or limited coverage?
> > > >
> > > > Our focus is on MDPs, and not partially observable MDPs (POMDPs). We believe that our robust approach can be extended to POMDPs to improve worst-case performance in those scenarios as well. However, this is not immediate, as the setting is quite different.
> > > >
> > > > Regarding the limited coverage, as mentioned also in a previous comment, we do not suffer from this issue as our offline data are used for estimating the occupancy measures, while we have access to online data for estimating the dynamics (as in most IRL works, e.g., see [8]).
> > > >
> > > > > What is the empirical complexity compared to baselines? how does it scale with state or trajectory length?
> > > >
> > > > Note that there is no baseline in the literature for addressing the ReL problem solved by Rob-ReL. Indeed, Rob-ReL is novel in two aspects:
> > > >
> > > > 1. It addresses a novel application, i.e., that of "assessing a policy preference" (see Table 2);
> > > > 2. It considers a novel mix of feedback types (demonstrations, trajectory comparisons, policy comparisons).
> > > >
> > > > The majority of existing ReL algorithms in the literature consider multiple feedback of the same type, while none of them consider this application. As such, there is no baseline in the literature that can be meaningfully applied to the ReL problem addressed by Rob-ReL.
> > > >
> > > > We would like to emphasize that, from a theoretical perspective, Rob-ReL is *provably* efficient, and its sample complexity in Theorem 5.3 is comparable to that of existing theoretical works on estimating the feasible set (see Appendix A), modulo an additional $H^2/\xi^2$ due to the optimization problem. Appendix F.2 presents numerical simulations exemplifying this complexity.

---

> > > > > ### Author Response · Authors · 2025-11-17
> > > > >
> > > > > [1] Pieter Abbeel and Andrew Y. Ng. Apprenticeship learning via inverse reinforcement learning. In International Conference on Machine Learning 21 (ICML), 2004.
> > > > >
> > > > > [2] Umar Syed and Robert E Schapire. A game-theoretic approach to apprenticeship learning. In Advances in Neural Information Processing System 20 (NeurIPS), 2007.
> > > > >
> > > > > [3] Jonathan Ho, Jayesh K. Gupta, and Stefano Ermon. Model-free imitation learning with policy optimization, 2016.
> > > > >
> > > > > [4] Gadot, U., Derman, E., Kumar, N., Elfatihi, M.M., Levy, K.Y., & Mannor, S. . Solving Non-Rectangular Reward-Robust MDPs via Frequency Regularization, 2023.
> > > > >
> > > > > [5] Joar Skalse, Matthew Farrugia-Roberts, Stuart Russell, Alessandro Abate, and Adam Gleave. Invariance in policy optimisation and partial identifiability in reward learning. In International Conference on Machine Learning 40 (ICML), 2023.
> > > > >
> > > > > [6] Arora, Saurabh and Prashant Doshi. A Survey of Inverse Reinforcement Learning: Challenges, Methods and Progress. Artif. Intell., 2018.
> > > > >
> > > > > [7] Filippo Lazzati, Mirco Mutti, and Alberto Maria Metelli. Offline inverse rl: New solution concepts and provably efficient algorithms. In International Conference on Machine Learning 41 (ICML), 2024.
> > > > >
> > > > > [8] Justin Fu, Katie Luo, and Sergey Levine. Learning robust rewards with adversarial inverse reinforcement learning. In International Conference on Learning Representations 5 (ICLR), 2017.
> > > > >
> > > > > [9] Hong Jun Jeon, Smitha Milli, and Anca Dragan. Reward-rational (implicit) choice: A unifying formalism for reward learning. In Advances in Neural Information Processing Systems 33 (NeurIPS), 2020.
> > > > >
> > > > > [10] Pierre Menard, Omar Darwiche Domingues, Anders Jonsson, Emilie Kaufmann, Edouard Leurent, and Michal Valko. Fast active learning for pure exploration in reinforcement learning. In International Conference on Machine Learning 38 (ICML), 2021.
> > > > >
> > > > > [11] A. Nedic and A. Ozdaglar. Subgradient methods for saddle-point problems. Journal of Optimization Theory and Applications, 2009.

---

> ### Comment · Reviewer_Kgr5 · 2025-11-28
> **Post-rebuttal response**
>
> I thank the authors for their detailed responses. The clarifications around Theorem 5.3 and the role of online vs offline data helped me better understand the intended scope. However, several of my main concerns remain largely unaddressed:
>
> - **Experiments:** The authors argue that the generality of their framework is independent of empirical scalability and that extensive experiments are essentially not needed. From an ICLR methods perspective, I still view the small-scale toy domains as insufficient to support the claimed practical generality. Some evidence on more standard RL  and preference learning benchmarks, or at least stronger ablations, would be important to demonstrate that the proposed formulation can be implemented and scaled in non-trivial settings.
> - **Ablations:** The authors explicitly state that no ablation or sensitivity study is performed. Given that the behavior of the robust solution critically depends on the feasible set and uncertainty geometry, this remains a significant gap.
> - **Novelty relative to robust MDPs and IRL.** The response mostly emphasizes contextual differences (multiple downstream tasks, feedback derived sets), but I still do not see a fundamentally new technical machinery beyond applying a standard minimax robust formulation once a feasible reward set is given.
>
> In light of this, my overall evaluation is largely unchanged.

---

> > ### Author Response · Authors · 2025-11-30
> >
> > We thank the Reviewer for engaging in a discussion with us. We feel the necessity to provide additional clarifications on the concerns raised by the Reviewer, because we disagree that:
> >
> > - Experiments and ablations on *large-scale benchmarks* are necessary to introduce a novel theoretically-grounded algorithm designed explicitly for tabular MDPs.
> > - A *new technical machinery* is needed to introduce a unifying conceptual framework.
> >
> > In particular, regarding the experiments concern, we would like to remark that, as already mentioned in the rebuttal, (i) the generality of the framework is independent of any numerical simulation that can be run, because it is a conceptual framework unifying a large variety of ReL problems (notice that we never claim a "practical generality" that the Reviewer mentions); (ii) our algorithm, being the first *robust ReL algorithm*, is explicitly designed to address tabular MDPs, and as such it cannot be directly applied to large-scale benchmarks, which require function approximation to be tackled. Although we sketched in the rebuttal a possible way for scaling our algorithm (introducing necessary additional assumptions), we believe that conducting an extensive empirical study of a *variant* of the algorithm that we studied in the paper is out of scope for this paper. Note that our paper contains various simulations (see Section 6 and Appendix F) for the specific algorithms studied, as it is the standard practice in RL theory (e.g., see [12]).
> >
> > Regarding the absence of ablation or sensitivity studies, as mentioned above, we emphasize that an extensive empirical validation of the proposed algorithm is beyond the scope of this work, whose main goal is to introduce the new framework, the robust method and to show that it can be applied in a *provably efficient* way. From a theoretical perspective, we note that if the size of the feasible set is measured in terms of the number of feedback, then Theorem 5.3 shows that our algorithm exhibits a logarithmic dependence in the sample complexity (hidden by the $\widetilde{\mathcal{O}}$ notation, see Lemma E.1 in Appendix E.3.1), and a linear dependence in the iteration complexity.
> >
> > Regarding the technical novelty concern, we would like to emphasize that (i) technical novelty is not needed for the introduction of a conceptual framework, whose goal is to unify different problems under the same umbrella, and propose a general solution approach for all of them. Note that also previous ReL frameworks [5,9] are *not* accompanied by any new technical machinery, but their core contribution lies in the conceptual unification of different problems based on the proposed model. (ii) We do provide technical novelty in the design and analysis of the algorithm, because, as already mentioned in the rebuttal, we originally combine the theoretical guarantees of the PDSM [11] with those of reward-free exploration [10], resulting in the polynomial sample and iteration complexity guarantees in Theorem 5.3.
> >
> >
> >
> > [12] Alekh Agarwal, Nan Jiang, Sham M. Kakade, Wen Sun. Reinforcement Learning: Theory and Algorithms. 2021.

---

### Official Review · Reviewer_kR8k · 2025-10-29

**Soundness:** 2
**Presentation:** 2
**Contribution:** 2
**Rating:** 2
**Confidence:** 2

**Summary:**

This paper introduces a framework for reward learning from preferences and demonstrations. The authors try to quantify the effect of  identifiability issues of the reward, and propose a minimax approach to optimize the worst-case scenario over the feasible reward set. They provide Rob-ReL, an algorithm for policy preference assessment, providing theoretical guarantees on sample and computation complexity.

**Strengths:**

- Clear formalization of the problem.
- The paper is well written and organized in a clear progression
- Clear sample-complexity results for their algorithm with illustrative numerical results

**Weaknesses:**

- A big limitation of this work is the limited number of applications. The authors provide a general framework, but then they limit themselves to a single scenario. This leaves the overall framework not properly tested, and severely limits the contribution of the paper.
- Another limitation, is that while the authors provide a general framework, the theoretical insights seem to be limited. For example, when solving a minimax game the equilibrium may not be a pure equilibrium; however, the authors do not seem to discuss this issue. Depending on the properties of ${\cal X}_g$, the loss, etc...we may have different situations. In the application they propose the solution they get seems a pure one (because it's estimating a scalar), but i would expect mixed solutions to appear depending on the problem (e.g., when ${\cal X}_g=\Pi$). While the authors do an effort at quantifying the error (eq 3), it is not very clear what are the properties of this minimax problem (which depends on the set of rewards, loss, etc.).

- Theorem 5.3: the dependency on $\xi$ seems quite large
- It is not clear why the chosen application (estimating policy-value differences) is an interesting one

- While the paper is well written, there is still lot of notation and it is hard to follow the proofs.

- The method is only tested on a simple problem, while larger problems are untested.

**Questions:**

Please see the weaknesses above

---

> ### Author Response · Authors · 2025-11-17
>
> We are glad that the Reviewer appreciated the formalization of the problem and the overall presentation of the paper, as well as the clarity of our theoretical results and numerical simulations. We provide detailed replies to their questions and comments below.
>
> > A big limitation of this work is the limited number of applications. The authors provide a general framework, but then they limit themselves to a single scenario. This leaves the overall framework not properly tested, and severely limits the contribution of the paper.
>
> We invite the reviewer to consider the contribution of the paper from a different perspective. We could have provided only a single application without presenting a general framework, and this would still have been of interest. Instead, our contribution is broader: we introduce a general framework and general results, which constitute the main contribution, and only then do we focus on a specific application. In other words, the existence of a large number of ReL problems that fit into our framework and can be solved with our robust approach is a *pro* of this paper, demonstrating the power and generality of the proposed framework. Note that the heterogeneity of potential applications makes it infeasible to test the general framework across all of them. Therefore, our paper represents a reference for practitioners who can use our framework to design *robust* algorithms for their own ReL problems.
>
> > Another limitation, is that while the authors provide a general framework, the theoretical insights seem to be limited. For example, when solving a minimax game the equilibrium may not be a pure equilibrium; however, the authors do not seem to discuss this issue. Depending on the properties of $\mathcal{X}\_g$, the loss, etc...we may have different situations. In the application they propose the solution they get seems a pure one (because it's estimating a scalar), but i would expect mixed solutions to appear depending on the problem (e.g., when $\mathcal{X}\_g = \Pi$). While the authors do an effort at quantifying the error (eq 3), it is not very clear what are the properties of this minimax problem (which depends on the set of rewards, loss, etc.).
>
> We remark that:
>
> 1. We are *not* solving games in this paper, but our minimax problem is a mere *optimization problem*.
> 2. Characterizing the properties of the minimax optimization problem for every possible ReL problem is *not* the objective for this work.
>
> Regarding 1., note that our goal is to find the best object $x\in\mathcal{X}\_g$ that minimizes the loss for the worst possible reward in the feasible set, i.e., $\max\_{r\in\mathcal{R}\_{\mathcal{F}}}\mathcal{L}\_g(r,x)$. This is a mere optimization problem. We are not interested in solutions with specific properties from this set (e.g., "pure" or "mixed" solutions as suggested by the Reviewer in case $\mathcal{X}\_g$ is a probability simplex), but *any* minimizer is fine. In the literature, analogous optimization problems have been studied in the context of robust MDPs [1] (see [2] and the references therein for work on reward-robust MDPs).
>
> Regarding 2., as already mentioned when answering to the comment above, in this paper we are not interested in listing all the possible ReL problems and, for each of them, characterizing the resulting minimax optimization problem as well as the complexity of estimating its parameters, because that would be quite impossible in a single paper and also less significant. Instead, this paper aims to:
>
> - Present a *unifying framework* for ReL problems, highlighting their similar nature in terms of information available (feedback) and ultimate goal (application);
> - Introduce the *robust method* as a general approach for maximizing worst-case performance that can be applied to any ReL problem;
> - Focus on a specific family of ReL problems and *show* how the robust approach can be applied in a provably efficient manner.
>
> In this way, our paper represents a reference for practitioners who can use our framework to design robust algorithms for their own ReL problems.

---

> > ### Author Response · Authors · 2025-11-17
> >
> > > Theorem 5.3: the dependency on $\xi$ seems quite large
> >
> > We agree with the Reviewer. However, it is quite standard (and unavoidable) in the RL literature to have large dependencies on the Slater's gap (e.g., see [3,4]).
> >
> > > It is not clear why the chosen application (estimating policy-value differences) is an interesting one
> >
> > We chose the application of "assessing a policy preference" because $(i)$ it is interesting per se, $(ii)$ it is explanatory of the robust approach, and $(iii)$ it generalizes existing applications in the literature.
> >
> > Specifically:
> >
> > $(i)$ It models all settings where the goal is to imitate or predict expert behavior in a new environment, but where only two (or a few) candidate policies are available. In such cases, by robustly comparing the expected returns of the available policies, one can easily choose which policy to deploy in the new environment and quantify its worst-case suboptimality. Concretely, imagine a setting similar to that described in Section 6, where our learning agent can either follow a policy that always goes right or one that always goes left, and it has to make its decision based on the feedback available.
> >
> > $(ii)$ By requiring the computation of a scalar value $x\in\mathcal{X}_g=\mathbb{R}$, it offers a clear illustration of our robust approach and its relevance. Intuitively, each reward in the feasible set corresponds to a number, and the "central" value is the most robust choice, as shown on the right side of Fig. 1. With more complex sets $\mathcal{X}_g$, this graphical intuition would be lost.
> >
> > $(iii)$ It generalizes the application of "assessing a trajectory preference" (see Appendix E.5), a problem that has been considered interesting since the early days of IRL (see the "Driver Route Modeling" section in [5]).
> >
> > > The method is only tested on a simple problem, while larger problems are untested.
> >
> > We would like to emphasize that in Appendix F.2 there are additional simulations on problems with larger state-action spaces, horizon and number of feedback (e.g., $S=100,A=10,H=15$, and $>30$ feedback). This appendix is referenced from Section 6.

---

> > > ### Author Response · Authors · 2025-11-17
> > >
> > > [1] Wiesemann, W., Kuhn, D., & Rustem, B. (2013). Robust Markov Decision Processes. Math. Oper. Res.
> > >
> > > [2] Gadot, U., Derman, E., Kumar, N., Elfatihi, M.M., Levy, K.Y., & Mannor, S. . Solving Non-Rectangular Reward-Robust MDPs via Frequency Regularization, 2023.
> > >
> > > [3] Ding, D., Zhang, K., Duan, J., & Başar, T. (2022). Convergence and sample complexity of natural policy gradient primal-dual methods for constrained MDPs. Journal of Machine Learning Research.
> > >
> > > [4] Montenegro, A., Mussi, M., Papini, M., & Metelli, A. (2024). Last-Iterate Global Convergence of Policy Gradients for Constrained Reinforcement Learning.
> > >
> > > [5] Brian D. Ziebart, Andrew Maas, J. Andrew Bagnell, and Anind K. Dey. (2008) Maximum entropy inverse
> > > reinforcement learning. In AAAI Conference on Artificial Intelligence 23 (AAAI).

---

### Official Review · Reviewer_tEva · 2025-11-04

**Soundness:** 3
**Presentation:** 4
**Contribution:** 3
**Rating:** 8
**Confidence:** 3

**Summary:**

This paper addresses the issue of partial identifiability in reward learning (ReL), a consequence of when the provided feedback is insufficient and does not allow for the identification of the target reward $r^\*$. Instead of traditional approaches, the authors propose a new framework that incorporates robust optimization to minimize the loss tied to the worst possible value for $r^*$ within some feasible set.

**Strengths:**

- The formalization, explanation, and clarity of the ReL problem as a pair $(\mathcal{F},g)$ is very well written, general, relevant, and well positioned against related work. A key piece is how the ReL problem is reformulated to finding the optimal object $x\in\mathcal{X}_g$ for some application $g$, when given the uncertainty set of rewards $\mathcal{R_F}$.
- The metric used to calculate the loss, $\mathcal{I}_{\mathcal{F},g}$, or how "uninformative" $\mathcal{F}$ is for application $g$, is very useful as it effectively allows for the principled comparison of various feedback sets and/or applications.
- The algorithm instantiation and associated analysis is novel. A key piece that facilitates this is in Proposition 5.1 where the minimax problem is simplified into two convex problem.

**Weaknesses:**

- While the authors present a powerful, general framework, they do not address tractability concerns. The limitations section significantly downplays this fact and speaks broadly on it.
- The experiments, though useful in demonstrating/verifying the author's theoretical, do not have any baselines to compare against. For example, in section 4, existing approaches are discussed. It would be interesting to see how the proposed algorithm compares against these non-robust baselines.

**Questions:**

- Can you expand on what you mean on lines 240-241 and/or provide a reference speaking towards a practical example?
- You make the assumption "that the feasible set $\mathcal{R}_\mathcal{F}$ contains a strictly feasible reward $\bar{r}.$ How realistic is this in practice?
- You make the assumption "that the feasible set $\mathcal{R}_\mathcal{F}$ contains a strictly feasible reward $\bar{r}.$ How realistic is this in practice?
- Consider changing the colored text in equation 7 so that it matches the blue in Lemma E.2.

---

> ### Author Response · Authors · 2025-11-17
>
> We thank the Reviewer for recognizing the novelty of our algorithm and theoretical analysis, as well as for appreciating the notion of "uninformativeness", which we believe is a key component of our new framework. Below, we respond to the Reviewer's comments and questions.
>
> > While the authors present a powerful, general framework, they do not address tractability concerns. The limitations section significantly downplays this fact and speaks broadly on it.
>
> The crucial point is that the tractability of our robust ReL formulation (see Eq. 2) depends heavily on the specific ReL problem at stake.
>
> For instance, Section 5 presents a class of ReL problems where the feedback and the application represent a *convex* optimization problem (whose parameters can be estimated efficiently, as we demonstrate), and so the robust approach is tractable. As another example, Appendix E.5 explains how to extend Rob-ReL to other analogous ReL problems with convex constraints and objective. In addition, we also mention imitation learning (i.e., the first line in Table 2), which represents another convex objective that can be estimated efficiently, and for which the robust approach is therefore tractable.
>
> Of course, there also exist ReL problems for which our robust approach is intractable as is, and which require either additional structure or some approximation to be tackled. An example is the "constrained planning" application (see Line 3 of Table 2), whose loss contains a *max* operator, which turns Eq. (2) into an "intractable" (min-)max-max problem in general (but can become tractable if, e.g., the set of feasible policies $\Pi_{c,k}$ is finite).
>
> We have updated the limitations section to take these considerations into account.
>
> > The experiments, though useful in demonstrating/verifying the author's theoretical, do not have any baselines to compare against. For example, in section 4, existing approaches are discussed. It would be interesting to see how the proposed algorithm compares against these non-robust baselines.
>
> Our algorithm, Rob-ReL, is novel in two aspects:
>
> 1. It addresses a novel application, namely that of "assessing a policy preference" (see Table 2);
> 2. It considers a novel mix of feedback types (demonstrations, trajectory comparisons, policy comparisons).
>
> The majority of existing ReL algorithms in the literature considers multiple feedback of the same type, while none of them consider this application. As such, there is no baseline in the literature that can be meaningfully applied to the ReL problem addressed by Rob-ReL.
>
> That said, as explained in Section 4, since all existing ReL algorithms solve a ReL problem by simply drawing an arbitrary reward from the feasible set, we might consider as a baseline an algorithm that uses a specific criterion to extract an arbitrary reward $r'$ from the feasible set $\mathcal{R}\_{\mathcal{F}}$ and output the corresponding value difference $\langle\widehat{d}^{\pi^1}-\widehat{d}^{\pi^2},r'\rangle$. Specifically, one simple option would be to use gradient descent for finding the reward that minimizes the (convex) sum of the constraints errors, namely:
> $$
> \mathop{\text{argmin}}\_r f(r)=\sum\nolimits\_{i\in[m_{\text{D}]}}
>   \big(\max\nolimits\_\pi J^\pi(r;\widehat{p}\_{\text{D},i})-\langle\widehat{d}^{\pi\_{\text{D},i}},r\rangle
>   -t\_i\big)+\sum\nolimits\_{i\in[m\_{\text{TC}]}}
>   \langle d^{\omega^1\_{\text{TC,i}}}-d^{\omega^2\_{\text{TC,i}}}, r\rangle+\sum\nolimits\_{i\in [m\_{\text{PC}}]}\langle\widehat{d}^{\pi^1\_{\text{PC},i}}
>   -\widehat{d}^{\pi^2\_{\text{PC},i}}, r\rangle.
> $$
> Of course, the set of minimizers is the feasible set $\mathcal{R}_{\mathcal{F}}$, so we are simply using a practical method to break ties.
>
> If the Reviewer desires, we can execute additional simulations including this "baseline" algorithm. However, note that the prediction of Rob-ReL is necessarily better than that of this algorithm in the *worst-case*, as by definition Rob-ReL gives the *best* possible prediction in the worst case.

---

> > ### Author Response · Authors · 2025-11-17
> >
> > > Can you expand on what you mean on lines 240-241 and/or provide a reference speaking towards a practical example?
> >
> > Roughly speaking, lines 240–241 explain that our robust approach is the best possible given the available information (feedback). However, it cannot exceed the limits of that information, as its performance is constrained by the quality and quantity of the feedback.
> >
> > In other words, assume that we have a certain ReL problem where a given ReL algorithm has a worst-case error of $k$. Then our robust approach, by definition, will have an error $k'\le k$. Assume that we consider the ReL problem "successfully addressed" if the error is below a given threshold $\Delta$. Clearly, even though our robust approach is better than any other ReL algorithm in the worst case, we still have no guarantee that $k'\le \Delta$, and therefore no guarantee that this problem can be adequately addressed with the available information (i.e., feedback) .
> >
> > We have rephrased that sentence for clarity.
> >
> > > You make the assumption "that the feasible set $\mathcal{R}_{\mathcal{F}}$ contains a strictly feasible reward $\overline{r}$. How realistic is this in practice?
> >
> > It depends on the amount and "quality" of feedback available in the specific ReL problem of interest. For instance, the majority of feedback types in Table 1 (namely, those considered by Rob-ReL) satisfy this property if given as the only feedback, while this might not hold anymore when combined with many other types of feedback.
> >
> > Nevertheless, we remark that Assumption 5.1, i.e., Slater's condition, is standard in the literature (e.g., see [1,2]), and it is made only for the theoretical analysis. This means that one can still apply our algorithm in practice even if this assumption does not hold, and may still observe convergence. The issue is that there might be some "difficult" problem instances where the nice guarantees of Theorem 5.3 no longer apply.
> >
> > > Consider changing the colored text in equation 7 so that it matches the blue in Lemma E.2.
> >
> > We have updated the submission by changing the color in Eq. (7).

---

> > > ### Author Response · Authors · 2025-11-17
> > >
> > > [1] A. Nedic and A. Ozdaglar. Subgradient methods for saddle-point problems. Journal of Optimization Theory and Applications. 2009.
> > >
> > > [2] Dongsheng Ding, Kaiqing Zhang, Tamer Basar, and Mihailo Jovanovic. Natural policy gradient primal-dual method for constrained markov decision processes. In Advances in Neural Information Processing Systems 33 (NeurIPS). 2020.

---

### Official Review · Reviewer_mwRB · 2025-11-04

**Soundness:** 4
**Presentation:** 4
**Contribution:** 4
**Rating:** 8
**Confidence:** 5

**Summary:**

This paper addresses the fundamental problem of partial identifiability in Reward Learning (ReL), where limited feedback makes it impossible to uniquely recover the target reward function. The authors propose a quantitative framework that allows measuring performance degradation due to identifiability issues, moving beyond prior qualitative approaches. They introduce a robust minimax approach that optimizes for the worst-case reward in the feasible set and develop Rob-ReL, an algorithm for policy preference assessment problems with provable sample and iteration complexity guarantees. The work combines demonstrations, trajectory comparisons, and newly introduced policy comparison feedback in a mixed offline-online setting. Theoretical analysis shows polynomial complexity in relevant problem parameters, and numerical experiments illustrate the approach on a low-dimensional navigation task.

**Strengths:**

**Rigorous Theoretical Analysis.** Theorem 5.3 provides polynomial sample and iteration complexity bounds under reasonable assumptions (Slater's condition). The proof technique combining visitation distribution estimation errors with primal-dual subgradient convergence is sound. The use of RF-Express for minimax-optimal reward-free exploration is appropriate.

**Clear Presentation and Organization.** The paper is well-structured with motivation, framework, approach, algorithm, and theory presented logically. The illustrative example in Section 6 effectively conveys the main ideas visually.

**Weaknesses:**

**Limited treatment of function approximation.** The tabular setting with explicit state-action representation limits applicability to high-dimensional problems. While the authors mention neural network parameterization in related work, Rob-ReL does not incorporate function approximation

**Questions:**

How does the method scale to continuous state-action spaces with function approximation?
Is there a path toward extending Rob-ReL to deep RL settings, perhaps using neural network reward parameterization?

---

> ### Author Response · Authors · 2025-11-17
>
> We are glad that the Reviewer appreciated the quality of the presentation and the rigor of our theoretical analysis. Below, we provide responses to the Reviewer's questions.
>
> > Limited treatment of function approximation. The tabular setting with explicit state-action representation limits applicability to high-dimensional problems. While the authors mention neural network parameterization in related work, Rob-ReL does not incorporate function approximation. How does the method scale to continuous state-action spaces with function approximation?
>
> Our algorithm, Rob-ReL, is designed for the "vanilla" *tabular* setting, where no structural assumption is made on the MDP nor on the target reward. Scaling Rob-ReL to *large or continuous* state spaces while *preserving* theoretical guarantees is possible as long as additional structural assumptions on the dynamics of the MDP and on the target reward are introduced, and if some changes are made to the algorithm. To give a simple example, assume that the MDP has an infinite/continuous state space and that it is a Linear MDP [1], i.e., the transition model $p$ and the reward $r^\star$ are linear in some given $d$-dimensional feature map $\phi: \mathcal{S} \times\mathcal{A}\to [-1,+1]^d$. Then, robust reward learning can be performed with a variant of Rob-ReL that:
>
> - Replaces the occupancy measures with the feature expectations [2];
> - Replaces RF-Express with any reward-free algorithm for linear MDPs (e.g., [3]);
> - Applies the primal-dual subgradient method in the $d$-dimensional space of reward parameters.
>
> Crucially, we believe that, through a proof analogous to that of our Theorem 5.3, it should be possible to derive sample and time complexity bounds that replace the dependence on the size of the state space $S$ with a dependence on the feature dimension $d$.
>
> If instead one wants to scale Rob-ReL *without* preserving the theoretical guarantees, then we do not even need a reward-free exploration subroutine nor the structure of Linear MDPs, but assuming access to an online planner subroutine and assuming the target reward is linear in some known $\phi$ suffice. Indeed, in such a case, we could work with feature expectations instead of occupancy measures, and the online planner would allow us to estimate the feature expectations of the optimal policy in the demonstration feedback. We have updated the submission to include this discussion in Appendix E, which is referenced in Section 5.
>
> >  Is there a path toward extending Rob-ReL to deep RL settings, perhaps using neural network reward parameterization?
>
> We believe that an extension of Rob-ReL to deep RL settings is possible, but it is different from the standard RLHF approach (e.g., see [4]), where the reward is easily learned through supervised learning. Indeed, our problem setting is more complex, as we have demonstration and policy comparison feedback in addition to trajectory comparison feedback, and because we aim to solve a minmax problem w.r.t. all the rewards in the feasible set, while [4] are satisfied with an arbitrary reward in the feasible set. For these reasons, we can adopt a neural network reward parameterization (which is also useful in complex high-dimensional domains), but we still need to train it through a method like the PDSM.

---

> > ### Author Response · Authors · 2025-11-17
> >
> > [1] Jin, C., Yang, Z., Wang, Z. & Jordan, M.I.. (2020). Provably efficient reinforcement learning with linear function approximation.
> >
> > [2] Arora, S. & Doshi, P.. (2020). A Survey of Inverse Reinforcement Learning: Challenges, Methods and Progress.
> >
> > [3] Wagenmaker, A.J., Chen, Y., Simchowitz, M., Du, S. & Jamieson, K.. (2022). Reward-Free RL is No Harder Than Reward-Aware RL in Linear Markov Decision Processes.
> >
> > [4] Christiano, P.F., Leike, J., Brown, T.B., Martic, M., Legg, S., & Amodei, D. (2017). Deep Reinforcement Learning from Human Preferences.

---

### Official Review · Reviewer_PqUe · 2025-11-11

**Soundness:** 3
**Presentation:** 3
**Contribution:** 4
**Rating:** 8
**Confidence:** 3

**Summary:**

This paper develops a new framework, robust reward learning, that tackles the known problem of partial identifiability in reward learning. The framework incorporates several important forms of data for reward learning (including trajectories from an optimal policy, trajectory comparisons, and policy comparisons) and considers several potential downstream applications (including imitation, finding an optimal policy, and learning to compare preferences). The paper proposes to use worst-case performance in a downstream application, across all rewards compatible with the training data, as a metric for success.

The paper provides an algorithm for solving robust reward learning problems and provides theoretical guarantees on the sample and computational complexities. They also run some experiments in a small RL setting to demonstrate their algorithm works.

**Strengths:**

I think the paper has a number of strengths:
1. The paper addresses an *important problem*:  improving the safety of reward learning, which is a salient problem for modern AI deployments.
2. The *novel formulation* of robustness for reward learning adds upon prior work addressing this problem. The uninformativeness measure is an interesting way to quantify the difficulty of doing reward learning.
3. The formulation is *very general* and explicitly considers three important kinds of reward learning feedback. (Prior work, e.g. Skalse et al. (2023) only considered two kinds).
4. A *(somewhat) tractable algorithm* for solving the novel problem is introduced (Rob-Rel), and theoretical guarantees on sample complexity and time complexity (at least in terms of the number of iterations) are given. Further, by using duality, the paper avoided any dependence of time complexity on the size of the feasible reward set.
5. The paper *thoroughly, clearly, and fairly considers existing work*. In particular, tables 1 and 2 give clear relationships to broad prior work, and appendix *A.2* gives a thorough comparison to Skalse et al. (2023) which helped me to understand its contribution.

**Weaknesses:**

### Scalability
The proposed algorithm is polynomial in the size of the state space. This is a limited weakness, as it is common to some of the prior literature.

However, some reward learning methods have been shown to work in realistic and large or continuous state spaces. For example, Christiano et al.'s (2017) reward learning has been applied to text settings (large, discrete state spaces) and physics simulations (continuous state spaces). Laidlaw et al. (2025) applied CIRL to a large game.

I think the impact of this work would be significantly improved if evidence was given to suggest the framework and algorithm were scalable. If the algorithm is not scalable, it would be helpful to describe these limitations.

### Practicality of worst-case return
The proposed algorithm maximizes worst-case robustness relative to a feasible set. As the authors themselves suggest on line 241, there may be some reward learning problems where it is infeasible to get an acceptable worst-case reward. (In these cases, I think the uninformativeness could be a helpful metric for measuring this infeasibility, as noted in the strengths above). I expect that, in many real-world applications with large feasible reward spaces, many reward learning problems will be infeasible.
While it is not reasonable for the paper to solve this problem, I do believe it is a notable limitation, relative to e.g. a Bayesian approach that only tries to minimize expected loss over a posterior reward distribution.  Evidence that robust reward learning is feasible in real-world environments might improve the paper.


### Clarity
I found the paper to be quite difficult to parse in a number of places, although I think the paper is relatively clear given its technical density. Three potential points for improvement might be:
1. Reduce the introduction of abbreviations: "ReL", "IL", "PBIRL", and other abbreviations could be far easier to parse if set out plainly.
2. Reduce the introduction of notation, or where possible, explain in plain English what terms mean. Section 5, in particular, introduces a whole range of new notation that is hard to keep track of and could be better supported with natural language explanations.
3. (Minor) Throughout the paper, citations are given without bracketing, where bracketing would be much clearer. For example, lines 214, 121 and 105.


### References
* Christiano, P., Leike, J., Brown, T. B., Martic, M., Legg, S., & Amodei, D. (2017). Deep reinforcement learning from human preferences. arXiv preprint arXiv:1706.03741.
* Laidlaw, C., Bronstein, E., Guo, T., Feng, D., Berglund, L., Svegliato, J., Russell, S., & Dragan, A. (2025). AssistanceZero: Scalably solving assistance games. arXiv preprint arXiv:2504.07091.

**Questions:**

Can the authors provide any insight about whether worst-case reward learning can scale as well as alternative approaches (such as cooperative inverse RL, or reinforcement learning from human feedback)?

---

> ### Author Response · Authors · 2025-11-17
>
> We thank the Reviewer for recognizing the importance of the research problem addressed, the novelty and generality of the proposed formulation, and for appreciating the theoretical guarantees of the presented algorithm. Below, we respond to the Reviewer's comments and questions.
>
> > Scalability. Can the authors provide any insight about whether worst-case reward learning can scale as well as alternative approaches (such as cooperative inverse RL, or reinforcement learning from human feedback)?
>
> Our algorithm, Rob-ReL, is designed for the "vanilla" *tabular* setting, where no structural assumption is made on the MDP nor on the target reward. In this setting, a polynomial dependence on the size of state space is almost always present in RL theory is and also unavoidable  (e.g., see [1]).
>
> Scaling Rob-ReL to *large or continuous* state spaces while *preserving* theoretical guarantees is possible as long as additional structural assumptions on the dynamics of the MDP and on the target reward are introduced, and if some changes are made to the algorithm. To make an example, assume that the MDP has an infinite/continuous state space and that it is a Linear MDP [2], i.e., the transition model $p$ and the reward $r^\star$ are linear in some given $d$-dimensional feature map $\phi: \mathcal{S} \times\mathcal{A}\to [-1,+1]^d$. Then, robust reward learning can be performed with a variant of Rob-ReL that:
>
> - Replaces the occupancy measures with the feature expectations [3];
> - Replaces RF-Express with any reward-free algorithm for linear MDPs (e.g., [4]);
> - Applies the primal-dual subgradient method in the $d$-dimensional space of reward parameters.
>
> Crucially, we believe that, through a proof analogous to that of our Theorem 5.3, it should be possible to derive sample and time complexity bounds that replace the dependence on the size of the state space $S$ with a dependence on the feature dimension $d$.
>
> If instead one wants to scale Rob-ReL *without* preserving the theoretical guarantees, then we do not even need a reward-free exploration subroutine nor the structure of Linear MDPs, but assuming access to an online planner subroutine and assuming the target reward is linear in some known $\phi$ suffice. Indeed, in such a case, we could work with feature expectations instead of occupancy measures, and the online planner would allow us to estimate the feature expectations of the optimal policy in the demonstration feedback. We have updated the submission to include this discussion in Appendix E, which is referenced in Section 5.
>
> > Practicality of worst-case return
>
> We agree with the Reviewer that assessing the feasibility of our robust reward learning approach represents an important direction for future work.
>
> We would like to mention here that the infeasibility of a reward learning problem depends on two main aspects:
>
> - The amount of information $\mathcal{F}$ available on the target reward $r^\star$;
> - The kind of guarantees we aim to provide on the downstream application $g$.
>
> With an equal amount of information on $r^\star$, a Bayesian approach would provide *different* and incomparable guarantees than our robust approach. Indeed, our robust approach gives *worst-case* guarantees (intuitively, it guarantees that, whatever the target reward, the loss will never exceed the uninformativeness $\mathcal{I}_{\mathcal{F},g}$), while Bayesian approaches provide guarantees *on average*.
>
> > Clarity
>
> We thank the Reviewer for the useful suggestions. We have updated the submission by dropping abbreviation "RLHF", added an explanation in plain English to Eq. (8), and added bracketing to citations where needed.

---

> > ### Author Response · Authors · 2025-11-17
> >
> > [1] Agarwal, A., Jiang, N., & Kakade, S. M. (2019). Reinforcement Learning: Theory and Algorithms.
> >
> > [2] Jin, C., Yang, Z., Wang, Z. & Jordan, M.I.. (2020). Provably efficient reinforcement learning with linear function approximation.
> >
> > [3] Arora, S. & Doshi, P.. (2020). A Survey of Inverse Reinforcement Learning: Challenges, Methods and Progress.
> >
> > [4] Wagenmaker, A.J., Chen, Y., Simchowitz, M., Du, S. & Jamieson, K.. (2022). Reward-Free RL is No Harder Than Reward-Aware RL in Linear Markov Decision Processes.

---

### Meta-Review · Area_Chair_mtQj · 2026-01-08

**Summary:**

This paper studies reward learning (ReL) under partial identifiability, in which the target reward is unknown, and one instead has access to some relevant form of feedback such as trajectories of an optimal policy, trajectory comparisons, and policy comparisons. The paper develops a new framework called robust ReL, which is a robust minimax approach that optimizes the worst-case performance in a downstream application (for all admissible rewards) is used as success criterion. This approach yields an algorithm called Rob-ReL, which is designed for the tabular setting, which is proven to enjoy sample and computational complexity bounds that depend polynomially on relevant problem parameters. Further, its empirical performance is demonstrated on simple domains.

The reviewers unanimously agree that the paper studies an important and interesting RL settings, and support the proposed robust ReL framework, especially its underlying worst-case formulation. They also appreciate the fact that the framework unifies various feedback types and downstream objectives in one formalism. Some reviewers raised concerns regarding limited empirical evaluations on small-scale domains, and with no proper ablation study. Since this is primarily a theory-oriented paper with a technically sound algorithmic development and analysis, small-scale empirical evaluation is deemed enough to demonstrate some empirical performance. Despite some close relation between the proposed framework and the robust MDP framework, the proposed algorithmic ideas and analysis justify the technical novelty of the paper. In view of these, I recommend acceptance.

**Reviewer Concerns:**

__Scalability to more complex environments (beyond tabular).__ Some reviewers raised questions regarding the scalability of Rob-ReL to more complex environments, potentially with continuous state-spaces. As the authors argue, the primary goal of the paper is to concretely investigate the robust reward learning problem while restricting to the tabular setting, which further indicates that extensions beyond this would be the topic of follow-up work. Nevertheless, they provide an elaborate discussion in the rebuttal on how their proposed framework and algorithmic ideas could be extended to a linear MDP setting. I agree with the authors; and believe that including settings beyond tabular would make the paper quite dense in terms of technical presentation and adversely impacts the presentation. The main mission here is to develop a new framework.

There have also been questions regarding whether the approach could be combined with deep RL. These were addressed adequately and convincingly in the rebuttal.

In summary, I flag this concern as addressed.

__Limited empirical evaluations and missing baselines.__ There were comments regarding lack of baselines. As the rebuttal explains, there is no robust baseline in the literature, and non-robust baselines may not offer a meaningful and fair comparison, as Rob-ReL (developed here) is designed for a worst-case scenario. The authors appear to open to include a relevant non-robust baseline and I tend to think that its inclusion can be justified. In this regard, a reviewer thinks that the paper lacks an ablation study.

Another reviewer commented that the empirical evaluation relies on small-scale domains, which is insufficient to fully support the empirical efficacy of the proposed algorithm. The lack of extensive experimental validation is a significant weakness, particularly given the expectations of ICLR. This issue is not settled in the discussion, before its termination. In my opinion, this paper is a foundation theory paper, whose key goal is to develop a new theoretically-ground framework, and to a provably efficient algorithm. That said, an empirical evaluation is supposed to convey some initial empirical insights, and using toy domains is deemed fine. Therefore, I do not see the aforementioned concerns a key issue.

__Positioning w.r.t. related notions such as IRL, RLHF, and robust MDPs.__ The rebuttal elaborates on how the proposed framework compares to related formulations and settings from multiple aspects, notably including IRL, RLHF, and robust MDPs. One reviewer remains unconvinced that the proposed robust ReL is fundamentally different from the robust MDP framework, and the reviewer subsequently thinks that this limits the technical novelty. The two notions are indeed closely related. Yet, as the rebuttal explains --and I agree-- the robust MDP framework assumes a known uncertainty set, which is often considered to be rectangular. In contrast, in robust ReL the uncertainty (feasibility) set must be estimated, and not necessarily rectangular. While the rebuttal convincingly supports the technical novelty relative to the existing work on robust MDPs, it reveals, on the other hand, that the link between the two notions is strong and should be more clearly emphasize in the paper.

**Reviewer Scores:**

- Reviewer PqUe: The reviewer is highly positive already. Considering that the rebuttal concretely addresses their concerns, I believe the current score would be maintained.
- Reviewer mwRB: The reviewer is highly positive already (with high confidence). Considering that the rebuttal concretely addresses their concerns, I believe the current score would be maintained.
- Reviewer tEva: The reviewer is highly positive already. Considering that the rebuttal concretely addresses their concerns --especially the one on including non-robust baselines--, I believe the current score would be maintained.
- Reviewer kR8k: This reviewer is least positive but also least confident. Beside some technical questions --which are sufficiently addressed in the rebuttal--, the reviewer listed two key weaknesses. I found the authors' response to the raised concerns highly convincing, which further indicates the one of the key concerns came from a misunderstanding. Given the rebuttal and the experiments of Appendix F2, I believe that the rebuttal has convincingly addressed all key concerns and technical questions. Thus, I believe would increase the score. If I were to step in, I would reasonably increase the score to, at least, 4 on their behalf.
- Reviewer Kgr5: The reviewer raised a long list of comments and questions on various aspects. The reviewer later confirmed that some comments are convincingly covered in the rebuttal (e.g., some technical questions, questions related to comparison with similar settings, and those related to performance bounds). However, the reviewer still thinks that more extensive experiments must have been included, together with an ablation study. Also, the reviewer still thinks that the novelty is limited considering the existing developments for robust MDPs and inverse RL. Thus, I tend to think that the reviewer would maintain the score.

---

### Decision · Program_Chairs · 2026-01-26

Accept (Poster)